# Mapping the suitability of groundwater dependent vegetation in a semi-arid Mediterranean area

Inês Gomes Marques[1]; João Nascimento[2]; Rita M. Cardoso[1]; Filipe Miguéns[2]; Maria Teresa Condesso de Melo[2]; Pedro M. M. Soares[1]; Célia M. Gouveia[1]; Cathy Kurz Besson[1]

[1] Instituto Dom Luiz; Faculty of Sciences, University of Lisbon, Campo Grande, Ed. C8, 1749-016, Lisbon, Portugal

[2] CERIS; Instituto Superior Técnico, University of Lisbon, 1049-001, Lisbon, Portugal

*Correspondence to*: Inês Gomes Marques (icgmarques@fc.ul.pt or icgmarques@isa.ulisboa.pt)

**Abstract**

Mapping the suitability of groundwater dependent vegetation in semi-arid Mediterranean areas is fundamental for the sustainable management of groundwater resources and groundwater dependent ecosystems (GDE) under the risks of climate change scenarios. For the present study the distribution of deep-rooted woody species in southern Portugal was modeled using climatic, hydrological and topographic environmental variables. To do so, *Quercus suber, Quercus ilex* and *Pinus pinea* were used as proxy species to represent the Groundwater Dependent Vegetation (GDV). Model fitting was performed between the proxy species Kernel density and the selected environmental predictors using 1) a simple linear model and 2) a Geographically Weighted Regression (GWR), to account for auto-correlation of the spatial data and residuals. When comparing the results of both models, the GWR modelling results showed improved goodness of fitting, as opposed to the simple linear model. Climatic indices were the main drivers of GDV density, followed with a much lower influence by groundwater depth, drainage density and slope. Groundwater depth did not appear to be as pertinent in the model as initially expected, accounting only for about 7% of the total variation against 88% for climate drivers

The relative proportion of model predictor coefficients was used as weighting factors for multicriteria analysis, to create a suitability map to the GDV in southern Portugal showing where the vegetation most likely relies on groundwater to cope with aridity. A validation of the resulting map was performed using independent data of the Normalized Difference Water Index (NDWI) a satellite-derived vegetation index. June, July and August of 2005 NDWI anomalies, to the years 1999-2009, were calculated to assess the response of active woody species in the region after an extreme drought. The results from the NDWI anomalies provided an overall good agreement with the suitability to host GDV. The model was considered reliable to predict the distribution of the studied vegetation.

The methodology developed to map GDV's will allow to predict the evolution of the distribution of GDV
according to climate change and aid stakeholder decision-making concerning priority areas of water
resources management.
**Keywords:** Groundwater dependent vegetation, aridity, agroforestry, suitability map, Normalized
Difference Water Index

## 1 Introduction

Mediterranean forests, woodlands and shrublands, mostly growing under restricted water availability, are one of the terrestrial biomes with higher volume of groundwater used by vegetation (Evaristo and McDonnell, 2017). Future predictions of decreased precipitation (Giorgi and Lionello, 2008; Nadezhdina et al., 2015), decreased runoff (Mourato et al., 2015) and aquifer recharge (Ertürk et al., 2014; Stigter et al., 2014) in the Mediterranean region threaten the sustainability of groundwater reservoirs and the corresponding dependent ecosystems. Therefore, a sustainable management of groundwater resources and the Groundwater Dependent Ecosystems (GDE) is of crucial importance.

A widely used classification of GDE was proposed by Eamus et al. (2006). This classification distinguishes three types: 1) Aquifer and cave ecosystems, which include all subterranean waters; 2) Ecosystems reliant on emerging groundwater (e.g. estuarine systems, wetlands; riverine systems) and 3) Ecosystems reliant on resident groundwater (e.g. systems where plants remain physiologically active during extended drought periods, without a visible water source). Mapping GDE constitutes a first and fundamental step to their active management. Several approaches have been proposed, from local field surveys measuring plant transpiration of stable isotopes (Antunes et al. 2018) up to larger spatial scales involving remote sensing techniques (e.g. Normalized Difference Vegetation Index – NDVI) (Barron et al., 2014; Eamus et al., 2015; Howard and Merrifield, 2010), remote-sensing combined with ground-based observations (Lv et al., 2013), geographic information system (GIS) (Pérez Hoyos et al., 2016a) GIS combining field surveys (Condesso de Melo et al., 2015), or even statistical approaches (Pérez Hoyos et al., 2016b).

Despite of a wide-ranging body of literature reviewing GDE's topics (Doody et al., 2017; Dresel et al., 2010; Münch and Conrad, 2007), most of regional scale studies do not include Mediterranean regions. Moreover, studies on ecosystems relying on resident groundwater frequently only focused on riparian environments (Lowry and Loheide, 2010; O'Grady et al., 2006), with few examples in Mediterranean areas (del Castillo et al., 2016; Fernandes, 2013; Hernández-Santana et al., 2008; Mendes et al., 2016). There is a clear knowledge gap on the identification of phreatophyte species reliant on resident groundwater and their associated vegetation (Robinson, 1958) in the Mediterranean region and the management actions that should be taken to decrease the adverse effects of climate change.

In the driest regions of the Mediterranean basin, the persistent lack of water during the entire summer periods gave an adaptive advantage to the vegetation that could either avoid or escape drought by reaching deeper stored water up to the point of entirely relying on groundwater (Chaves et al., 2003; Canadell et al., 1996; Miller et al., 2010). This drought-avoiding strategy is often associated to the development of a dimorphic root system in woody species (Dinis 2014, David et al., 2013) or to hydraulic lift and/or hydraulic redistribution mechanisms (Orellana et al., 2012). Those mechanisms provide the ability to move water from deep soil layers, where water content is higher, to more shallow layers where water content is lower (Horton and Hart, 1998; Neumann and Cardon, 2012). Hydraulic lift and redistribution have been reported for several woody species of the Mediterranean basin (David et al.,

2007; Filella and Peñuelas, 2004) and noticeably for Cork oak (*Quercus suber* L.) (David et al., 2013;
Kurz-Besson et al., 2006; Mendes et al., 2016).
Mediterranean cork oak woodlands (Montados) are agro-silvo-pastoral systems considered as semi-
natural ecosystems of the southwest Mediterranean basin (Joffre et al., 1999) that have already been
referenced has a groundwater dependent terrestrial ecosystem (Mendes et al., 2016). Montados must be
continually maintained through human management by thinning, understory use through grazing,
ploughing and shrub clearing (Huntsinger and Bartolome, 1992) to maintain a good productivity,
biodiversity and ecosystems service (Bugalho et al., 2009). In the ecosystems of this geographical area,
the dominant tree species are the cork oak (*Quercus suber* L.) and the Portuguese holm oak (*Quercus ilex*
subs *rotundifolia* Lam.*) (Pinto-Correia et al., 2011). Additionally, stone pine (*Pinus pinea* L.) has become
a commonly co-occurrent species in the last decades (Coelho and Campos, 2009). The use of groundwater
has been frequently reported for both *Pinus* (Antunes et al. 2018; Filella and Peñuelas, 2004; Grossiord et
al., 2016; Peñuelas and Filella, 2003) and *Quercus* genre (Barbeta and Peñuelas, 2017; David et al., 2007,
2013, Kurz-Besson et al., 2006, 2014; Otieno et al., 2006). Furthermore, the contribution of groundwater
to tree physiology has been shown to be of a greater magnitude for *Quercus* sp. as compared with *Pinus*
sp. (del Castillo et al., 2016; Evaristo and McDonnell, 2017).
*Q. suber* and *Q. ilex* have been associated with high resilience and adaptability to hydric and thermic
stress, and to recurrent droughts in the southern Mediterranean basin (Barbero et al., 1992). In Italy and
Portugal, during summer droughts *Q. ilex* used a mixture of rain-water and groundwater and was able to
take water from very dry soils (David et al., 2007; Valentini et al., 1992). An increasing contribution of
groundwater in the summer has also been shown for this species (Barbeta et al., 2015). Similarly, *Q.*
*suber* showed a seasonal shift in water sources, from shallow soil water in the spring to the beginning of
the dry period followed by a progressive higher use of deeper water sources throughout the drought
period (Otieno et al., 2006). In addition, the species roots are known to reach depths as deep as 13m in
southern Portugal (David et al., 2004). *P. pinea* has been recently included in the facultative phreatophyte
species (Antunes et al. 2018). This species shows a very similar root system (Montero et al., 2004) as
compared to cork oak (David et al., 2013), with large sinker roots reaching 5 m depth (Canadell et al.,
1996). Given the information available on water use strategies by the phreatophyte arboreous species of
the cork oak woodlands, *Q. ilex*, *Q. suber* and *P. pinea* were considered as proxies for arboreous
vegetation that belongs to GDE relying on resident groundwater (from here onwards designed as
Groundwater Dependent Vegetation – GDV).
GDV of the Mediterranean basin is often neglected in research. Indeed, still little is known about the
GDV distribution, but research has already been done on the effects of climate change in specific species
distribution, such as *Q. suber*, in the Mediterranean basin (Duque-Lazo et al., 2018; Paulo et al., 2015).
While the increase in atmospheric $CO_2$ and the rising temperature can boost tree growth (Barbeta and
Peñuelas, 2017; Bussotti et al., 2013; Sardans and Peñuelas, 2004), water stress can have a counteracting
effect on growth of both *Quercus ilex* (López et al., 1997; Sabaté et al., 2002) and *P. pinaster* (Kurz-
Besson et al., 2016). Therefore, it is of crucial importance to identify geographical areas where subsurface
GDV is present and characterize the environmental conditions this vegetation type is thriving in. This
would contribute to the understanding of how to manage these species under unfavorable future climatic
conditions.
The aim of this study was to address the mentioned gaps by creating a suitability map of the arboreous
phreatophyte species in southern Portugal, traducing their potential dependency on groundwater. We used
an integrated multidisciplinary methodology combining a geospatial modeling approach based on the
Geographically Weighted Regression (GWR) and a GIS multicriteria analysis approach, both relying on
forest inventory, edaphoclimatic conditions and topographic information. We expected this new
integrated procedure to grant a more reliable estimation of the vegetation dependency on groundwater
sources at the regional scale.
The Mapping methodology was based on the occurrence of known subsurface phreatophyte species and
well-known environmental conditions affecting water resources availability. Several environmental
predictors were selected according to their expected impact on water use and storage and then used in
GWR to model the density of *Q. suber*, *Q. ilex* and *P. pinea* occurrence in the Alentejo region (NUTSII)
of southern Portugal. To our knowledge, very few applications of GWR have been used to model species
distribution and only recently its use has spread in ecological research (Hu et al., 2017; Li et al., 2016;
Mazziotta et al., 2016). The coefficients obtained from the model equation for each predictor and
expressed as proportion of total sum of absolute coefficients were used as weights to build the suitability
map with GIS multi-factor analysis, after reclassifying each relevant environmental driver. The resulting
map was validated using the remote sensed vegetation index NDWI.
Based on former knowledge gathered from field surveys conducted in the region (Antunes et al. 2018,
Condesso de Melo et al., 2015, Kurz-Besson et al. 2006 & 2014, Otieno et al. 2006, David et al. 2013,
Pinto et al. 2013), on environmental conditions and the species ecophysiological needs, we hypothesized
that 1) groundwater depth together with climatic conditions play one of the most important environmental
roles in GDV's distribution and 2) groundwater depth between 1.5 and 15 m associated with xeric
conditions should favor a higher density of GDV and thus a larger use of groundwater by the vegetation.


**2 Material and Methods**

**2.1 Study area**
The administrative region of Alentejo (NUTSII) (fig01) covers an area of 31 604.9 km$^2$, between 37.22°
and 39.39° N in latitude and between 6.55° and 9.00° W in longitude. This study area is characterized by a
Mediterranean temperate mesothermic climate with hot and dry summers, defined as Csa in the Köppen
classification (APA, n.d.; ARH Alentejo, 2012a, 2012b). It is characterized by a sub-humid climate,
which has recently quickly drifted to semi-arid conditions (Ministério da Agricultura do Mar do
Ambiente e do Ordenamento do Território, 2013). A large proportion of the area (above 40%) is covered
by forestry systems (Autoridade Florestal Nacional and Ministério da Agricultura do Desenvolvimento
Rural e das Pescas, 2010) providing a high economical value to the region and the country (Sarmento and
Dores, 2013).

**2.2 Kernel Density estimation of GDV**
Presence datasets of *Quercus suber, Quercus ilex* and *Pinus pinea* of the last Portuguese forest inventory
completed in 2010 (ICNF, 2013) were used to calculate Kernel density (commonly called heat map) as a
proxy for GDV suitability. The inventory registered the occurrence of each species on a 500m mesh grid
resolution, corresponding to a maximum occurrence of 4 counts per km$^2$. Only data points with one of the
three proxy species selected as primary and secondary occupation were used. The resulting Kernel density
was weighted according to tree cover percentage and was calculated using a quartic biweight distribution
shape, a search radius of 10 km, and an output resolution of 0.018 degrees, corresponding to a cell size of
1km. This variable was computed using QGIS version 2.14.12 (QGIS Development Team, 2017).


**2.3 Environmental variables**
Species distribution is mostly affected by limiting factors controlling ecophysiological responses,
disturbances and resources (Guisan and Thuiller, 2005). To characterize the study area in terms of GDV's
suitability, environmental variables expected to affect GDV's density were selected according to their
constraint on groundwater uptake and soil water storage. Within possible abiotic variables, landscape
topography, geology, groundwater availability and regional climate were considered. The twelve selected
variables for modeling purposes, retrieved from different data sources, are listed in Table 1. The software
used in spatial analysis was ArcGIS® software version 10.4.1 by Esri and R program software version
3.4.2 (R Development Core Team, 2016).

**2.3.1 Slope and soil characteristics**
The NASA and METI ASTER GDEM product was retrieved from the online Data Pool, courtesy of the
NASA Land Processes Distributed Active Archive Center (LP DAAC), USGS/Earth Resources
Observation and Science (EROS) Center, Sioux Falls, South
Dakota, https://lpdaac.usgs.gov/data_access/data_pool. Spatial Analyst Toolbox was used to calculate the
slope from the digital elevation model. Slope was used as proxy for the identification of shallow soil
water interaction with vegetation.
The map of soil type was obtained from the Portuguese National Information System for the Environment
- SNIAmb (© Agência Portuguesa do Ambiente, I.P., 2017) and uniformized to the World Reference
Base with the Harmonized World Soil Database v 1.2 (FAO et al., 2009). The vector map was converted
to raster using the Conversion Toolbox. To reduce the analysis complexity involving the several soil
types present in the map, soil types were regrouped in three classes, according to their capacity to store or
drain water (Table A1 in appendix A). The classification was based on the characteristics of each soil unit
(available water storage capacity, drainage and topsoil texture) from the Harmonized World Soil
Database v 1.2 (FAO et al., 2009). In the presence of dominant soil with low drainage capacity, a high
clay fraction in the top soil and a high available water content, lower scores were given in association to
decreased suitability for GDV by favoring non-GDV species. Otherwise, when soil characteristics
suggested water storage at deeper soil depths, lower water content, drainage and sandy topsoil texture,
higher scores were given.
Effective soil thickness (Table 1) was also considered for representing the maximum soil depth explored
by the vegetation roots. It constrains the expansion and growth of the root system, as well as the available
amount of water that can be absorbed by roots.

**2.3.2 Groundwater availability**
Root access to water resources is one of the most limiting factors for GDV's growth and survival,
especially during the dry season. The map of depth to water table was interpolated from piezometric
observations from the Portuguese National Information System on Water Resources (SNIRH) public data
base (http://snirh.apambiente.pt, last accessed on March 31[st] 2017) and the Study of Groundwater
Resources of Alentejo (ERHSA) (Chambel et al., 2007). Data points of large-diameter wells and
piezometers were retrieved for the Alentejo region (fig02) and sorted into undifferentiated, karst or
porous geological types to model groundwater depth. In the studied area, piezometers are exclusively
dedicated small diameter boreholes for piezometric observations, in areas with high abstraction volumes
for public water supply. Large diameter wells in this region are usually low yielding and mainly devoted
to private use and irrigation. Due to the large heterogeneity of geological media, groundwater depth was
calculated separately for each sub-basin. A total of 3158 data points corresponding to large wells and
piezometers were used, with uneven measurements between 1979 and 2017. For each piezometer an
average depth was calculated from the available observations and used as a single value. In areas with
undifferentiated geological type, piezometric level and elevation were highly correlated (>0.9), thus a
linear regression was applied to interpolate data. Ordinary kriging was preferred for the interpolation of
karst and porous aquifers, combining large wells and piezometric data points. The ordinary kriging was
calculated using a semi-variogram in which the sill, range and nugget were optimized to create the best fit
of the model to the data. To build a surface layer of the depth to water table, the interpolated surface of
the groundwater level was subtracted from the digital elevation model. Geostatistical Analyst ToolBox
was used for this task.
Drainage density is a measure of how well the basin is drained by stream channels. It is defined as the
total length of channels per unit area. Drainage density was calculated for a 10km grid size for the
Alentejo region, by the division of the 10km square area (A) in km$^2$ by the total stream length (L) in km,
as in Eq. (1).

230        $D = \frac{L}{A},$                                                    (1)


### 232  2.3.3 Regional Climate

Temperature and precipitation datasets were obtained from the E-OBS
(http://eca.knmi.nl/download/ensembles/ensembles.php, last accessed on March 31st 2017) public
database (Haylock et al., 2008). Standardized Precipitation Evapotranspiration Index (SPEI), Aridity
Index ($A_i$) and Ombrothermic Indexes were computed from long-term (1951-2010) monthly temperature
and precipitation observations. The computation of potential evapotranspiration (PET) was performed
according to Thornthwaite (1948) and was calculated using the SPEI package (Beguería and Vicente-
Serrano, 2013) in R program.
SPEI multi-scalar drought index (Vicente-Serrano et al., 2010) was calculated over a 6 month interval to
characterize drought severity in the area of study using SPEI package (Beguería and Vicente-Serrano,
2013) for R program. SPEI is based on the normalization of the water balance calculated as the difference
between cumulative precipitation and PET for a given period at monthly intervals. Normalized values of
SPEI typically range between -3 and 3. Drought events were considered as severe when SPEI values were
between -1.5 and -1.99, and as extreme with values below -2 (Mckee et al., 1993). Severe and extreme
SPEI predictors were computed as the number of months with severe or extreme drought, counted along
the 60 years of the climate time-series.
While the SPEI index used in this study identifies geographical areas affected with more frequent extreme
droughts, the Aridity index distinguishes arid geographical areas prone to annual negative water balance
(with low $A_i$ value) to more mesic areas showing positive annual water balance (with high $A_i$ value). $A_i$
gives information related to evapotranspiration processes and rainfall deficit for potential vegetative
growth. It was calculated following Eq. (2) according to Middleton et al. (1992), where PET is the
average annual potential evapotranspiration and P is the average annual precipitation, both in mm for the
60 years period of the climate time-series. Dry lands are defined by their degree of aridity in 4 classes:
Hyperarid ($A_i$<0.05); Arid (0.05<$A_i$<0.2); Semi-arid (0.2<$A_i$<0.5) and Dry Subhumid (0.5<$A_i$<0.65)
(Middleton et al., 1992).
$A_i = \dfrac{P}{PET}$ ,            (2)
Ombrothermic Indexes were used to better characterize the bioclimatology of the study region (Rivas-
Martínez et al., 2011), by evaluating soil water availability for plants during the driest months of the year.
Four ombrothermic indexes were calculated according to a specific section of the year stated in Table 1,
and following Eq. (3), where Pp is the positive annual precipitation (accumulated monthly precipitation
when the average monthly mean temperature is higher than 0°C) and Tp is the positive annual
temperature (total in tenths of degrees centigrade of the average monthly temperatures higher than 0°).
Ombrothermic index presenting values below 2 for the analyzed months, can be considered as
Mediterranean bioclimatically. For non-Mediterranean areas, there is no dry period in which, for at least
two consecutive months, the precipitation is less than or equal to twice the temperature.
$O = \dfrac{Pp}{Tp}$ ,            (3)

**2.4 Selection of model predictors**
The full set of environmental variables was evaluated as potential predictors for the suitability of GDV
(based on the Kernel density of the proxy species). A preliminary selection was carried out, first by
computing Pearson's correlation coefficients between environmental variables and second by performing
a Principal Components Analysis (PCA) to detect multicollinearity. Covariates were discarded for
modeling according to a sequential procedure. Whenever pairs of variables presented a correlation value
above 0.4, the variable with the highest explained variance on the first axis of the PCA was selected. In
addition, selected variables had to show the lowest possible correlation values between them. Variables
showing low correlations and explaining a higher cumulative proportion of variability with the lowest
number of PCA axis were later selected as predictors for modeling. PCA was performed using the GeoDa
Software (Anselin et al., 2006) and Pearson's correlation coefficients were computed with Spatial Analyst
Tool .

**2.5 Model development**
When fitting a linear regression model based on the selected variables, the normal distribution and
stationarity of the response variable and residuals must be assured.
The Kernel density of the proxy GDV species, *Q. suber*, *Q. ilex* and *P. pinea,* showed a skewed normal
distribution. Therefore, a square-root transformation of the data was applied on the response variable,
before model fitting. To be able to compare the resulting model coefficients and use them as weighting
factors of the multi-criteria analysis to build the suitability map, the predictor variables were normalized
using the z-score function. This allows to create standardized scores for each variable, by subtracting the
mean of all data points from each individual data point, then dividing those points by the standard
deviation of all points, so that the mean of each z-predictor is zero and the deviation is 1.
Spatial autocorrelation and non-stationarity are common when using linear regression on spatial data. To
overcome these issues, the Geographically Weighted Regression (GWR) was used. This extension of the
Ordinary Least Squares (OLS) linear regression considers the spatial non stationarity in variable
relationships and allows the use of spatially varying coefficients while minimizing spatial autocorrelation
(Stewart Fotheringham et al., 1996). In this study, simple linear regression and GWR were both applied to
the dataset and their performances compared. Models were fitted on a 5% random subsample of the entire
dataset (reaching a total of 6214 selected data points), due to computational restrictions and to decrease
the spatial autocorrelation effect (Kühn, 2007). This methodology has already been applied with a
subsample of 10%, with points distant 10km from each other (Bertrand et al., 2016). In spite of the
subsampling, the minimum and maximum distance between two random data points were, respectively,
3.6 km and 16.7 km, providing a good representation of local heterogeneity, as shown in figures 05 and
06. An additional analysis showing an excellent agreement between the two datasets is presented in
FigA1 in appendix A.
Initially the model was constructed containing all selected predictors through the PCA and Pearson's
correlation analysis. Afterwards, predictors were sequentially discarded to ascertain the model presenting
lower second-order Akaike Information Criteria (AICc) and higher quasi-global $R^2$ chosen to predict the
suitability of GDV.
Adaptive Kernel bandwidths for the GWR model fitting were used due to the spatial irregularity of the
random subsample. Local search radius were obtained by minimizing the CrossValidation score (Bivand
et al., 2008) and thus minimizing the error of the local regressions. To analyze the performance of the
GWR model alone, the local and global adjusted $R^2$ were considered. To compare between the GWR
model and the simple linear model, the distribution of the model residuals was used to identify clustered
values as well as the AICc. The spatial autocorrelation of the models residuals was evaluated with the
Moran's I test (Moran, 1950) calculated from the Spatial Statistics Tool, and also graphically. GWR
model was fitted using the *spgwr* package from R program (Bivand and Yu, 2017).

**2.6 Suitability map building**

To create the suitability map all predictor layers included in the GWR model were classified, similarly to
Condesso de Melo et al. (2015) and Aksoy et al. (2017) . The likelihood of an interaction between the
vegetation and groundwater resources was scored from 1 to 3 for each predictor. Scores were assigned
after bibliographic review and expert opinion. The higher the score, the higher the likelihood, 1
corresponding to a weak likelihood and 3 indicating very high likelihood.
Groundwater depth was divided in two classes, according to the accessibility to shallow soil water above
1.5 m and the maximum rooting depth for Mediterranean woody species reaching 13 m, reported by
Canadell et al. (1996). Throughout the manuscript water between 0 and 1.5 m depth was designated as
shallow soil water, while water below 1.5 m depth was considered as groundwater. The depth class
between 0 and 1.5m was based on the riparian vegetation in semi-arid Mediterranean areas which is
mainly composed of shrub communities (Salinas et al., 2000) and presents a mean rooting depth of 1.5m
(Silva and Rego, 2004). The most common tree species rooting depth in riparian ecosystems is normally
similar to the depth of fine sediment not reaching gravel substrates (Singer et al., 2012) and not reaching
levels as deep as deep-rooted species. The minimum score was given to areas where groundwater depth
was too shallow (below 1.5 m) considered to belong to emerging groundwater dependent vegetation.
Areas with steep slope were considered to have superficial runoff and less recharge and influence
negatively tree density (Costa et al., 2008). Those areas were treated as less suitable to GDV. Values of
the Ombrothermic Index of the summer quarter and the immediately previous month ($O_4$) were split in 3
classes according to Jenks natural breaks, with higher suitability corresponding to higher aridity. The
higher values of $A_i$, corresponding to lower aridity had a score of 1, because a higher humid environment
would decrease the necessity of the arboreous species to use deep water sources. Accordingly, an increase
in aridity (lower values of $A_i$) has already been shown to increase tree decline (Waroux and Lambin,
2012) and so lower $A_i$ values corresponded to a score of 3, leaving the score 2 to intermediate values of
$A_i$. Drainage density scoring was based on the drainage capability of the water through the
hydrographical network of the river. A low drainage density (below 0.5) implies a high loss of water
through runoff along the hydrographic network. This water lost for shallow soil horizons would be less
available to the vegetation thus favoring a higher use of water from deep groundwater reservoirs
(Rodrigues, 2011).
A direct compilation of the predictor layers could have been performed for the multicriteria analysis.
However, some predictors might have a stronger influence on GDV's distribution and density than others.
Therefore, there was a need to define weighting factors for each layer of the final GIS multicriteria
analysis. Yet, due to the intricate relations between all environmental predictors and their effects on the
GDV, experts and stakeholders suggested very different scoring for a same layer. Instead the relative
proportion of each predictor was used locally, according to the GWR model (Eq. 4) as weighting factors.
The final GIS multicriteria analysis was performed using the Spatial Analyst Tool by applying local
model equations obtained for each of the 6214 coordinates of the Alentejo map (Eq.4),
*$S_{GDV}$= Intercept + $coef_{p1}$ \* [reclassified value $X_1$] + $coef_{p2}$ \* [reclassified value $X_2$] + $coef_{p3}$ \**
*[reclassified value $X_3$] + ...,*
*(4)*
with $S_{GDV}$ representing the suitability to Groundwater Dependent Vegetation, brackets representing the
reclassified GIS X layer corresponding to the scoring and *$coef_x$* indicating the relative proportion for the
predictor *x* calculated as the ratio between the modulus of the local coefficient *x* and the sum of the
modulus of all local coefficients..
According to this equation, lower values indicate a lower occurrence of groundwater use representing a
lower GDV suitability while higher values correspond to a higher use of groundwater representing a
higher GDV suitability. To allow for an easier interpretation, the data on suitability to GDV was
subsequently classified based on their distribution value, according to Jenks natural breaks. This resulted
in 5 suitability classes: "Very poor", "Poor", "Moderate", "Good" and "Very Good".

## 2.7 Map evaluation

Satellite derived remote-sensing products have been widely used to follow the impact of drought on land cover and the vegetation dynamics (Aghakouchaket al. 2015). Vegetation indexes offer excellent tools to assess and monitor plant changes and water stress (Asrar et al. 1989). The Normalized Difference Water Index (NDWI) (Gao, 1996) is a satellite-derived index that aims to estimate fuel moisture content (Maki et al., 2004) and leaf water content at canopy level, widely used for drought monitoring (Anderson et al., 2010, Gu et al., 2007; Ceccato et al., 2002a). This index was chosen to be more sensitive to canopy water content and a good proxy for water stress status in plants. Moreover, NDWI has been shown to be best related to the greenness of Cork oak woodland's canopy, expressed by the fraction of intercepted photosynthetically active radiation (Cerasoli et al., 2016).

In order to validate the GDV suitability map obtained in our study, we calculated anomalies of the Normalized Difference Water Index (NDWI) (Gao, 1996) between an extreme dry year (2005) and the median value of the surrounding 10 year period (1999-2009). NDWI is computed using the near infrared (NIR) and the short-wave infrared (SWIR) reflectance, which makes it sensitive to changes in liquid water content and in vegetation canopies (Gao, 1996; Ceccato et al., 2002a, b). The index computation (Eq. 5) was further adapted by Gond et al. (2004) to SPOT-VEGETATION instrument datasets, using NIR (0.84 µm) and MIR (1.64 µm) channels, as described by Hagolle et al. (2005).

$$NDWI = \frac{\rho_{NIR} - \rho_{MIR}}{\rho_{NIR} + \rho_{MIR}}. \tag{5}$$

Following Eq. (5), NDWI data was computed using B3 and MIR data acquired from VEGETATION instrument on board of SPOT4 and SPOT5 satellites. Extraction and corrections procedures applied to optimize NDWI series are fully described in Gouveia et al. (2012).

The NDWI anomaly was computed as the difference between NDWI observed in June, July and August of 2005 and the median NDWI for the considered month for the period 1999 to 2009. June was selected to provide the best signal from a still fully active canopy of woody species while the herbaceous layer had usually already finished its annual cycle and dried out. The hydrological year of 2004/2005 was characterized by an extreme drought event over the Iberian Peninsula, where less than 40% of the normal precipitation was registered in the southern area (Gouveia et al., 2009). Thus, in June 2005 the vegetation of the Alentejo region was already coping with an extreme long-term drought, which was well captured by the anomaly of the NDWI index (negative values), as formerly shown by Gouveia et al. 2012.

## 2.8 Sensitivity analysis

Sensitivity analyses are conducted to identify model inputs that cause significant impact and/or uncertainty in the output. They can be used to identify key variables that should be the focus of attention to increase mode robustness in future research or to remove redundant inputs from the model equation because they do not have significant impact on the model output. Based on bootstrapping simulations (Tian et al., 2014), a sensitivity analysis was conducted on the GWR model by perturbing one input

predictor at time while keeping the rest of the equation unperturbed. To simulate perturbations, 10000
values were randomly selected within the natural range of each input variable observed in the Alentejo
region. Those random values were then used to run 10000 simulations of the local equation of the GWR
model for each of the 6214 coordinates of the geographical area. Local outputs corresponding to the
predicted GDV density were then calculated for each perturbed input variable ($A_i$, $O_4$, W, D and s). The
range of output values was calculated to reflect the sensibility of the model for the perturbed input
variable. The overall sensibility of the model to all input variables was estimated as the absolute
difference between the minimum output value and the sum of maximum output values of all predictors,
thus representing the maximum possible output range observed after perturbing all predictors.



**3 Results**

**3.1 Kernel Density**

Within the studied region of Portugal, the phreatophyte species *Quercus suber, Quercus ilex* and *Pinus pinea* were not distributed uniformly throughout the territory. Areas with higher Kernel density (or higher distribution likelihood) were mostly spread between the northern part of Alentejo region and the western part close to the coast, with values ranging between 900 and 1200 occurrences in 10 km search radius (fig03). Two clusters of high density also appeared below the Tagus river. The remaining study area presented mean density values, with very low densities in the area of the river Tagus and in the center south.

**3.2 Environmental conditions**

The exploratory analysis of the variables performed through the PCA and the Pearson's correlation matrix confirmed the presence of multicollinearity. From the initial variables (Table 1), Thickness (T), number of months with severe and extreme SPEI (respectively, $SPEI_s$ and $SPEI_e$), Annual Ombrothermic Index (O), Ombrothermic Index of the hottest month of the summer quarter($O_1$) and Ombrothermic Index of the summer quarter ($O_3$) were discarded, while the variables slope (s), drainage density (D), soil type ($S_t$), groundwater depth (W), $A_i$ and $O_4$ were maintained for analysis (figA2 and Table A2 in appendix). A sequential removal of one predictor from the initial modeling including six variables was performed (Table 2), after which the model was reduced to 5 variables. Therefore, out of the initial 12 variables considered (fig04) to explain the variation of the Kernel density of GDV in Alentejo, the following variables were endorsed: $A_i$, $O_4$, W, D and s.

In most part of the Alentejo region, slope was below 10% (fig04e) and coastal areas presented the lowest values and variability. Highest values of groundwater depth (fig04c), reaching a maximum of 255 m, were found in the Atlantic margin of the study area, mainly in Tagus and Sado river basins. Several other small and confined areas in Alentejo also showed high values, corresponding to aquifers of porous or karst geological types. Most of the remaining study area showed groundwater depths ranging between 1.5 m and 15 m. Figures 04a and 04b indicate the southeast of Alentejo as the driest area, given by minimum values of $A_i$ (0.618), and much higher potential evapotranspiration than precipitation. Besides, $O_4$ presented a maximum value (1.166) for this region (meaning that soil water availability was not compensated by the precipitation of the previous M-J-J-A months). This is also supported by the higher drainage density in the southeast which indicates a lower prevalence of shallow soil water due to higher stream length by area.

Combining all variables, it was possible to distinguish two sub-regions with distinct conditions: the southeast of Alentejo and the Atlantic margin. The latter is mainly distinguished by its low slope areas, shallower groundwater and more humid climatic conditions than the southeast of Alentejo.


### 3.3 Regression models

The best model to describe the GDV distribution was found through a sequential discard of each variable
(Table 2) and corresponded to the model with a distinct lower AICc (18050.76) than the second lowest
AICc (27389.74) and showed an important increase in quasi-global $R^2$ (from 0.926 for the second best
model to 0.992 for the best one). The best model fit was obtained with $A_i$, $O_4$, W, D and s. This final
model was then applied to the GIS layers to map the suitability of GDV in Alentejo, according to Eq. 6.
$S_{GDV} = Intercept + A_i\ coef_p * [reclassified\ A_i\ value] + O_4\ coef_p * [reclassified\ O_4\ value] + W\ coef_p *$
$[reclassified\ W\ value] + D\ coef_p * [reclassified\ D\ value] + s\ coef_p * [reclassified\ s\ value],$

461                                                                                                      $(6)$

Local adjusted $R^2$ of the GWR model was highly variable throughout the study area, ranging from 0 to
0.99 (fig05), however the local $R^2$ values below 0.5 corresponded to only 0.3% of the data. The lower $R^2$
values were distributed throughout the Alentejo area, with no distinct pattern. The overall fit of the GWR
model was high (Table 3). The adjusted regression coefficient indicated that 99% of the variation in the
data was explained by the GWR model, while only 2% was explained by the simple linear model (Table
3). Accordingly, GWR had a substantially lower AICc when compared with the simple linear model,
indicating a much better fit.
The spatial autocorrelation given by the Moran Index (Griffith, 2009; Moran 1950) retrieved from the
geospatial distribution of residual values was significant for both the GWR and the linear models,
indicating that observations are geospatially dependent on each other to a certain level . However, this
dependence was substantially lower for the GWR model than for the linear model (z-score of 50.24 and
147.56 respectively). In the GWR model (fig06a) the positive and negative residual values were much
more randomly scattered throughout the study region than in the linear model (fig06b), highlighting a
much better performance of the GWR, which minimized residual autocorrelation. Indeed, in the linear
model (fig06b), positive residuals were condensed in the right side of Tagus and Sado river basins, while
negative values were mainly present on the left side of the Tagus river and in the center-south of Alentejo.
The spatial distribution of the coefficients of GWR predictors is presented in Fig07. They were later used
for the computation of the GDV suitability score for each data point (Eq.6). The coefficient variability
was three times higher for the $A_i$ as compared to $O_4$ (fig08a), reaching 66% and 22% respectively. For W,
D and s, the coefficient variation was much lower, representing only about 6.2%, 3.8% and 1.2% of the
total variation observed in the coefficients, respectively. The remaining variables showed a median close
to 0 and the $O_4$ was the second with higher variability followed by the W. The coefficient median values
were, respectively, -3.40, 0.29, -0.015, -0.018 and 0.022 for $A_i$, $O_4$, W, D and s variables.
The distributions of negative coefficients were similar for $A_i$ and the $O_4$ variables (fig07a and fig07b),
with lower values in the southern coastal area, and in the Tagus river watershed. The highest absolute
values were mostly found for $A_i$ in the southern area of the Alentejo region and on smaller patches in the
northern region. In the center and eastern areas of Alentejo, a higher weight of the groundwater depth
coefficient could be found (fig07c), approximately matching a higher influence of slope (fig07e). The
groundwater depth seemed to have almost no influence on GDV density in the Tagus river watershed,
expressed by coefficients mostly null around the riverbed (fig07c). The coefficient distribution of D and
$O_4$ shows some similarities, mostly in the center and southeast of Alentejo (fig07d). Extreme values of $O_4$
coefficients were mostly concentrated in the eastern part of the Tagus watershed and in the southern
coastal area included in the Sado watershed. Slope coefficient values showed the lowest amplitude
throughout the study area (fig07e), with prevailing high positive values gathered mainly in the center of
the study area and in the Tagus river watershed (northwest of the study center).

**3.4 GDV Suitability map**
The classification of the 5 endorsed environmental predictors is presented in Table 4 and their respective
maps in figure B1 in appendix B. Rivers Tagus and Sado had an overall large impact on GDV's
suitability for each predictor, with the exception of W. This is due to a higher water availability reflected
by the values of $O_4$, D and lower slopes due to the alluvial plains of the Tagus river (figs. B1b, d and e in
appendix B). Moreover, those regions presented higher humidity conditions (through analysis of the $A_i$ in
fig B1a in appendix B) and groundwater depths outside the optimum range (Fig. B1c in appendix B),
therefore less suitable for GDV. Optimal conditions for groundwater access were mainly gathered in the
interior of the study region (fig. B1c in appendix B), with the exception of some confined aquifers in the
northeast and southeast of the study region. Favorable slopes for GDV were mostly highlighted in the
Tagus river basin area, where a good likelihood of interaction between GDV and groundwater could be
identified (fig. B1e in appendix B).
The final map illustrating the suitability to GDV is shown in Fig. 09. The largest classified area (8
787km$^2$) presented a very poor suitability to GDV, corresponding to approximately a quarter of the total
study area (29%). This percentage was followed closely by the moderate suitability to GDV which
occupied 26% (8000km$^2$). Overall, the two less suitable classes (very poor and poor) represented 47% of
the study area, whilst the two best ones and the moderate class (very good, good and moderate)
represented 53%. Consequently, most of the study area showed moderate to high suitability to GDV. The
very good and good suitability classes cover an arch from the most south and northeastern area of the
Alentejo region, passing through the Sado and southern Guadiana river basins and close to the coastal line
at 38ºN. Most of the center of the study area showed moderate to very good suitability to GDV, while the
areas corresponding to the alluvial deposits of the Tagus river showed poor to very poor suitability.
The suitability to GDV in the Alentejo region was mainly driven by $A_i$, given that the highest coefficient
variability was associated to the $A_i$ predictor in the GWR model equation. Consequently a similar
distribution pattern can be observed between the suitability map and the aridity index predictor (fig04a
and fig09). Areas with good or very good suitability mostly matched areas of $A_i$ with score 3,
corresponding to aridity index values above 0.75 (Fig. B1a in appendix B). On the other hand, the lowest
suitability classes showed a good agreement with the lowest scores given to W (fig. B1c in appendix B),
mostly in the coastal area and in the Tagus river basin.
**3.5 Map evaluation**
To evaluate the suitability map developed in the present study, the results were compared with the NDWI
anomaly considering the month of June of the dry year of 2005 in the Alentejo area (fig10). Both maps
(figs 09 and 10) showed similar patterns, with higher presence of GDV satisfactorily matching areas with
the lowest NDWI anomaly. From June to September in an extremely dry year, non-DGV plants can be
expected to experience a severe drought stress as in any regular summer period. Thus, those plants should
show almost zero anomaly. By opposition, GDV plants coping well with usual summer drought can be
expected to suffer an unusual stress under an extreme dry year even having access to groundwater (Kurz-
Besson et al. 2006 & 2014, Otieno et al. 2006, David et al. 2013), with a negative impact of groundwater
drawdown (Antunes et al., 2018). Therefore, GDV plants should show negative NDWI anomalies.
The NDWI anomaly was mostly negative over the Alentejo territory indicating a lower leaf water content
in June and July 2005 than usual. The loss of water attributed to the extreme drought was mostly
matching geographical areas with the highest GDV suitability (fig09). Water loss was less pronounced in
the central area of the Alentejo region between the Guadiana and Sado river basins, where the vegetation
is less dense (fig03). Areas with null NDWI anomaly values (indicating no NDWI change) were mostly
distributed on the coastal area of the Atlantic ocean or close to riverbeds, namely in the Tagus and Sado
floodplains, matching areas of very poor suitability for GDV in Figure 09.
Despite an overall good agreement, the adequation between the density, suitability and NDWI maps was
not perfect. Indeed, some patches showing a high vegetation occurrence/density and large NDWI
anomalies also matched an area of very poor suitability for GDV.
**3.6 Sensitivity analysis**
The sensitivity of the model in response to the perturbation of each one of the input variables ($A_i$, $O_4$, W,
D and s) is presented on Figures11a to 11e. The overall sensitivity of the model is further presented on
Figure 11f. For any input variable, the model sensitivity (fig11a to 11e) was higher where absolute values
of local coefficients were also higher (fig07a to 07e). The maximum impact on GDV's density,
corresponding to the maximum output range observed after perturbation (fig08b), was observed when
perturbing the $A_i$, accounting for 66% of the total variability. The second highest impact was observed
after perturbing the $O_4$, corresponding to 22%. The variability in the model outputs observed after
perturbing the remaining variables W, D and s accounted for 7%, 4% and 1% of the total accumulated
variability, respectively (fig08b). The highest variability in the GWR model output was mostly observed
in the central part of the southern half of the Alentejo region, as well as close to the main channels of the
Guadiana and Tagus rivers (fig11f). Furthermore, areas with higher model sensitivity (fig11f)
significantly matched higher model performance expressed by $R^2$ (fig05), assessed with a Kruskall-Wallis
test (p<0.0001***).

## 4 Discussion

### 4.1 Modeling approach

The Geographically Weighted Regression model has been used before in ecological studies (Li et al., 2016; Mazziotta et al., 2016), but never for the mapping of GDV, to our knowledge. This approach considerably improved the goodness of fit when compared to the linear model, with a coefficient of regression ($R^2$) increasing from 0.02 to 0.99 at the global level, and an obvious reduction of residual clustering. Despite those improvements, it has not been possible to completely eliminate the residual autocorrelation after fitting the GWR model.

Kernel density for the study area provided a strong indication of presence and abundance of the tree species considered as GDV proxy for modeling. The Mediterranean cork woodlands dominate about 76% of the Alentejo region (while only 7% is covered by stone pine). In those systems, tree density is known to be a tradeoff between climate drivers (Joffre 1999, Gouveia & Freitas 2008) and the need for space for pasture or cereal cultivation in the understory (Acácio & Holmgreen 2014). In our study, the anthropologic management of agroforestry systems in the Alentejo region has not been taken into account. According to a recent study of Cabon et al. (2018) where thinning played an important role in *Q. ilex* density in a Mediterranean climate site, anthropologic management could, at least partially, explain the non-randomness of the residual distribution after GWR model fitting as well as the mismatches between the GDV and the NDWI evaluation maps.

Another explanation of the reminiscent autocorrelation after GWR fitting could be the lack of groundwater dependent species in the model. For example, *Pinus pinaster* Aiton was excluded due to its more humid distribution in Portugal, and due to conflicting conclusions driven from previous studies to pinpoint the species as a potential groundwater user (Bourke, 2004; Kurz-Besson et al., 2016). In addition, olive trees were also excluded although the use of groundwater by an olive orchard has been recently proved (Ferreira et al., 2018), however with a weak contribution of groundwater to the daily root flow, and thus with no significant impact of groundwater on the species physiological conditions.

Methods previously used by Doody et al., (2017) and Condesso de Melo et al. (2015) to map specific vegetation relied solely on expert opinion, e.g. Delphi panel, to define weighting factors of environmental information for GIS multicriteria analysis. In our study, the GWR modelling approach was used to assess weighting factors for each environmental predictor in the study area, to build a suitability map for the GDV in southern Portugal. This allowed an empirical determination of the local relevance of each environmental predictor in GDV distribution, thus avoiding the inevitable subjectivity of Delphi panels.

Also, by combining the GWR and GIS approaches we believe the final suitability map provides a more reliable indication of the higher likelihood for groundwater dependency and a safer appraisal of the relative contribution of groundwater by facultative deep-rooted phreatophytes species in the Alentejo region.

Modelling of the entire study area at a regional level did not provide satisfactory results. Therefore, we
developed a general model varying locally according to local predictor coefficients. The local influence of
each predictor was highly variable throughout the study area, especially for climatic predictors reflecting
water availability and stress conditions. The application of the GWR model did not only allow for a
localized approach, by decreasing the residual error and autocorrelation over the entire studied region, but
also provided insights on how GDV's density can be explained by the main environmental drivers locally.
The GWR model appeared to be highly sensitive to coefficient fitting corresponding to a good model fit,
as expected in a spatially varying model. As so, high coefficients are highly reliable in the GWR model in
our study. Yet, the high spatial variability of local coefficients might reflect a weak physical meaning of
the GWR model that challenges its direct application in other regions, even under similar climate
conditions. Predictor coefficients showed a similar behavior in the spatial distribution of the coefficients.
This was noticeable for the aridity index and the groundwater depth in the Tagus and Sado river basins.
Groundwater depth had no influence on GDV's density in these areas and similarly, the coefficient of
aridity index showed a negative effect of increased humidity on GDV's density. In addition, a cluster of
low drainage density values matched these areas. Due to the lower variability and impact of the drainage
density and slope on the GDV's density, these variables might not impact significantly this vegetation
density in future climatic scenarios.

**4.2 Suitability to Groundwater Dependent Vegetation**
According to our results, more than half of the study area appeared suitable for GDV. However, one
quarter of the studied area showed lower suitability to GDV. The lower suitability to this vegetation in the
more northern and western part of the studied area included the coastal area and the Tagus river basin.
Those are the moist humid areas of the study area, where GDV is unlikely to rely on groundwater during
the drought season because rainfall water stored in shallow soil horizons is mostly available.
The proxy species (Cork oak, Holm oak and Stone pine) can perfectly grow under sub-humid
Mediterranean climate conditions, without relying as much on groundwater to survive as in more xeric
semi-arid areas (Abad Vinas et al., 2016). As facultative phreatophyte species, their presence/abundance
is only an indication of a possible use of groundwater. The study provided by Pinto et al. (2013) have
shown that Cork oak, for example, can perfectly thrive were very shallow groundwater is available while
suffering drought stress were groundwater source is lower but still extracted by trees. Also, former studies
have shown that in the extreme dry year of 2005, Cork oak experienced a severe drought stress, close to
the cavitation threshold, although its main water source was groundwater (David et al. 2013, Kurz-Besson
et al. 2006, 2014). These findings can explain that part of the maximum density (Fig. 04) matches the area
of very poor suitability for GDV (Fig. 09). Elsewhere, the better agreement between the two maps reflects
the dominance of the aridity index on the vegetation's occurrence. Groundwater depth appeared to have a
lower influence on GDV density than climate drivers, as reflected by the relative low magnitude of the W
coefficient and outputs of our model outcomes. This surprisingly disagrees with our initial hypothesis
because groundwater represents a notable proportion of the transpired water of deep-rooting
phreatophytes, reaching up to 86% of absorbed water during drought periods and representing about
30.5% of the annual water absorbed by trees (David et al. 2013, Kurz-Besson et al. 2014). Nonetheless,
this disagreement should be regarded cautiously due to the poor quality of piezometric data used and the
complexity required for modelling the water table depths. Besides, the linear relationship between water
depth and topography applied to areas of undifferentiated geological type can be weakened by a complex
non-linear interaction between topography, aridity and subsurface conductivity (Condon and Maxell,
2015). Moreover, the high variability in geological media, topography and vegetation cover at the
regional scale did not allow to account for small changes in groundwater depth (<15 m deep), which has a
huge impact on GDV suitability (Canadell et al., 1996; Stone and Kalisz, 1991). Indeed, a high spatial
resolution of hydrological database is essential to rigorously characterize the spatial dynamics of
groundwater depth between hydrographic basins (Lorenzo-Lacruz et al., 2017). Unfortunately, such
resolution was not available for our study area.
The aridity and ombrothermic indexes were the most important predictors of GDV density in the Alentejo
region, according to our model outcomes. Our results agree with previous findings linking tree cover
density and rooting depth to climate drivers such as aridity, at a global scale (Zomer et al., 2009; Schenk
and Jackson, 2002) and specifically for the Mediterranean oak woodland (Gouveia and Freitas 2008,
Joffre et al. 1999). Through previous studies showing the similarities in vegetation strategies to cope with
water scarcity in the Mediterranean basin (Vicente-Serrano et al., 2013) or the relationship between
rooting depth and water table depth increased with aridity at a global scale (Fan et al., 2017) we can admit
that the most relevant climate drivers pinpointed here are similarly important to map GDV in other semi-
arid regions. In this study, the most important environmental variables that define GDV's density in a
semi-arid region were identified, helping to fill the gap of knowledge for modelling this type of
vegetation. However, the coefficients to be applied when modelling each variable need to be calculated
locally, due to their high spatial variability.
Temporal piezometric data would further help discriminate areas of optimal suitability to GDV, either
during the wet and the dry seasons, because the seasonal trends in groundwater depth are essential under
Mediterranean conditions. Investigations efforts should be invested to fill the gap either by improving the
Portuguese piezometric monitoring network, or by assimilating observations with remote sensing
products focused on soil moisture or groundwater monitoring. This has already been performed for large
regional scale such as GRACE satellite surveys, based on changes of Earth's gravitational field. So far,
these technologies are not applicable to Portugal's scale, since the coarse spatial resolution of GRACE
data only allows the monitoring of large reservoirs (Xiao et al. 2015).

**4.3 Validation of the results**
The understory of woodlands and the herbaceous layer of grasslands areas in southern Portugal usually
ends their annual life cycles in June (Paço et al. 2009), while the canopy of woody species is still fully
active with maximum transpiration rates and photosynthetic activities (Kurz-Besson et al. 2014, David et
al. 2007, Awada et al. 2003). This is an ideal period of the year to spot differential response of the canopy
of woody species to extreme droughts events using satellite derived vegetation indexes (Gouveia 2012).
The spatial patterns of NDWI anomaly in June 2005 seem to indicate that the woody canopy showed a
strong loss of canopy water in the areas were tree density and GDV suitability were higher (figs03, 09 and
10). This occurred although trees minimized the loss of water in leaves with a strong stomatal limitation
in response to drought (Kurz-Besson et al. 2014, Grant et al. 2010). In the most arid area of the region
were Holm oak is dominant but tree density is much lower, the NDWI anomaly was generally less
negative thus showing a lower water stress or higher canopy water content.  Holm oak (*Quercus ilex* spp
*rotundifolia*) is well known to be the most resilient species to dry and hot conditions in Portugal, due to
its capacity to use groundwater, associated to a higher water use efficiency (David et al. 2007).
Furthermore, the dynamics of NDWI anomaly over the summer period (fig10a, b and c) pointed out that
the lower water stress status on the map is progressively spreading from the most arid areas to the milder
ones from June to August 2005, despite the intensification of drought conditions. This endorses the idea
that trees manage to cope with drought by relying on deeper water sources in response to drought,
replenishing leaf water content despite the progression and intensification of drought conditions. Former
studies support this statement by showing that groundwater uptake and hydraulic lift were progressively
taking place after the onset of drought by promoting the formation of new roots reaching deeper soil
layers and water sources, typically from July onwards, for cork oak in the Alentejo region (Kurz-Besson
et al., 2006, 2014). Root elongation following a declining water table has also been reported in a review
on the effect of groundwater fluctuations on phreatophyte vegetation (Naumburg et al. 2005).
Our results and the dynamics of NDWI over summer 2005 tend to corroborate the studies of Schenk and
Jackson (2002) and Fan et al. (2017), by suggesting a larger/longer dependency of GDV on groundwater
with higher aridity. Further investigation needs to be carried on across aridity gradients in Portugal and
the Iberian Peninsula to fully validate this statement, though.
Overall, the map of suitability to GDV showed a good agreement with the NDWI validation maps. The
main areas showing good GDV suitability and highest NDWI anomalies are mostly matching in both
maps. The good agreement between our GDV suitability maps, and NDWI dynamic maps opens the
possibility to apply and extend the methodology to larger geographical areas such as the Iberian Peninsula
and to the simulation of the impact of climate changes on the distribution of groundwater dependent
species in the Mediterranean basin.
Simulations of future climate conditions based on RCP4.5 and RCP8.5 emission scenarios (Soares et al.,
2015, 2017) predict a significant decrease of precipitation for the Guadiana basin and overall decrease for
the southern region of Portugal within 2100. Agroforestry systems relying on groundwater resources,
such as cork oak woodlands, may show a decrease in productivity and ecosystem services or even face
sustainability failure. Many studies carried out on oak woodlands in Italy and Spain identified drought as
the main driving factor of tree die-back and as the main climate warning threatening oak stands
sustainability in the Mediterranean basin (Gentilesca et al. 2017). An increase in aridity and drought
frequency for the Mediterranean (Spinoni et al., 2017) will most probably induce a geographical shift of
GDV vegetation toward milder/wetter climates (Lloret et al., 2004; Gonzalez P., 2001).

**4.4 Key limitations**

The GWR modelling approach used to estimate weighting factors is mostly stochastic. Consequently, the large spatial variability and symmetrical fluctuations around zero (Fig 08b) denote a weak physical meaning of the estimated coefficients, at least at the resolution chosen for the study. Also, the local nature of the regression coefficients makes the model difficult to directly apply in other regions, even with similar climate conditions, unless the methodology is properly fitted to local conditions/predictors.

With the methodology applied in this study, weighting factors can be easily evaluated solely from local and regional observations of the studied area. Nonetheless, the computation of model coefficients or expert opinion to assess weighting factors, require recurrent amendments, associated with updated environmental data, species distribution and revised expert knowledge (Doody et al., 2017).

The evolution of groundwater depth in response to climate change is difficult to model on a large scale based on piezometric observations because it requires an excellent knowledge of the components and dynamics of water catchments. Therefore, a reliable estimation of the impact of climate change on GDV suitability in southern Portugal could only been performed on small scale studies. However, the GWR model appeared to be much more sensitive to climate drivers than the other predictors, given that 88% of the model outputs variability was covered by climate indexes $A_i$ and $O_4$. Nevertheless, changes in climate conditions only represent part of the water resources shortage issue in the future. Global-scale changes in human populations and economic progresses also rule water demand and supply, especially in arid and semi-arid regions (Vörösmarty et al., 2000). A decrease in useful water resources for human supply can induce an even higher pressure on groundwater resources (Döll, 2009), aggravating the water table drawdown caused by climate change (Ertürk et al., 2014). Therefore, additional updates of the model should include human consumption of groundwater resources, identifying areas of higher population density or intensive farming. Future model updates should also account for the interaction of deep rooting species with the surrounding understory species. In particular, shrubs surviving the drought period, which can benefit from the redistribution of groundwater by deep rooted species (Dawson, 1993; Zou et al., 2005).

**5 Conclusions**

Our results show a highly dominant contribution of water scarcity of the last 30 years (Aridity and Ombrothermic indexes) on the density and suitability of deep-rooted groundwater dependent species in southern Portugal. Therefore, in geographical regions of the world with similar semi-arid climate conditions (Csa according to Köppen-Geigen classification, Peel et al. 2007) and similar physiological responses of the groundwater dependent vegetation (Vicente-Serrano et al., 2013), the use of the aridity and ombrothermic indexes could be used as first approximation to model and map deep rooted phreatophyte species and the evolution of their distribution in response to climate changes. The contribution of groundwater depth was lower than initially expected; however, this might be underestimated due to the poor quality of the piezometric network, especially in the central area of the studied region.

The current pressure applied by human consumption of water sources has reinforced the concern on the future of economic activities dependent on groundwater resources. To address this issue, several countries have developed national strategies for the adaptation of water sources for Agriculture and Forests against Climate Change, including Portugal (FAO, 2007). In addition, local drought management as long-term adaptation strategy has been one of the proposals by Iglesias et al. (2007) to reduce the climate change impact on groundwater resources in the Mediterranean. The preservation of Mediterranean agroforestry systems, such as cork oak woodlands and the recently associated *P. pinea* species, is of great importance due to their high socioeconomic value and their supply of valuable ecosystem services (Bugalho et al., 2011). Management policies on the long-term should account for groundwater resources monitoring, accompanied by defensive measures to ensure agroforestry systems sustainability and economical income from these Mediterranean ecosystems are not greatly and irreversibly threatened.

Our present study, and novel methodology, provides an important tool to help delineating priority areas of action for species and groundwater management, at regional level, to avoid the decline of productivity and cover density of the agroforestry systems of southern Portugal. This is important to guarantee the sustainability of the economical income for stakeholders linked to the agroforestry sector in that area. Furthermore, mapping vulnerable areas at a small scale (e.g.by hydrological basin), where reliable groundwater depth information is available, should provide further insights for stakeholder to promote local actions to mitigate climate change impact on GDV.

Based on the methodology applied in this work, future predictions on GDV suitability, according to the RCP4.5 and RCP8.5 emission scenarios will be shortly introduced, providing guidelines for future management of these ecosystems in the allocation of water resources.

**6 Acknowledgements**

The authors acknowledge the E-OBS dataset from the EU-FP6 project ENSEMBLES (http://ensembles-eu.metoffice.com) and the data providers in the ECA&D project (http://www.ecad.eu). The authors also wish to acknowledge the ASTER GDEM data product, a courtesy of the NASA Land Processes Distributed Active Archive Center (LP DAAC), USGS/Earth Resources Observation and Science (EROS) Center, Sioux Falls, South Dakota, https://lpdaac.usgs.gov/data_access/data_pool. We are grateful to ICNF for sharing inventory database performed in 2010 in Portugal continental. We also thank Cristina Catita, Ana Russo and Patrícia Páscoa for the advice and helpful comments as well as Ana Bastos for the elaboration of the satellite datasets of the vegetation index NDWI and Miguel Nogueira for the insights on model sensitivity analysis. We are very grateful to Eric Font for the useful insights on soil properties. I Gomes Marques and research activities were supported by the Portuguese National Foundation for Science and Tecnhology (FCT) through the PIEZAGRO project (PTDC/AAG-REC/7046/2014). This publication was also supported by FCT- project UID/GEO/50019/2019 – Instituto Dom Luiz. The authors further thank the reviewers and editor for helpful comments and suggestions on an earlier version of the manuscript.

The authors declare that they have no conflict of interest.

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

**Figure and Table Legends**

Table 1: Environmental variables for the characterization of the suitability of GDV in the study area.

Table 2: Effect of variable removal in the performance of GWR model linking the Kernel density of *Quercus suber,* *Quercus ilex* and *Pinus pinea* ($S_{GDV}$) to predictors Aridity Index ($A_i$); Ombrothermic Index of the summer quarter and the immediately previous month ($O_4$); Slope (s); Drainage density (D); Groundwater Depth (W) and Soil type ($S_t$). The model with all predictors is highlighted in grey and the final model used in this study is in bold.

Table 3: Comparison of Adjusted $R^2$ and second-order Akaike Information Criterion (AICc) between the simple regression and the GWR models.

Table 4: Classification scores for each predictor. A score of 3 was given to highly suitable areas and 1 to less suitable areas for GDV.

Table A1: Classification scores for soil type predictor.

Table A2: Correlations between predictor variables and principal component axis. The most important predictors for each axis (when squared correlation is above 0.3) are showed in bold. The cumulative proportion of variance explained by each principal component axis is shown at the bottom of the table. s is slope; Ai is Aridity Index; O, O1, O3, O4 are ombrothemic indices of, respectively, the year, the hottest month of the summer quarter, the summer quarter and the summer quarter and the immediately previous month; SPEIs and SPEIe are, respectively, the number of months with severe and extreme Standardized Precipitation Evapotranspiration Index; W is Groundwater Depth; D is the Drainage density; St refers to soil type and T is thickness.

Figure 01: Study area. On the left the location of Alentejo in the Iberian Peninsula; on the right, the elevation characterization of the study area with the main river courses from Tagus, Sado and Guadiana basins (white lines). Names of the main rivers are indicated near to their location in the map.

Figure 02: Large well and piezometer data points used for groundwater depth calculation. Squares represent piezometers data points and triangle represent large well data points.

Figure 03: Map of Kernel Density weighted by cover percentage of *Q. suber, Q. ilex* and *P. pinea*. The scale unit represent the number of occurrences per 10 km search radius (~314 km$^2$).

Figure 04: Map of environmental layers used in model fitting. (a) – Aridity Index; (b) – Ombrothermic Index of the summer quarter and the immediately previous month; (c) – Groundwater Depth; (d) –Drainage density; (e) –Slope.

Figure 05: Spatial distribution of local $R^2$ from the fitting of the Geographically Weighted Regression.

Figure 06: Spatial distribution of model residuals from the fitting of the Geographically Weighted Regression (a) and the Simple Linear model (b).

Figure 07: Map of local model coefficients for each variable. (a) – Aridity Index; (b) - Ombrothermic Index of the summer quarter and the immediately previous month; (c) – Groundwater Depth; (d) – Drainage density and (e) - Slope.

Figure 08: Boxplot of GWR model coefficient values for each predictor (a) and boxplot of the GWR model outputs, corresponding to GDV's density after each of the predictors was disturbed for the sensitivity analysis (b). $A_i$ stands for Aridity Index; $O_4$ for the ombrothemic index of the hottest month of the summer quarter and the immediately previous

month; W for the groundwater depth; D for the drainage density and s for the slope. Error bars represent the 25th and 75th percentile while crosses indicate the 95th percentile.

Figure 09: Suitability map for Groundwater Dependent Vegetation.

Figure 10: Spatial patterns of NDWI anomaly values considering the months of June, July and August of the extremely dry year of 2005, in reference to the same months of the period 1999-2009, in the Alentejo region. Dark brown colors (corresponding to extreme negative NDWI anomaly values) indicate the vegetation that experienced the highest loss of water in leaves in summer 2005 as compared to the reference period 1999-2009, while light brown colors show NDWI anomaly values very close to the usual vegetation moisture condition of the considered month.

Figure 11: Sensitivity analysis performed on the GWR model by perturbing one of the predictors, while remaining the rest of the model equation constant. Graphics present the output range of GDV's density when the aridity index (a), the ombrothermic index (b), the groundwater depth (c), the drainage density (d) or the slope variable (e) was perturbed; and the maximum possible range combining all predictors (f). The 95th percentile was used for the maximum value of the color bar for a better statistical representation of the spatial variability.

Figure A1: Boxplot of the main predictors used for the Geographically Weighted Regression model fitting (top) and the response variable (below), for the total data (left) and for the 5% subsample (right).

Figure A2: Correlation plot between all environmental variables expected to affect the presence of the Groundwater Dependent Vegetation. $O_1$, $O_3$ and $O_4$ are ombrothermic indices of, respectively, the hottest month of the summer quarter, the summer quarter and the summer quarter and the immediately previous month; O is the annual ombrothermic index, $SPEI_e$ and $SPEI_s$ are, respectively, the number of months with extreme and severe Standardized Precipitation Evapotranspiration Index; $A_i$ is Aridity index; W is groundwater depth; D is the Drainage density; T is thickness and $S_t$ refers to soil type.

Figure B1 – Predictors maps after score classification. (a) – Aridity Index; (b) – Ombrothermic Index of the summer quarter and the immediately previous month; (c) – Groundwater Depth; (d) – Drainage density and (e) – Slope.

 **Table 1: Environmental variables for the characterization of the suitability of GDV in the study area.**

| Variable code | Variable type | Source | Resolution and Spatial extent |
|---|---|---|---|
| s | Slope (%) | This work | 0.000256 degrees (25m) raster resolution |
| $S_t$ | Soil type in the first soil layer | SNIAmb (© Agência Portuguesa do Ambiente, I.P., 2017) | Converted from vectorial to 0.000256 degrees (25m) resolution raster |
| T | Soil thickness (cm) | EPIC WebGIS Portugal (Barata et al., 2015) | Converted from vectorial to 0.000256 degrees (25m) resolution raster |
| W | Groundwater Depth (m) | This work | 0.000256 degrees (25m) raster resolution |
| D | Drainage Density | This work | 0.000256 degrees (25m) raster resolution |
| $SPEI_s$ | Number of months with severe SPEI | This work | 0.000256 degrees (25m) raster resolution Time coverage 1950-2010 |
| $SPEI_e$ | Number of months with extreme SPEI | This work | 0.000256 degrees (25m) raster resolution Time coverage 1950-2010 |
| $A_i$ | Aridity Index | This work | 0.000256 degrees (25m) raster resolution Time coverage 1950-2010 |
| O | Annual Ombrothermic Index Annual average (January to December) | This work | 0.000256 degrees (25m) raster resolution Time coverage 1950-2010 |
| $O_1$ | Ombrothermic Index of the hottest month of the summer quarter (J, J and A) | This work | 0.000256 degrees (25m) raster resolution Time coverage 1950-2010 |
| $O_3$ | Ombrothermic Index of the summer quarter (J, J and A) | This work | 0.000256 degrees (25m) raster resolution Time coverage 1950-2010 |
| $O_4$ | Ombrothermic Index of the summer quarter and the immediately previous month (M, J, J and A) | This work | 0.000256 degrees (25m) raster resolution Time coverage 1950-2010 |

**Table 2: Effect of variable removal in the performance of GWR model linking the Kernel density of *Quercus***
***suber, Quercus ilex* and *Pinus pinea* (S$_{GDV}$) to predictors Aridity Index (A$_i$); Ombrothermic Index of the**
**summer quarter and the immediately previous month (O$_4$); Slope (s); Drainage density (D); Groundwater**
**Depth (W); and Soil type (S$_t$). The model with all predictors is highlighted in grey and the final model used in**
**this study is in bold.**

| Type | Model | Discarded predictor | AICc | Quasi-global R² |
|------|-------|---------------------|------|-----------------|
| GWR | S$_{GDV}$~ O$_4$ + A$_i$ + s + D + W + S$_t$ | | 27389.74 | 0.926481 |
| GWR | S$_{GDV}$ ~ O$_4$ + s + D + W + S$_t$ | A$_i$ | 28695.14 | 0.9085754 |
| GWR | S$_{GDV}$ ~ A$_i$ + s + D + W + S$_t$ | O$_4$ | 28626.88 | 0.9095033 |
| GWR | S$_{GDV}$ ~ O$_4$ + A$_i$ + s+ W + S$_t$ | D | 27909.86 | 0.9184337 |
| GWR | S$_{GDV}$ ~ O$_4$ + A$_i$ + D + W + S$_t$ | s | 27429.55 | 0.924176 |
| GWR | S$_{GDV}$ ~ O$_4$ + A$_i$ + s + D + S$_t$ | W | 27742.67 | 0.9208344 |
| GWR | **S$_{GDV}$ ~ O$_4$ + A$_i$ + s + D + W** | **S$_t$** | **18050.76** | **0.9916192** |


**Table 3: Comparison of Adjusted R$^2$ and second-order Akaike Information Criterion (AICc) between the simple**
**linear regression and the GWR model.**

| Model | R$^2$ | AICc | p-value |
|-------|-------|------|---------|
| OLS | 0.02 | 42720 | <0.001 |
| GWR | 0.99 * | 18851 | - |

*Quasi-global R$^2$

**Table 4: Classification scores for each predictor. A score of 3 was given to highly suitable areas and 1 to less**
**suitable areas for GDV.**

| Predictor | Class | Score |
|-----------|-------|-------|
| Slope | 0%-5% | 3 |
| | 5%-10% | 2 |
| | >10% | 1 |
| Groundwater Depth | >15 m | 1 |
| | 1.5m-15m | 3 |
| | ≤1.5m | 1 |
| Aridity Index | 0.6-0.68 | 3 |
| | 0.68-0.75 | 2 |
| | ≥0.75 | 1 |
| Ombrothermic Index of the summer quarter and the immediately previous month | <0.28 | 1 |
| | 0.28-0.64 | 2 |
| | ≥0.64 | 3 |
| Drainage Density | ≤0.5 | 3 |
| | >0.5 | 1 |



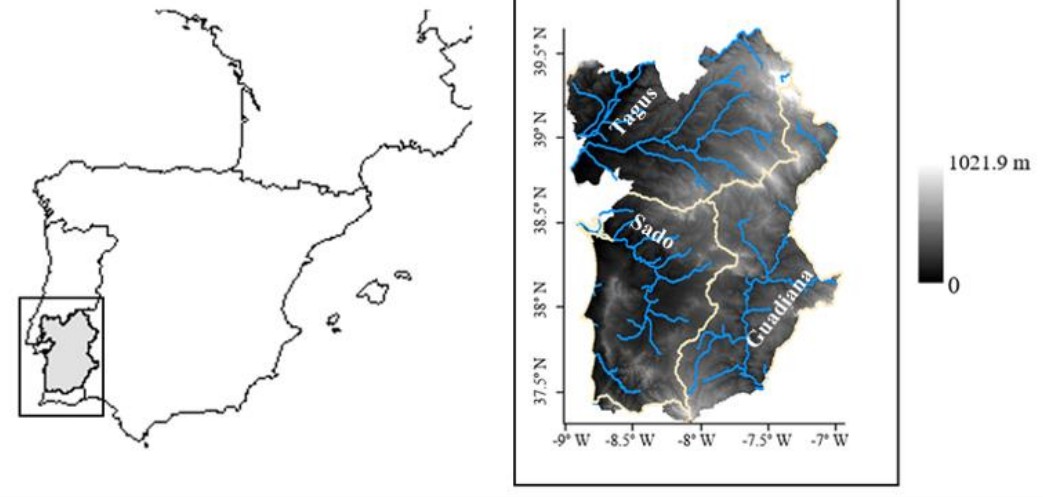


**Figure 01: Study area. On the left the location of Alentejo in the Iberian Peninsula; on the right, the elevation**
**characterization of the study area with the main river courses from Tagus, Sado and Guadiana basins (white**
**lines). Names of the main rivers are indicated near to their location in the map.**


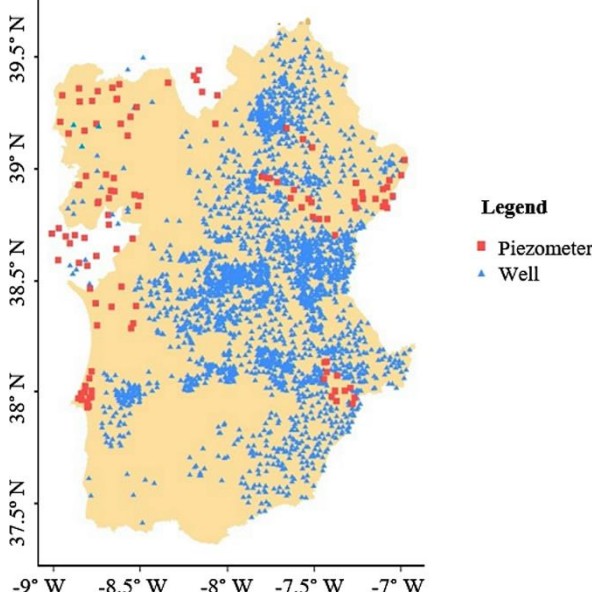


**Figure 02: Large well and piezometer data points used for groundwater depth calculation. Squares represent**
**piezometers data points and triangle represent large well data points.**

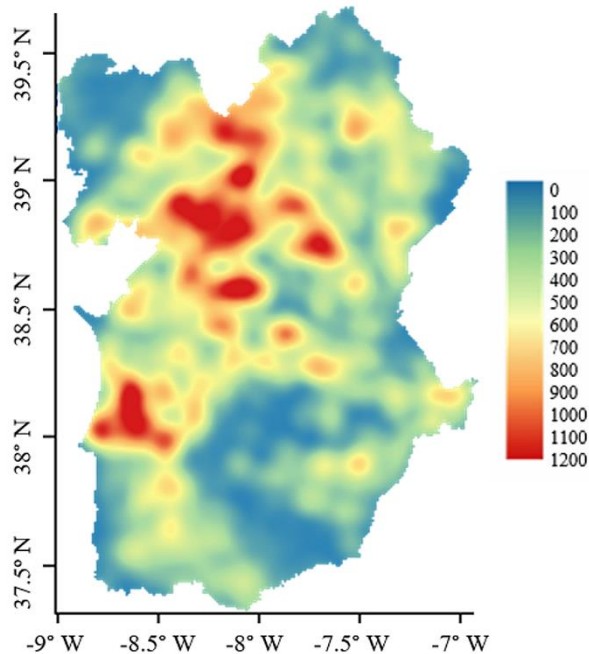


**Figure 03: Map of Kernel Density weighted by cover percentage of** *Q. suber, Q. ilex* **and** *P. pinea*. **The scale unit**
**represent the number of occurrences per 10 km search radius (~314 km$^2$).**


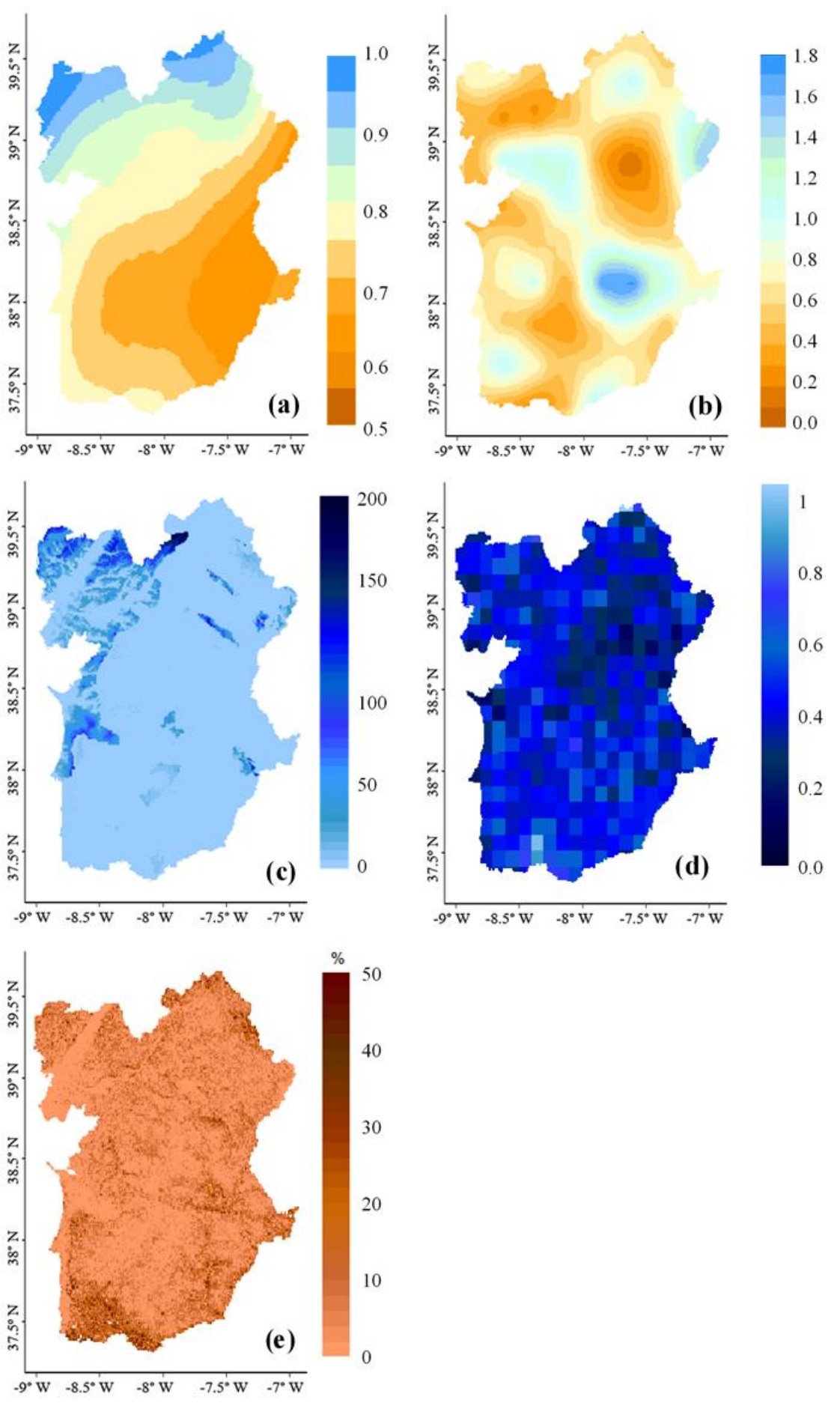

**Figure 04: Map of environmental layers used in model fitting. (a) – Aridity Index; (b) – Ombrothermic Index of**
**the summer quarter and the immediately previous month; (c) – Groundwater Depth; (d) –Drainage density; (e)**
**–Slope.**

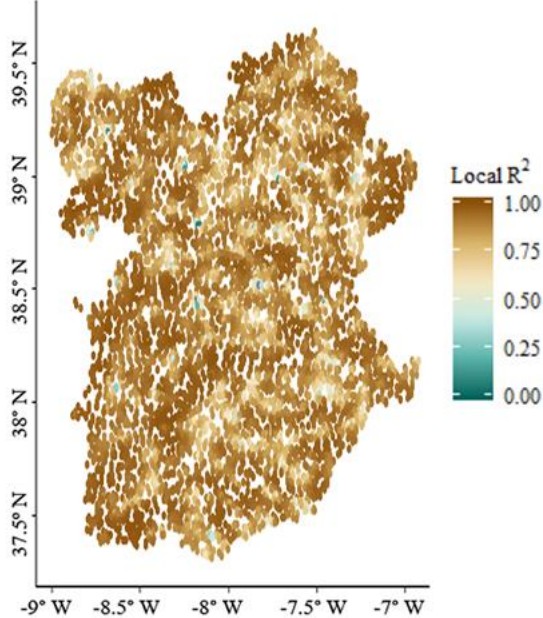


**Figure 05: Spatial distribution of local R$^2$ from the fitting of the Geographically Weighted Regression.**

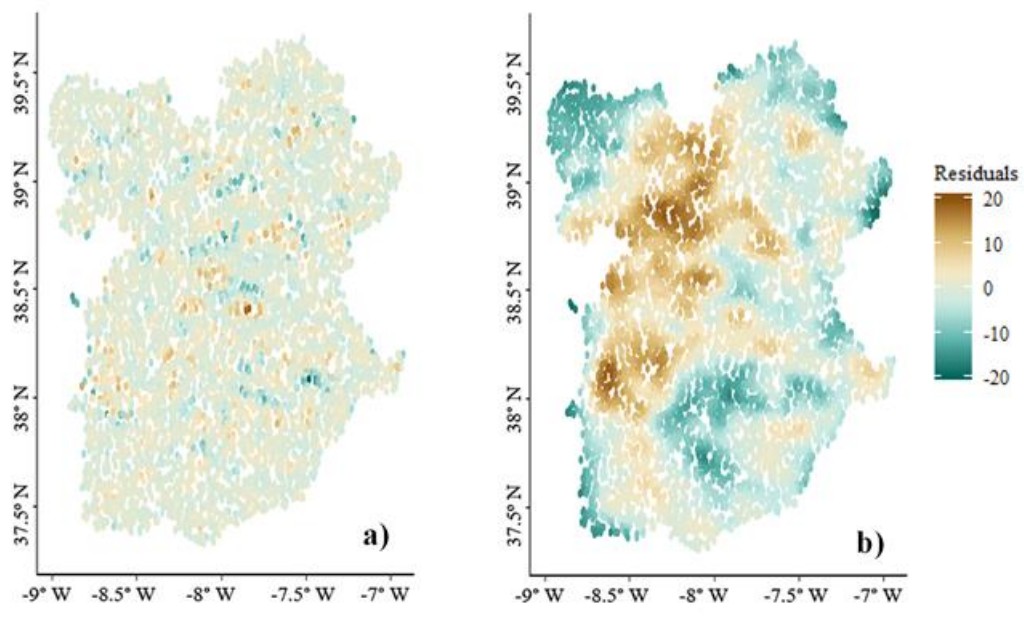


**Figure 06: Spatial distribution of model residuals from the fitting of the Geographically Weighted Regression**
**(a) and Simple Linear model (b).**



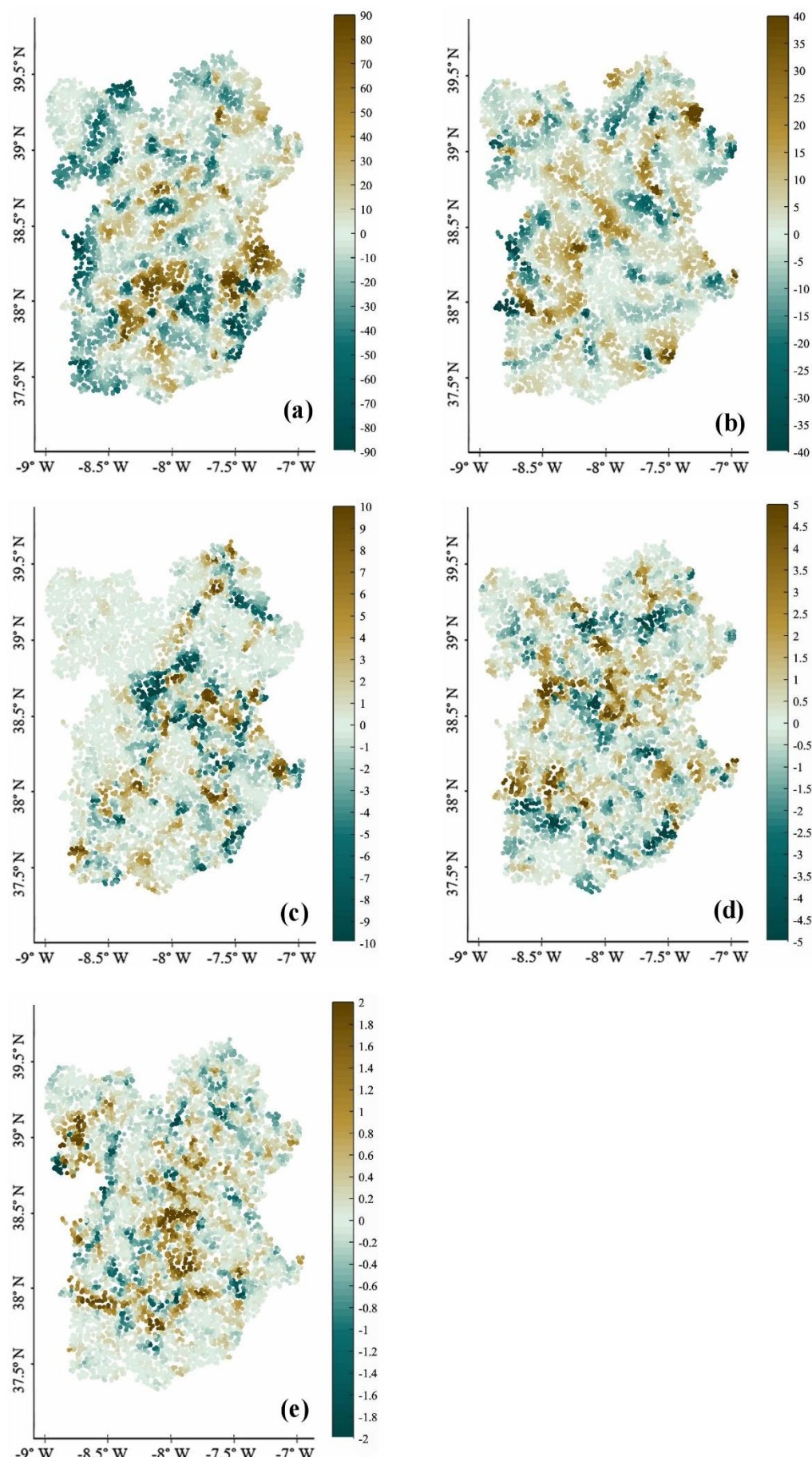


**Figure 07: Map of local model coefficients for each variable. (a) – Aridity Index; (b) - Ombrothermic Index of**
**the summer quarter and the immediately previous month; (c) – Groundwater Depth; (d) – Drainage density and**
**(e) – Slope.**

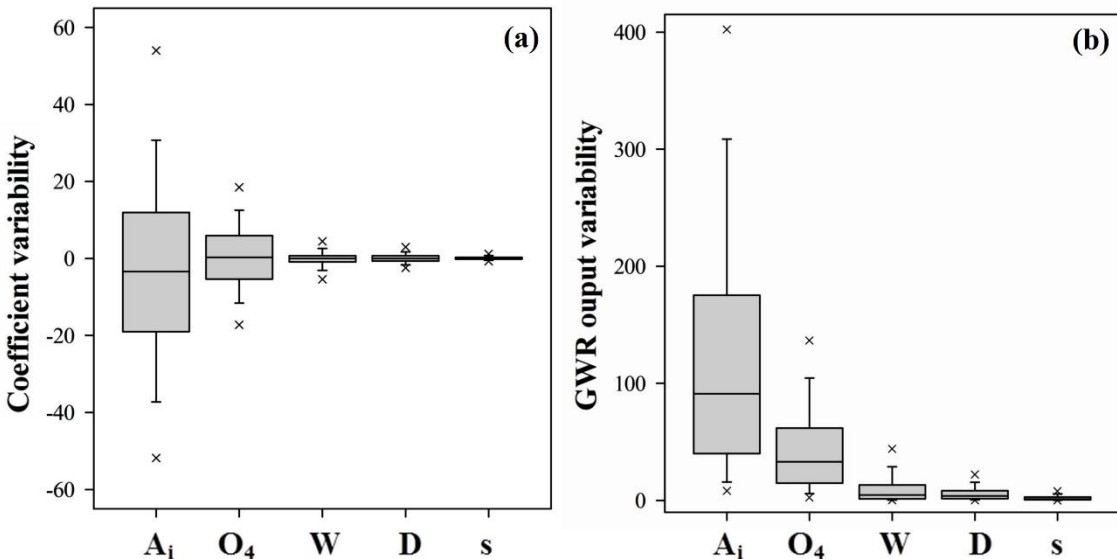


**Figure 08: Boxplot of GWR model coefficient values for each predictor (a) and boxplot of the GWR model**
**outputs, corresponding to GDV's density after each of the predictors was disturbed for the sensitivity analysis**
**(b). $A_i$ stands for Aridity Index; $O_4$ for the ombrothemic index of the hottest month of the summer quarter and**
**the immediately previous month; W for the groundwater depth, D for the drainage density and s for the slope.**
**Error bars represent the 25th and 75th percentile while crosses indicate the 95th percentile.**


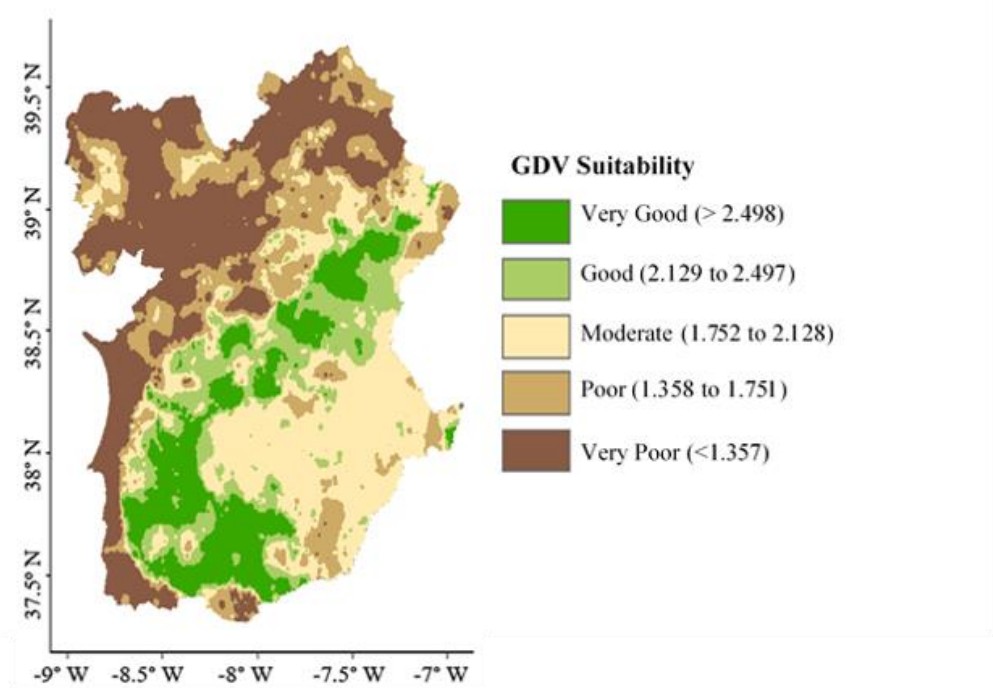


**Figure 09: Suitability map for Groundwater Dependent Vegetation.**

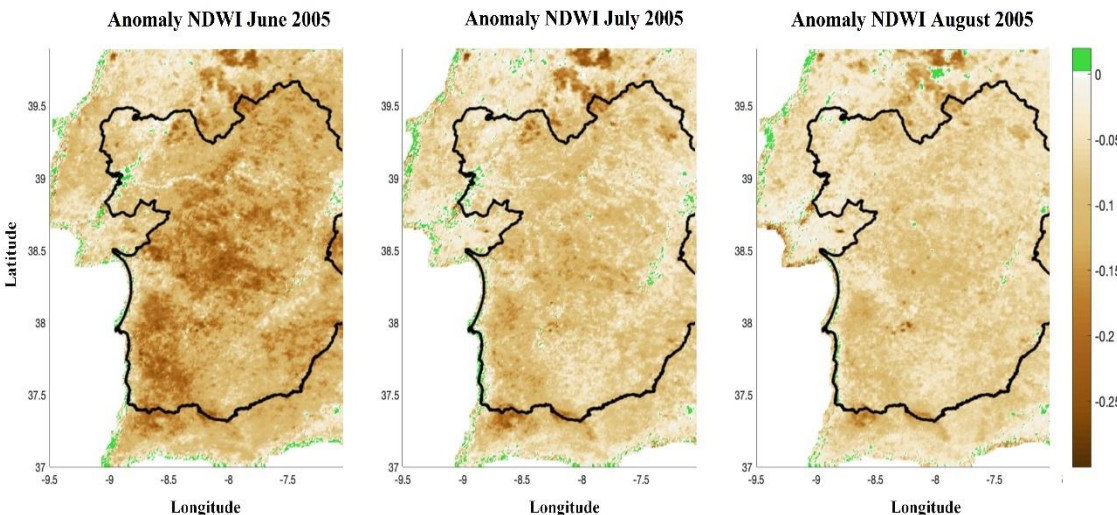

**Figure 10: Spatial patterns of NDWI anomaly values considering the months of June, July and August of the extremely dry year of 2005, in reference to the same months of the period 1999-2009, in the Alentejo region. Dark brown colors (corresponding to extreme negative NDWI anomaly values) indicate the vegetation that experienced the highest loss of water in leaves in summer 2005 as compared to the reference period 1999-2009, while light brown colors show NDWI anomaly values very close to the usual vegetation moisture condition of the considered month.**

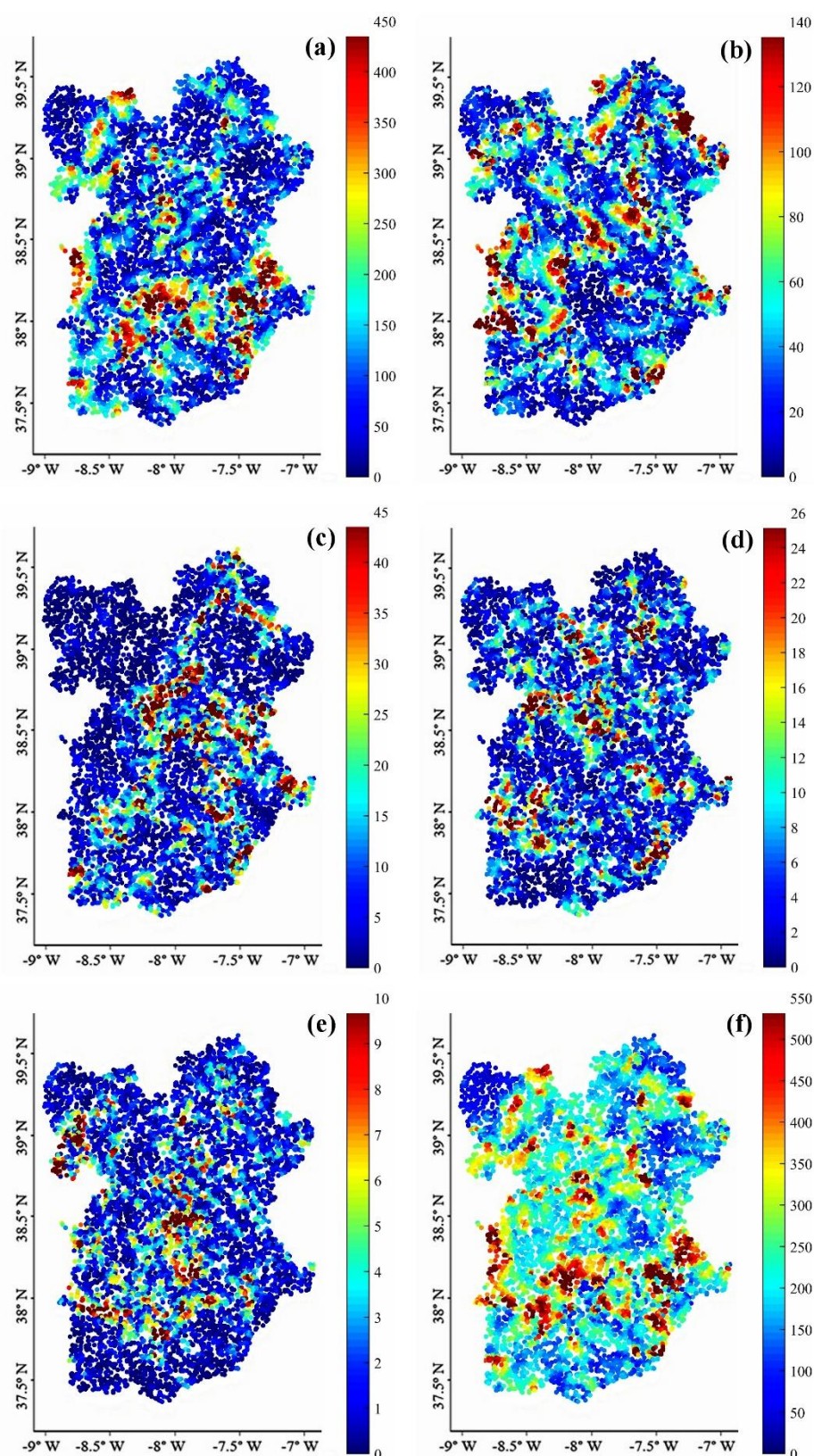

1311

Figure 11: Sensitivity analysis performed on the GWR model by perturbing one of the predictors, while remaining the rest of the model equation constant. Graphics present the output range of GDV's density when the aridity index (a), the ombrothermic index (b), the groundwater depth (c), the drainage density (d) or the slope variable (e) was perturbed; and the maximum possible range combining all predictors (f). The 95th percentile was used for the maximum value of the color bar for a better statistical representation of the spatial variability.