# Peer review of "Mapping the suitability of groundwater dependent vegetation in a semi-arid Mediterranean area"

_Hydrology and Earth System Sciences, 2018_

## Referee Comment (RC1) · Anonymous Referee #1 · 4 Jul 2018

General Comments

Gomes Marques et al. present an analysis of the spatial distribution of groundwater dependent vegetation across the Iberian Peninsula. While the method used is perhaps not as novel as suggested in the text, the paper's main strength lies in the validation of the maps created against a fairly robust eternal dataset. The text is generally well written, although it is not as clear as it could be when discussing how the "model" was parameterized and validated. In general, the paper is a solid contribution to the literature on phreatophytes, but needs revision to enhance its clarity and address some lingering questions about the work.

Specific Comments

1. Throughout: What exactly is meant by a "suitability map"? Suitability for what? Or do you mean suitability of the terrain for hosting phreatophtyes? The concept is fine, but the word choice seems odd.

2. Line 147 - 149: How heavily managed are these forestry systems? What species are harvested? And with what methods?

3. Line 170: Cite ASTER GDEM data in the manner requested on the NASA webpage (https://lpdaac.usgs.gov/citing_our_data).

4. Line 172: What is meant by superficial water use? Shallow groundwater? Surface water in streams? It's used several places but isn't well defined.

5. Line 179: What were the three classes? How did soil parameters influence in classification?

6. Line 187-202, Figure 2: The location of piezometer data and well data are quite biased. What is this attributable to and how might it affect the results? It seems like the kriging in the south-central region could be quite problematic. Also, what is the distinction between a well and a piezometer here? This is also concerning because the most dense of the GDV species are roughly in this corridor as well.

7. Line 199 -202: I disagree with this method - the groundwater elevations should be determined by first determining the groundwater elevation at the piezometers and then interpolating that through kriging. This should introduce fewer errors and be more realistic.

8. Line 303 - 312: The rationale for this validation method is a bit shaky and could use more explanation. If the presence/absence of these trees is a good indicator, why is the rest of the analysis necessary? Is it more expansive? Precise? Also, how is this not a bit autoregressive, given that it sounds like kernal density was derived from the tree data. It starts to make more sense as the results are discussed, but it needs more clarity here. What about using a remote sensing method for validation instead (e.g.,

Munch et al. 2007, Barron et al. 2012, Gou et. al 2014)? How would that compare?

9. Line 389-397: What soil types were the most likely to host phreatophtyes? What does "soil type 3" represent?

10. Line 480-484: This paragraph seems to be saying that there must be some threshold by which no woody species can be supported, even if they are GDVs. These species get replaced by shortlived grasses and forbes, converting savanna to grassland. Is this correct? If so, this seems to contradict the next line about woody vegetation being replaced by shrublands. Wouldn't that presume shrubs are less susceptible to drought than trees? Please clarify.

11. Line 495-511: This part of the discussion is problematic, because, as the authors note, the factor expected to be most key is poorly mapped. Regardless, they still say that soil type, as opposed to groundwater depth, is the most influential and claim that soil type defines the capacity for "groundwater storage". This appears to be overreach.

12. Figure 7: This figure needs more color variation. It is difficult to tell moderate, good, and very good apart.

Technical Corrections

13. Line 88: Replace "genders" with genre

14. Line 102: Replace "5m" with "5 m". Noticed number/unit spacing issues in several other locations as well.

15. Lines 132 - 135: Replace "chapters" with "sections". But really, this whole paragraph isn't necessary, as the format doesn't deviate from standard expectations.

16. Line 154: "Proxy for" not "proxy to".

17. Line 175: Is the copyright symbol here a typo?

18. Line 201: Don't need to repeatedly cite Spatial Analyst and its version so frequently.

Can this be converted to one mention at the beginning of the section?

19. Line 295: Put equation right after first mention.

20. Line 306: Replace "to a" with "of a".

21. Line 434, Line 454: Delete first names of authors.

22. Line 450: Pinpoint is one word.

23. Lines 451-453: Awkward wording makes the sentence hard to parse.

24. Line 466: Delete stray "s".

---

## Referee Comment (RC2) · Anonymous Referee #2 · 4 Sep 2018

**General comments**

The current manuscript provides an interesting insight into the use of mapping and spatial regression to assess the occurrence of groundwater dependent vegetation (GDV). Such maps can subsequently be used to predict the effect of change in any of the explanatory variables, such as climate or groundwater depth, on the spatial distribution of GDV in an area. The paper is well written and structured, and is subdivided into two parts: the first on the building of regression models for predicting GDV occurrence based on actual data, and the second part where a parameter-based index is calculated to construct so-called "suitability maps" for GDV. While I find the first part strong and with high potential of publication by adding a scenario analysis, my main concern lies with the second part. In my opinion this part is less well developed and

the interpretation of the results is largely straightforward, and to a certain degree incorrect. Interpretation is largely straightforward because of the large bias in weighting of the parameters, where the contribution of soil largely exceeds all other parameters, thus making it essentially a soil map. Additionally, interpretation is to a certain degree incorrect, due to an apparent mistake in the generation of the suitability map, where (inadvertently) a negative weighting was assigned to the aridity index, resulting in the inverse impact of this parameter on the soil map. I would therefore recommend focusing and further elaborating on the regression modelling, as I will specify in my detailed comments below.

Specific comments

Abstract

Line 13-19: The first part of the abstract is more of an introduction. I suggest starting with what was actually done (line 20). Moreover, groundwater depletion will not occur merely as a result of climate change. Finally, as groundwater level seems to have such a low impact in the regression model, the question rises to what extent groundwater depletion will play a role in the spatial distribution of GDV.

Line 48: When referring to climate change impact studies on recharge in Mediterranean areas add the paper of doi.org/10.1007/s10113-012-0377-3, where such a study is being reported.

PART 1: REGRESSION MODELLING

Parameter selection for the regression model

Soil type: the authors only use the first layer of the soil. To understand the importance of capillary rise feeding into the root zone, the texture of deeper soils also needs to be considered. The latter could further affect the role of groundwater depth in the model, as fine soils have a much higher capillary (and water-holding) capacity. In the model, soil type is subdivided into two sub-parameters (2 and 3, Equation 4 and Table 2), but

this is not further explained. Evidently, this increases the weight of soil type in the regression model.

Groundwater depth: please comment on the reliability of the results in the empty areas (areas without wells or piezometers). Can wells and piezometers be used together, in other words, are all wells installed in unconfined aquifers?

Drainage density: Drainage density was calculated for six river basins. That gives little variation across the area. Is it possible to map drainage density at a higher resolution, e.g. sub-basin scale, or a 10 km grid size? This would increase the importance of this parameter.

Climate: The authors should provide a bit more explanation on the SPEI and particularly the ombrothermic index calculations. Please explain how/where the latter differs significantly from (and is thus not correlated to) the aridity index.

Model development

It is not clear how the parameters were normalized before entering the regression model. How was the soil parameter transformed into a quantitative variable? If all parameters were classified/categorized (as is often done in e.g. factorial regression analysis), this can explain the low influence of the groundwater depth parameter, as there is very little variation (in large part of the area groundwater depth is between 1.5 and 15 m). In this case, I strongly suggest increasing the number of classes for groundwater depth.

Please provide references showing that it is common practice to fit the model on a 5% random subsample. Also explain more lines 264-265.

Results

Overall section 3.2 on environmental conditions mainly consists of an explanation of each of the maps. To support the selection of the five parameters, the authors should provide all the results on correlation and PCA as supplementary material.

The results of the model suggest, as stated by the authors, a low importance of the groundwater depth on explaining the spatial distribution of GDV (eq. 4). However, nothing is said on how this varies locally within in the area. Are there regions where the role of groundwater is larger? Can these regions be identified?

Line 343-344: This requires quite a bit more explanation, but can be easier to follow once the calculation of Ios4 has been better explained in the methods section.

Line 362-364: Please elaborate on this outcome on the Moran index.

In eq. 4 the appearance of Soil type 2 and Soil type 3 is not explained.

The results would become more interesting with:

1) a more local/regional analysis of the explanatory model and the importance of each of the parameters (in particular groundwater depth);

2) an assessment of the use of more/less/different parameters on the final model. It seems the soil type and to a lesser extent the aridity index are the dominating parameters in the regression model. How does a model based solely on these two parameters perform? And what about including a deeper (2nd-3rd layer) soil parameter to account for water holding and capillary rise capacity? Not much can be stated on the importance of soil type for the groundwater storage (as mentioned in line 495) if only the first soil layer is assessed.

3) scenario analysis: what happens if one or more of the parameters (such as climate or groundwater level) change? You do not have to develop climate scenarios, but an assessment of the impact of a relative change in aridity index or groundwater level on the resulting map would be of high added value.

Discussion

Much of the discussion on the modelling approach is more of a summary of the manuscript, particularly lines 425-439. I miss the interpretation of the results obtained

by regression modelling, and the this could further be enriched by the discussion of the added results as proposed above.

PART 2: SUITABILITY MAPPING

Suitability map building

The authors decide to attribute the minimum score (in terms of suitability) to areas where groundwater depth is smaller than 1.5 m, considering that vegetation extracting water from shallow depths belongs to another type of GDV. This distinction between shallow and deep groundwater dependent vegetation, which I indeed think is useful (as most vegetation can use water in the first 1.5 m if present) needs to be briefly elaborated upon.

Line 284-286. I do not understand why shallow groundwater flow would be expected at steep slopes. Normally steeper slopes are found in mountainous areas, where ground-water levels are deep.

Results

The main finding here is that "suitability to GDV in the Alentejo region was mainly driven by soil type". That is obvious, as the weight of this parameter is by far the largest in the suitability index (and given by two soil type variables)! The same holds for the observation "The aridity index also showed a strong influence on GDV's suitability", as the weight of the aridity index is highest following that of soil type. I would strongly suggest analysing alternative weights for each parameter (based for instance on the Delphi panel) and evaluating the corresponding sensitivity of the outcome, as well as the degree of success in the validation procedure.

Line 395-396: "high aridity values restricted GDV's suitability in the south". Again, in my view it is exactly the opposite, as a high aridity is classified as class 3, i.e. of high suitability. In the south in fact aridity index is lowest, indicating the highest aridity and therefore higher suitability for GDV.

I think I might have detected a mistake in the resulting map of Figure 7. Where aridity index (AI) values are low, corresponding suitability value is high (Figure B1b), which means that overall suitability should also increase in those areas (towards the southeast). In the map of Figure 7 the values actually decrease in that area, which is contrary to what would be expected and could result from a negative weight being assigned to this parameter (as it also has a negative coefficient in the regression equation). If this is the case, the presentation and interpretation of the results on suitability mapping needs to be redone.

One example of this wrong interpretation is in lines 376-378, where the authors state that the positive impact of the rivers on the GDV suitability "is due to a higher water availability reflected by the values of omborthermic and aridity indexes. In my view it should be the contrary, i.e. due to a lower water availability, indicating a higher suitability for GDV. Moreover, the positive impact is not visible in the map of Figure 7. And why is there a higher groundwater depth near the river? You would expect groundwater levels to be shallowest near the river.

Another example of this is in discussion section, where the authors state that "The lower suitability to this vegetation in the eastern part of the studied area can be explained by less favorable climatic and geological conditions, resulting from the combination of a high aridity index and low water retention at deep soil layers". It is again the contrary, as the ariditiy index in this (south)eastern area is lower, indicating a higher suitability and therefore higher values on the map of Figure 7. Moreover, it is not clear why the "deep soil" layer is mentioned here now, if only the first soil layer has been analysed.

In Figure 7 please indicate how the values were calculated.

If the authors decide to do the analysis per river basin, they should indicate the river basin boundaries in Figure 1.

Line 382-383: "this high likelihood was hindered by the type of soil present in that area

[Figure]

In terms of soil type in the Tagus basin". That is not true, as the suitability is mostly class 3 in the Tagus river basin.

Line 416-419 belongs to the discussion section, not the results section.

Technical corrections

Overall the text is well written and structured, the main comments above concern the content of the manuscript.

Line 47: decreased precipitation

Line 56: An integrated multidisciplinary methodology

Line 63: do not include

Line 167: listed in Table 1

Line 169: 2.3.1 Slope and soil characterisitcs

Line 205: division of the basin area by the total stream length

Line 244: was evaluated

Line 256: based on the selected variables

Line 277: score from 1 to 3

Line 367: In the GWR model

Line 380: with the exception of

Line 948: Table 2: Groundwater Depth

Line 956: suitable areas for GDV

Figure 1: add catchment limits

Figure 4: change soil colours, or combine

Line 990: what kind of residuals?

Figure 7: consider changing the colour coding

Figure B1: present the maps in the same order as in Figure 4.

---

## Author Comment (AC1) · 25 Nov 2018

Dear Referee1,

Please find enclosed the revised version of the manuscript "Mapping the suitability of groundwater dependent vegetation in a semi-arid Mediterranean area".

All the suggestions were carefully considered and addressed accordingly. In the present letter, you will find the responses to your comments and the changes made, point by point. Particularly, we have clarified comments on the methodology on the calculation of the map of water table. Also, we changed the validation method as suggested. As a result of the introduction of remote sensing datasets to validate the suitability map, we added the author Célia M. Gouveia to the authors list.

We are very thankful for all the comments, which allowed a very significant improvement of the manuscript quality. A version of the new manuscript was uploaded in the journal platform.

All the information included in this manuscript is completely original and has been approved by all authors. The authors declare no conflict of interest. This manuscript has not been published previously or concurrently submitted for publication elsewhere.

Thank you for considering this revised manuscript for publication. Please do not hesitate to contact us if you require further details.

With our best regards, sincerely

####################################################

Referee Comments 1

General Comments Gomes Marques et al. present an analysis of the spatial distribution of groundwater dependent vegetation across the Iberian Peninsula. While the method used is perhaps not as novel as suggested in the text, the paper's main strength lies in the validation of the maps created against a fairly robust eternal dataset. The text is generally well written, although it is not as clear as it could be when discussing how the "model" was parameterized and validated. In general, the paper is a solid contribution to the literature on phreatophytes, but needs revision to enhance its clarity and address some lingering questions about the work.

Specific Comments 1. Throughout: What exactly is meant by a "suitability map"? Suitability for what? Or do you mean suitability of the terrain for hosting phreatophtyes? The concept is fine, but the word choice seems odd.

Answer: We appreciate the reviewer's comment on this matter. With a suitability map we aim to ascertain the suitability of the arboreous phreatophyte species to the climatic and local conditions. To clarify this matter, this information was provided in lines 113-114 of the introduction.

2. Line 147 - 149: How heavily managed are these forestry systems? What species are harvested? And with what methods?

Answer: In the Alentejo region, Cork oak, Holm oak and Stone pine represent 83% of the forest cover, covering about 36% of the geographical area. Cork oak covers 46% of the total forest area of the region, Holm oak 30%, and Stone pine only 7%, according to the last forest inventory. These species were already dominant species in the region and in Portugal two millennia after the beginning of holocene (Bugalho et al. 2009, Proença 2009). Since the 15th century, the agro-silvopastoral systems is largely dominant and steady in the province of Alentejo, on flat terrain. The system has a low tree density (40 to 80 trees/ha), trees being exploited for cork or seeds to feed cattle and the understory cleared of shrubs for pasture, crops (mainly wheat, barley and oats), or both. Tree density is determined by the need for space for pasture or cereal cultivation in the understory (Acacio & Holmgreen 2014) and by climate drivers, especially mean annual precipitation (Joffre 1999, Gouveia & Freitas 2008). Agro-silvopastoral systems are considered semi-natural ecosystems, which must be continually maintained through human management by thinning and understory use through grazing, ploughing and shrub clearing (Huntsinger and Bartolome 1992) to maintain a good productivity, biodiversity and ecosystem services. Cork oak trees are protected and cannot be harvested unless the tree has died, while holm oak trees are maintained with a low tree density (20 to 40 trees/ha) to guard against soil erosion and to provide shelter and shadow for cattle. Holm oaks are known to be more resilient to drought (David et al. 2007) and are mostly distributed in the most xeric area, on the oriental part of the Alentejo region. Some of this information has been added to the discussion section in lines 498-505.

Bugalho M , Plieninger T, Aronson J , Ellatifi M, Gomes D Crespo 2009. Revista especializada Cork oak woodlands on the edge. Ecology, adaptive management, and restoration, 1st edn. Society for Ecological Restoration International, Island Press, WashingtonChapter 3. Open Woodlands: A Diversity of Uses Proença 2009, Galicio-

Portuguese oak forest of Quercus robur and Quercus pyrenaica: biodiversity patterns and forest response to fire. PhD Thesis, https://core.ac.uk/download/pdf/12421965.pdf Acácio V. & Holmgreen M. 2009 Pathways for resilience in Mediterranean cork oak land use systems. Annals of Forest Science 71:5-13. DOI: 10.1007/s13595-012- 0197-0 Joffre R, Rambal S, Ratte PJ. 1999 The dehesa system of southern Spain and Portugal as a natural ecosystem mimic. Agrofor Syst 45:57–79 Gouveia A. & Freitas H., 2008 Intraspecific competition and water use efficiency in Quercus suber: evidence of an optimum tree density? Trees, 22 (2008), pp. 521-530 Huntsinger L, Bartolome JW. 1992 Ecological dynamics of Quercus dominated woodlands in California and southern Spain: a state transition model. Veg 99–100:299–305 David T.S., Henriques M. O., Kurz-Besson C., Nunes J., Valente F., Vaz M., Pereira J. S., Siegwolf R., Chaves M.M., Gazarini L.C. and David J.S. Water use strategies in two co-occurring Mediterranean evergreen oaks: surviving the summer drought. Tree Physiology 27(6): 793-803 https://doi.org/10.1093/treephys/27.6.793

3. Line 170: Cite ASTER GDEM data in the manner requested on the NASA webpage (https://lpdaac.usgs.gov/citing_our_data).

Answer: Done. The missing citation was added in the proper place, in line 162-165. The acknowledgement is already properly done in the acknowledgement section.

4. Line 172: What is meant by superficial water use? Shallow groundwater? Surface water in streams? It's used several places but isn't well defined.

Answer: We substituted the expression superficial water/groundwater by shallow soil water, which refers to the water between 0 and 1.5m depth, in lines 166-167. All water below 1.5m depth was considered as groundwater. This was clarified in lines 301-302.

5. Line 179: What were the three classes? How did soil parameters influence in classification?

Answer: The three classes are presented in Table 4 and an additional explanation to clarify the conditions who led to each scoring was added to lines 175-178. After revision of the suitability model the predictor soil type was no longer included in model fitting, and therefore no further explanation on its effect on the suitability to GDV was added.

6. Line 187-202, Figure 2: The location of piezometer data and well data are quite biased. What is this attributable to and how might it affect the results? It seems like the kriging in the south-central region could be quite problematic. Also, what is the distinction between a well and a piezometer here? This is also concerning because the most dense of the GDV species are roughly in this corridor as well.

Answer: The region under study is an area with a very low population density, which reflected in the lack of points for piezometric level measurement, mainly in unconfined aquifers (∼96% of the total area). Once the correlation between the piezometric level and the topography was successfully tested it was possible to estimate the piezometric level by kriging with external drift in areas where information was not enough. In the studied area, the presence of piezometers (exclusively dedicated structures for piezometric observations) is mostly associated with karst aquifers and areas with high abstraction volumes for public water supply. Oppositely, large wells are mainly devoted to private use and low volume abstractions. To complement the information given on the groundwater level estimation the following sentence was added to the ms, in lines 190-193: "In the studied area, piezometers are exclusively dedicated structures for piezometric observations, in areas with high abstraction volumes for public water supply. Oppositely, large wells are mainly devoted to private use and low volume abstractions."

7. Line 199 -202: I disagree with this method - the groundwater elevations should be determined by first determining the groundwater elevation at the piezometers and then interpolating that through kriging. This should introduce fewer errors and be more realistic.

Answer: The relation between piezometric level and topography is quite high in most of the unconfined aquifers (Marcily, 1986). This relation allows to estimate the piezometry in areas with few piezometric points with enough confidence using external drift kriging. On the other hand, through this method the piezometric surface respects the orographic structures such as valleys, which is not the case with traditional interpolation methods.

"In the aquifer, the water flows toward the outlets, which are the low points in the topography (springs, streams in the surface flow network)." from Ghislain de Marsily, 1986. Quantitative Hydrogeology, Academic Press, Orlando. ISBN: 9780122089169, 9780080917634.

8. Line 303 - 312: The rationale for this validation method is a bit shaky and could use more explanation. If the presence/absence of these trees is a good indicator, why is the rest of the analysis necessary? Is it more expansive? Precise? Also, how is this not a bit autoregressive, given that it sounds like kernal density was derived from the tree data. It starts to make more sense as the results are discussed, but it needs more clarity here. What about using a remote sensing method for validation instead (e.g., Munch et al. 2007, Barron et al. 2012, Gou et. al 2014)? How would that compare?

Answer: After consulting the authors of the EPIC suitability maps (Magalhães M. and Mesquita S.) we understood that the latter were indeed constructed based on the last forest inventories. Therefore, there was indeed autoregression in our validation (see Mesquita, S. and Capelo, J. (2016). Aptidão Bioclimática às Espécies Arbóreas. In Magalhães, M.R. coord): Ordem Ecológica e Desenvolvimento - o futuro do território português. Pp. 63-85. Centro de Estudos de Arquitectura Paisagista "Professor Caldeira Cabral". ISA Press. Lisboa. ISBN: 978-972-8669-64-5).

We thus followed the reviewer suggestion to use remote sensing data to validate our GDV suitability map. We therefore compared our GDV suitability map with NDWI anomalies of June 2005 (extremely dry hydrological year in Portugal) with the median

June NDWI of year 1999-2009 with a dataset shared by Gouveia et al. (2012). We chose the Normalized Difference Water Index (NDWI) for being more representative of water content in vegetation's leaves. This index is thus a proxy for vegetation stress, with low NDWI values representing less leave water content, corresponding to a higher drought stress. The NDWI map we present show in yellow and brown color the areas were the vegetation was more sensitive to the extreme drought of 2005. We obtain a very good agreement between maps that we commented in the results and discussion sections, in lines 477-487, 598-601 and 620-623. The method and dataset used are described in the M&M section lines 341-360.

9. Line 389-397: What soil types were the most likely to host phreatophtyes? What does "soil type 3" represent?

Answer: Soil type 3 represented soils with prevailing water storage at deeper soil depths, and therefore these soils were considered as more likely to host phreatophytes.

10. Line 480-484: This paragraph seems to be saying that there must be some threshold by which no woody species can be supported, even if they are GDVs. These species get replaced by shortlived grasses and forbes, converting savanna to grassland. Is this correct? If so, this seems to contradict the next line about woody vegetation being replaced by shrublands. Wouldn't that presume shrubs are less susceptible to drought than trees? Please clarify.

Answer: The referenced paragraph was removed from the discussion, after the calculation of a new suitability map. Instead we discussed the strong relation between aridity and tree density and the degradation of ecosystems linked to increasing water scarcity, in lines 557-565 and 644-651.

11. Line 495-511: This part of the discussion is problematic, because, as the authors note, the factor expected to be most key is poorly mapped. Regardless, they still say that soil type, as opposed to groundwater depth, is the most influential and claim that soil type defines the capacity for "groundwater storage". This appears to be overreach.

Answer: The most influential factors in the reviewed version of the manuscript are climate drivers. After removing soil types from the final GWR model, the contribution of the GWDepth variable in the model improved (now corresponding to the 3rd most relevant variable in the model), but still remained far less relevant than climate drivers. The part of the discussion was slightly modified according to new results, lines 566-572.

12. Figure 7: This figure needs more color variation. It is difficult to tell moderate, good, and very good apart.

Answer: Done. The colour scale was modified to improve readability in Figure 09.

Technical Corrections 13. Line 88: Replace "genders" with genre

Answer: Done in line 85.

14. Line 102: Replace "5m" with "5 m". Noticed number/unit spacing issues in several other locations as well.

Answer: Done in line 99. All other places where the same issued was found, were corrected.

15. Lines 132 - 135: Replace "chapters" with "sections". But really, this whole paragraph isn't necessary, as the format doesn't deviate from standard expectations.

Answer: Done. The paragraph was eliminated from the manuscript.

16. Line 154: "Proxy for" not "proxy to".

Answer: Done in line 145.

17. Line 175: Is the copyright symbol here a typo?

Answer: No, the copyright symbol is requested to reference the database.

18. Line 201: Don't need to repeatedly cite Spatial Analyst and its version so frequently. Can this be converted to one mention at the beginning of the section?

Answer: Done. We added the sentence "The software used in spatial analysis were ArcGIS® software version 10.4.1 by Esri and R program software version 3.4.2 (R Development Core Team, 2016)." to the ms, under the chapter 2.3 in lines 158-159. All mentions to R and ArcGIS software versions were removed from the text.

19. Line 295: Put equation right after fist mention.

Answer: We have added a general equation of the model (Eq. 4) in lines 330-331 and maintained the equation with the final predictors (Eq. 6) in lines 40-405 so that only in the results section we would present the final model equation used to calculate the suitability map.

20. Line 306: Replace "to a" with "of a".

Answer: This paragraph has been deleted from the ms, after the validation was performed with a different dataset.

21. Line 434, Line 454: Delete first names of authors. Answer: The first reference has been removed from the paragraph. On lines 54, 295 and 517 however, the name "Condesso de Melo" was right, thus remained unchanged.

22. Line 450: Pinpoint is one word.

Answer: Done in line 512.

23. Lines 451-453: Awkward wording makes the sentence hard to parse.

Answer: The sentence was improved in lines 513-515.

24. Line 466: Delete stray "s".

Answer: This paragraph was completely re-structured.

Please also note the supplement to this comment:
https://www.hydrol-earth-syst-sci-discuss.net/hess-2018-208/hess-2018-208-AC1- supplement.pdf

[Figure]

**Fig. 1.** Fig01

[Figure]

**Fig. 2.** Fig04

[Figure]

**Fig. 3.** Fig05

[Figure]

**Fig. 4.** Fig06

[Figure]

**Fig. 5.** Fig07

[Figure]

**Fig. 6.** Fig08

[Figure]

**Fig. 7.** Fig09

[Figure]

**Fig. 8.** Fig10

[Figure]

**Fig. 9.** FigA1

[Figure]

**Fig. 10.** FigA2

[Figure]

**Fig. 11.** FigB1

**Supplement:**

[revised manuscript text omitted]

The final map illustrating the suitability to GDV is shown in Fig. 09. The proportion of each suitability class was quite evenly distributed throughout the study area. The largest area (8 787km$^2$) presented a very poor suitability to GDV but corresponded only to approximately a quarter of the total study area (0.29%). This percentage was followed closely by the moderate suitability to GDV which occupied 0.26% (8000km$^2$). Overall, the two less suitable classes (very poor and poor) represented 0.47% of the study area, whilst the two best ones and the moderate class (very good, good and moderate) represented 0.53%. Consequently, most of the study area showed high to moderate suitability to GDV. The very good and good suitability classes corresponded to the most southern and eastern center area of the Alentejo region, mainly close to the coastal line, passing through the Sado Guadiana river basins. 
[revised manuscript text omitted]

Zomer et al. (2009) attempted to quantify the extent of agroforestry at the global level by performing a
geospatial analysis of remote sensing derived global datasets. They showed that the average tree cover
density within agricultural land can were closely linked to aridity with similar trends for different
geographical areas. Our results agree with these findings since the aridity and ombrothermic indexes were
the most important predictors of GDV density in the Alentejo region, according to our model outcomes.
This is in agreement with former studies linking tree cover/density of Mediterranean oak woodland to
climate drivers derived from precipitation (Gouveia and Freitas 2008, Joffre et al. 1999). Also, Waroux
and Lambin (2012) studied the degradation of the argania woodlands in semi-arid to arid Southwest
Morocco and found that a 44% decline of the forest density was mostly driven by the increasing aridity in
the region between 1970 and 2007. Similarly, many studies carried out on oak woodlands in Italy and
Spain identified drought as the main driving factor of tree die-back and as the main climate warning
threatening oak stands sustainability in the Mediterranean basin (Gentilesca et al. 2017). Tree mortality
linked to increasing drought stresses can also be associated to a geographical shift in vegetation
communities (Lloret et al., 2004). For example, xeric plant species Sahel have expanded in the north of
Sahel since the last half of the 20th century, toward areas of higher rainfall at an average rate of 500 to
600 m yr$^{-1}$ (Gonçalez P., 2001).

In environments with scarce water sources such as the Mediterranean basin, plants have developed
strategies to either avoid or escape drought stress (Chaves et al., 2003). The development of a dimorphic
root systems in woody species is an adaptation strategy to escape drougth (Dinis 2014, David et al.,
2013). When comparing different water limited ecosystems from a global dataset, Schenk and Jackson (2002) showed that rooting depth increased with aridity. Furthermore, a clear relationship between rooting depth and the water table depth was evidenced at global scale (Fan et al. 2017).

[revised manuscript text omitted]

that groundwater depth was only accounting for about 6% of the coefficient variation in the studied area,
against 89% of the variation represented by climate indexes AI and Ios4. Changes in climate conditions
only represents part of the water resources shortage issue in the future. Global-scale changes in human
populations and economic progresses also rules water demand and supply, especially in arid and semi-
arid regions (Vörösmarty et al., 2000). A decrease in useful water resources for human supply can induce
an even higher pressure on groundwater resources (Döll, 2009), aggravating the water table drawdown
caused by climate change (Ertürk et al., 2014). Therefore, additional updates of the model should include
human consumption of groundwater resources, identifying areas of higher population density or intensive
farming. Future model updates should also account for the interaction of deep rooting species with the
surrounding understory species. In particular, shrubs surviving the drought period, which can benefit from
the redistribution of groundwater by deep rooted species (Dawson, 1993; Zou et al., 2005).

**5 Conclusions**

Our results show a highly dominant contribution of water scarcity (Aridity and Ombrothermic indexes) on the density and suitability of deep-rooted groundwater dependent species. The contribution of groundwater depth was much lower than we initially expected, accounting only for 6% of the total coefficient variation. This might be underestimated however, 
[revised manuscript text omitted]

Drainage density; Thickness and Soil type refer to soil properties.

Figure B1 – Predictors maps after score classification. (a) – Aridity Index (AI); (b) – Ombrothermic Index of the
summer quarter and the immediately previous month (Ios4); (c) – Groundwater Depth (GWDepth); (d) – Drainage
density (Dd); (e) – Slope.

**Table 1: Environmental variables for the characterization of the suitability of GDV in the study area.**

| Variable code | Variable type | Source | Resolution and Spatial extent |
|---|---|---|---|
| Slope | Slope (%) | This work | 0.000256 degrees (25m) raster resolution |
| Soil type | Soil type in the first soil layer | SNIAmb (© Agência Portuguesa do Ambiente, I.P., 2017) | Converted from vectorial to 0.000256 degrees (25m) resolution raster |
| Thickness | Soil thickness (cm) | EPIC WebGIS Portugal (Barata et al., 2015) | Converted from vectorial to 0.000256 degrees (25m) resolution raster |
| GWDepth | Depth to groundwater (m) | This work | 0.000256 degrees (25m) raster resolution |
| Dd | Drainage Density | This work | 0.000256 degrees (25m) raster resolution |
| Spei_severe | Number of months with severe SPEI | This work | 0.000256 degrees (25m) raster resolution Time coverage 1950-2010 |
| SPEI_extreme | Number of months with extreme SPEI | This work | 0.000256 degrees (25m) raster resolution Time coverage 1950-2010 |
| AI | Aridity Index | This work | 0.000256 degrees (25m) raster resolution Time coverage 1950-2010 |
| Io | Annual Ombrothermic Index Annual average (January to December) | This work | 0.000256 degrees (25m) raster resolution Time coverage 1950-2010 |
| Ios1 | Ombrothermic Index of the hottest month of the summer quarter (J, J and A) | This work | 0.000256 degrees (25m) raster resolution Time coverage 1950-2010 |
| Ios3 | Ombrothermic Index of the summer quarter (J, J and A) | This work | 0.000256 degrees (25m) raster resolution Time coverage 1950-2010 |
| Ios4 | Ombrothermic Index of the summer quarter and the immediately previous month (M, J, J and A) | This work | 0.000256 degrees (25m) raster resolution Time coverage 1950-2010 |

**Table 2: Effect of variable removal in the performance of GWR model linking the Kernel density of *Quercus***
***suber*, *Quercus ilex* and *Pinus pinea* to predictors Aridity Index (AI); Ombrothermic Index of the summer**
**quarter and the immediately previous month (Ios4); Groundwater Depth (GWDepth); Drainage density (Dd);**
**Slope; and Soil type. The model with all predictors is highlighted in grey and the final model used in this study**
**is in bold.**

| Type | Model | Discarded predictor | AICc | Quasi-global R² |
|------|-------|---------------------|------|-----------------|
| GWR | Density~ios4 +ai + slope + Dd + GWDepth + soiltype | | 27389.74 | 0.926481 |
| GWR | Density~ios4 + slope + Dd + GWDepth + soiltype | Ai | 28695.14 | 0.9085754 |
| GWR | Density~ai + slope + Dd + GWDepth + soiltype | Ios4 | 28626.88 | 0.9095033 |
| GWR | Density~ios4 +ai + GWDepth + slope + soiltype | Dd | 27909.86 | 0.9184337 |
| GWR | Density~ios4 +ai + Dd + GWDepth + soiltype | Slope | 27429.55 | 0.924176 |
| GWR | Density~ios4 +ai + Dd + slope+ soiltype | GWDepth | 27742.67 | 0.9208344 |
| GWR | **Density~ios4 +ai + Dd + GWDepth + slope** | **Soiltype 3 levels** | **18050.76** | **0.9916192** |

**Table 3: Comparison of Adjusted R-squared and second-order Akaike Information Criterion (AICc) between**
**the simple linear regression and the GWR model.**

| Model | R-squared | AICc | p-value |
|-------|-----------|------|---------|
| OLS | 0.02 | 42720 | <0.001 |
| GWR | 0.99 * | 18851 | - |

*Quasi-global R$^2$

**Table 4: Classification scores for each predictor. A score of 3 was given to highly suitable areas and 1 to highly**
**less suitable areas for GDV.**

| Predictor | Class | Score |
|---|---|---|
| Slope | 0%-5% | 1 |
| | 5%-10% | 2 |
| | >10% | 3 |
| | | |
| Groundwater Depth | >15 m | 1 |
| | 1.5m-15m | 3 |
| | ≤1.5m | 1 |
| Aridity Index | 0.6-0.68 | 1 |
| | 0.68-0.75 | 2 |
| | ≥0.75 | 3 |
| Ios4 | <0.28 | 1 |
| | 0.28-0.64 | 2 |
| | ≥0.64 | 3 |
| Dd | ≤0.5 | 3 |
| | >0.5 | 1 |

[Figure]

**Figure 01: Study area. On the left the location of Alentejo in the Iberian Peninsula; on the right, the elevation characterization of the study area with the main river courses from Tagus, Sado and Guadiana basins. Names of the main rivers are indicated near to their location in the map.**

[Figure]

**Figure 02: Large well and piezometer data points used for groundwater depth calculation. Squares represent piezometers data points and triangle represent large well data points.**

[Figure]

**Figure 03: Map of Kernel Density weighted by cover percentage of** *Q. suber, Q. ilex* **and** *P. pinea***.**

[Figure]

**Figure 04: Map of environmental layers used in model fitting. (a) – Soil type; (b) – Slope; (c) – Groundwater**
**Depth (Depth); (d) – Ombrothermic Index of the summer quarter and the immediately previous month (Ios4);**
**(e) – Aridity Index (AI).**

[Figure]

**Figure 05: Spatial distribution of local $R^2$ from the fitting of the Geographically Weighted Regression.**

[Figure]

**Figure 06: Spatial distribution of model residuals from the fitting of the Geographically Weighted Regression**
**(a) and Simple Linear model(b).**

[Figure]

**Figure 07: Map of local model coefficients for each variable. (a) – Aridity Index; (b) - Ombrothermic Index of**
**the summer quarter and the immediately previous month (Ios4); (c) – Groundwater Depth (GWDepth); (d) –**
**Drainage density and (e) – Slope.**

[Figure]

**Figure 08 – Boxplot of GWR model coefficient values for each predictor. AI is Aridity Index; Ios4 is the**
**ombrothemic index of the hottest month of the summer quarter and the immediately previous month; GWDepth**
**is Groundwater Depth and Dd is drainage density.**

[Figure]

**Figure 09: Suitability map for Groundwater Dependent Vegetation.**

[Figure]

**Figure 10: Validation map corresponding to the NDWI anomaly considering the months of June, July and**
**August of the extremely dry year of 2005 in the Alentejo area. Brown colors (corresponding to more negative**
**values) indicate vegetation in water stress.**

**Appendix A**

**Table A1: Classification scores for the soil type predictor.**

| Predictor | Class | Score |
|---|---|---|
| Soil type | Eutric Cambisols; Dystric Regosol; Humic Cambisols; Haplic Luvisols; Gleyic Luvisols; Ferric Luvisols; Chromic Luvisols associated with Haplic Luvisols; Ortic Podzols | 3 |
| | Calcaric Cambisols; Dystric Regosol associated with Umbric Leptosols; Eutric Regosols; Vertic Luvisols; Eutric Planosols; Cambic Arenosols | 2 |
| | Chromic Cambisols; Eutric fluvisols; Chromic Luvisols; Gleyic Solonchak; Eutric Vertisols | 1 |

[Figure]

**Figure A1: Boxplot of the main predictors for the final Geographically Weighted Regression model fitting**
**(top) and the response variable (below), for the total data (left) and for the 5% subsample (right).**

[Figure]

**Figure A2: Correlation plot between all environmental variables expected to affect the presence of the Groundwater Dependent Vegetation. Ios1, Ios3, Ios4 are ombrothemic indices of, respectively, the hottest month of the summer quarter, the summer quarter and the summer quarter and the immediately previous month; Io is the annual ombrothermic index, Spei_extreme and Spei_severe are, respectively, the number of months with extreme and severe Standardized Precipitation Evapotranspiration Index; AI is Aridity Index; GWDepth is Groundwater Depth, ; Dd is the Drainage density; Thickness and Soiltype refer to soil properties.**

**Table A2: Correlations between predictor variables and principal component axis. The most important predictors for each axis (when squared correlation is above 0.3) are showed in bold. The cumulative proportion of variance explained by each principal component axis is shown at the bottom of the table**

| | PC1 | PC2 | PC3 | PC4 | PC5 | PC6 | PC7 | PC8 | PC9 | PC10 | PC11 | PC12 |
|---|---|---|---|---|---|---|---|---|---|---|---|---|
| **Slope** | <0.001 | **0.32** | 0.13 | 0.06 | 0.14 | 0.18 | 0.18 | <0.001 | 0.03 | 0.03 | <0.01 | <0.01 |
| **AI** | **0.94** | <0.001 | 0.01 | <0.01 | <0.001 | <0.01 | <0.001 | <0.001 | 0.22 | **0.33** | **0.40** | **0.68** |
| **Io** | **0.93** | <0.01 | 0.01 | <0.01 | <0.001 | <0.01 | <0.001 | <0.001 | 0.24 | **0.38** | 0.24 | **0.72** |
| **Ios1** | **0.89** | 0.02 | 0.04 | 0.01 | <0.001 | <0.001 | <0.001 | 0.02 | 0.03 | 0.14 | **0.82** | 0.10 |
| **Ios3** | 0.21 | 0.18 | **0.47** | <0.01 | <0.01 | <0.001 | <0.01 | 0.11 | **0.64** | **0.33** | <0.01 | <0.01 |
| **Ios4** | 0.15 | 0.19 | **0.53** | <0.001 | <0.001 | <0.01 | <0.001 | **0.33** | **0.53** | **0.33** | 0.05 | <0.01 |
| **Spei_severe** | **0.66** | 0.08 | 0.01 | <0.01 | <0.001 | -0.02 | <0.01 | **0.77** | 0.08 | **0.40** | 0.11 | 0.01 |
| **Spei_extreme** | **0.72** | 0.01 | 0.04 | 0.05 | <0.01 | <0.001 | <0.01 | **0.36** | **0.44** | 0.57 | 0.29 | 0.05 |
| **GWDepth** | 0.16 | 0.05 | 0.01 | **0.33** | 0.14 | 0.26 | 0.06 | 0.06 | 0.04 | 0.06 | 0.04 | 0.01 |
| **Dd** | <0.01 | 0.25 | 0.11 | 0.20 | 0.08 | **0.32** | <0.01 | 0.29 | 0.06 | 0.04 | <0.01 | <0.01 |
| **Soil type** | 0.02 | 0.19 | 0.03 | 0.22 | **0.46** | 0.05 | 0.02 | 0.06 | 0.03 | 0.05 | 0.03 | <0.01 |
| **Thickness** | 0.02 | **0.46** | 0.09 | 0.03 | 0.06 | 0.01 | **0.32** | 0.11 | 0.03 | 0.09 | 0.01 | <0.01 |
| **Cumulative proportion** | 0.39 | 0.54 | 0.66 | 0.74 | 0.81 | 0.88 | 0.93 | 0.96 | 0.98 | 0.99 | 0.99 | 1 |

**Appendix B**

[Figure]

**Figure B1 – Predictors maps after score reclassification. (a) – Aridity Index (AI); (b) – Ombrothermic Index of**
**the summer quarter and the immediately previous month (Ios4); (c) – Groundwater Depth (GWDepth); (d) –**
**Drainage density (Dd); (e) – Slope.**

---

## Author Comment (AC2) · 25 Nov 2018

Dear Referee2, We are very grateful for your rigorous assessment and the valuable comments and suggestions you provided to improve our manuscript. Please find enclosed the revised version of the manuscript "Mapping the suitability of groundwater dependent vegetation in a semi-arid Mediterranean area". We believe that all your suggestions were carefully addressed. In the present letter, you will find our responses to each comments and change made. Particularly, we have corrected the methodology to calculate drainage density. We also clarified the error you detected regarding a mistake in the map of Figure 7 where for a low aridity index (AI) we predicted a high suitability value (Figure B1b), which you attributed to the negative sign assigned to AI used as weighting factor. We evaluated the impact of each predictor on the final model,

and discovered that soil type actually considerably worsened the performance of the GWR model. We therefore decided to remove it from the final model equation selected to build the suitability map. We also attempted to provide a better evaluation of the importance of each predictor in the final model, and improved the discussion section accordingly. A version of the new manuscript was uploaded in the journal platform. All the information included in this manuscript is completely original and has been approved by all authors. The authors declare no conflict of interest. This manuscript has not been published previously or concurrently submitted for publication elsewhere. Also we thank you for considering this revised manuscript for publication. Please do not hesitate to contact us for any further needed detail.

With our best regards, sincerely

##############################################################

Referee Comments 2

General comments

The current manuscript provides an interesting insight into the use of mapping and spatial regression to assess the occurrence of groundwater dependent vegetation (GDV). Such maps can subsequently be used to predict the effect of change in any of the explanatory variables, such as climate or groundwater depth, on the spatial distribution of GDV in an area. The paper is well written and structured, and is subdivided into two parts: the first on the building of regression models for predicting GDV occurrence based on actual data, and the second part where a parameter-based index is calculated to construct so-called "suitability maps" for GDV. While I find the first part strong and with high potential of publication by adding a scenario analysis, my main concern lies with the second part. In my opinion this part is less well developed and the interpretation of the results is largely straightforward, and to a certain degree incorrect. Interpretation is largely straightforward because of the large bias in weighting of the parameters, where the contribution of soil largely exceeds all other parameters, thus making it essentially a soil map. Additionally, interpretation is to a certain degree incorrect, due to an apparent mistake in the generation of the suitability map, where (inadvertently) a negative weighting was assigned to the aridity index, resulting in the inverse impact of this parameter on the soil map. I would therefore recommend focusing and further elaborating on the regression modelling, as I will specify in my detailed comments below.

Specific comments

Abstract

Line 13-19: The first part of the abstract is more of an introduction. I suggest starting with what was actually done (line 20). Moreover, groundwater depletion will not occur merely as a result of climate change. Finally, as groundwater level seems to have such a low impact in the regression model, the question rises to what extent groundwater depletion will play a role in the spatial distribution of GDV.

Answer: The first paragraph of the abstract was deleted. The abstract was corrected according to new results.

Line 48: When referring to climate change impact studies on recharge in Mediterranean areas add the paper of doi.org/10.1007/s10113-012-0377-3, where such a study is being reported.

Answer: Done. The suggested reference has been added in the text lines 45-46 and reference list lines 1038-1041.

PART 1: REGRESSION MODELLING

Parameter selection for the regression model

Soil type: the authors only use the first layer of the soil. To understand the importance of capillary rise feeding into the root zone, the texture of deeper soils also needs to be considered. The latter could further affect the role of groundwater depth in the model,

as fine soils have a much higher capillary (and water-holding) capacity. In the model, soil type is subdivided into two sub-parameters (2 and 3, Equation 4 and Table 2), but this is not further explained. Evidently, this increases the weight of soil type in the regression model.

Answer: The classification of soils into 3 categories was explained in lines 175-178 of the M&M section and in Table A1 in appendix A. This predictor was removed from the model fitting after revision from the authors. It has not been possible to add the texture of deeper horizon into our study because such information was only available on inaccessible printed maps. Unfortunately, no such digital data were available when the manuscript was prepared or revised.

Groundwater depth: please comment on the reliability of the results in the empty areas (areas without wells or piezometers). Can wells and piezometers be used together, in other words, are all wells installed in unconfined aquifers?

Answer: Please notice the previous answer to the reviewer 1. The region under study is an area with a very low population density, which reflected in the lack of points for piezometric level measurement, mainly in unconfined aquifers ($\sim$96% of the total area). Once the correlation between the piezometric level and the topography was successfully tested it was possible to estimate the piezometric level by kriging with external drift in areas where information was not enough. The estimation of the groundwater depth did not consider the simultaneous use of large wells and piezometers, with exception of the northwestern area, due to the lack of large wells. In the studied area, the presence of piezometers (exclusively dedicated structures for piezometric observations) is mostly associated with karst aquifers and areas with high abstraction volumes for public water supply. Oppositely, large wells are mainly devoted to private use and low volume abstractions.

Drainage density: Drainage density was calculated for six river basins. That gives little variation across the area. Is it possible to map drainage density at a higher resolution, e.g. sub-basin scale, or a 10 km grid size? This would increase the importance of this parameter.

Answer: Indeed, there was little spatial variation of the drainage density for the studied area, therefore, as suggested by the reviewer, we recalculated this variable considering a 10km resolution grid. The methodology concerning this calculation was corrected in the ms, in lines 202-204. Due to the creation of a new drainage density map, we performed a reassessment of the multicollinearity between variables and the selection of predictors (see section 2.4). This implied recalculating Pearson's coefficients and Principal Components Analysis (PCA), presented in table A2 and figure A2 both in appendix A. It also affected predictors and coefficients in the model linking GDV density to environmental predictors). By affecting model development, model performance (Tables 2 and 3), suitability and coefficient maps were also affected (Figures 7 to 9).

Climate: The authors should provide a bit more explanation on the SPEI and particularly the ombrothermic index calculations. Please explain how/where the latter differs significantly from (and is thus not correlated to) the aridity index.

Answer: Done. Clear explanations on SPEI calculations were already provided in lines 216-223. Since the SPEI predictor was excluded from modeling further explanation would unnecessarily extend the manuscript length. We however briefly altered paragraph lines 224-227, to better explain the discrepancies between SPEI and Ios, and to clarify Ios calculation according to Table 1.

Model development

It is not clear how the parameters were normalized before entering the regression model.

Answer: The explanation of the normalization based on the z-score function was improved and changed to the M&M section, under the chapter 2.5 of Model development. Variables were standardized before entering the regression model through the calculation of a z-score. To clarify how the standardization was done the following sentence was added to lines 266-268 of the ms: "This allows to create standardized scores for each variable, by subtracting the mean of all data points from each individual data point, then divide those points by the standard deviation of all points, so that the mean of each z-predictor is zero and the deviation is 1.".

How was the soil parameter transformed into a quantitative variable? If all parameters were classified/categorized (as is often done in e.g. factorial regression analysis), this can explain the low influence of the groundwater depth parameter, as there is very little variation (in large part of the area groundwater depth is between 1.5 and 15 m). In this case, I strongly suggest increasing the number of classes for groundwater depth.

Answer: We greatly appreciate your comment. First, we would like to clarify that the main purpose of the model construction is to attribute coefficients of importance to each variable, so that these coefficients can be applied to classification scores given to each variable by expert judgement (table 4) and return a suitability map to groundwater dependent vegetation. This will allow the production of a suitability map where the coefficients of importance applied to each variable were calculated empirically. Therefore, the classification scores given to each variable were not applied in the model calculation, but rather after the local model coefficients were calculated (as a mean to construct the suitability map). The soil parameter was used has a numeric categorical variable (with the values given initially from 1 to 3), through the use of the function as.factor() in R. The usage of this function will insure that the factor is seen as nominal and not as ordinal. Because the remaining variables showed continuous values, only the soil type variable was categorized, and the remaining variables used to run the model were continuous. The scoring applied is presented is Table A1and the explanation in lines 175-178 of the methods section. The reviewer is correct about the groundwater depth variable and its very low variation above 15m. As further explained below, it has not been possible to increase the number of classes for GWDepth, for the weighting factors to be correctly applied to the GWDepth layer in the multicriteria analysis. To overcome this situation the values of water depth above 15 m were replaced by a value above 15m (15.1m), in order to emphasize the variation observed between 0 and 15m depth, which matters the most to GDV. These values were only used for the model fitting. The species used as proxies for groundwater dependent vegetation are less probable to use water at depths lower than 15m, and so all the range of values above this threshold would be considered as inaccessible by those species.

Please provide references showing that it is common practice to fit the model on a 5% random subsample. Also explain more lines 264-265.

Answer: The sub-sampling size was mostly dictated by computing limitation in the sense that the random subsample size was decreased down to 5% until the GWR model could be fitted. The mean distance between neighbor points using 5% of the original dataset was about 6 km, with a maximum distance of 15km. Nevertheless, we could find a few studies using a 10% random sub-sample of the data corresponding to a 10km resolution grid to perfrom GWR modeling (Bertrand R., 2017), as well as linear regression (Bertrand et al. 2016). The authors were using such subsampling to restrain autocorrelation issues according to Kühn (2007). We modified our text to include those references as well as the benefit for autocorrelation issues in our study, lines 272-277 and in the reference list, lines 743-745 and 900-901. In addition, we calculated basic statistic indicators for the totality of the data and compared with the random subsample. Results are presented in line 276- 279 of the ms and in Figure A1 in appendix A.

Bertrand R. Unequal contributions of species′ persistence and migration on plant communities′ response to climate warming throughout forests. bioRxiv, doi.org/10.1101/217497, 2017. Bertrand R., Riofrío-Dillon G., Lenoir J., Drapier J., de Ruffray P., Gégout J.C., Loreau M.. Ecological constraints increase the climatic debt in forests. Nature Communications 7, 12643, doi: 10.1038/ncomms12643, 2016. Kühn, I. Incorporating spatial autocorrelation may invert observed patterns. Divers. Distrib. 13, 66–69 (2007).

Results Overall section 3.2 on environmental conditions mainly consists of an explanation of each of the maps. To support the selection of the five parameters, the authors should provide all the results on correlation and PCA as supplementary material.

Answer: Done. PCA results have now been provided as supplementary material in Table A2 in appendix A and was modified according to new results, due to the construction of a new map of drainage density.

The results of the model suggest, as stated by the authors, a low importance of the groundwater depth on explaining the spatial distribution of GDV (eq. 4). However, nothing is said on how this varies locally within in the area. Are there regions where the role of groundwater is larger? Can these regions be identified?

Answer: We have plotted the local coefficients of all predictors and present it in Figures 07 and 08. In addition, we added some paragraphs with an explanation of the spatial variation of each predictor in the results section, in lines 424-443 and in the discussion section in lines 524-528.

Line 343-344: This requires quite a bit more explanation, but can be easier to follow once the calculation of Ios4 has been better explained in the methods section.

Answer: We altered lines 224-227 and 234-235 to better explain the discrepancies between SPEI and Ios and to clarify Ios calculation according to Table 1.

Line 362-364: Please elaborate on this outcome on the Moran index.

Answer: Bibliographic references for the Moran Index were added in line 416 and the respective references were added to the bibliography. In addition, we extended the results explanation on the Moran Index and the z-score, in lines 415-419.

In eq. 4 the appearance of Soil type 2 and Soil type 3 is not explained.

Answer: After revising the methodology and predictors selection, the predictor soil type was no longer included in the model.
The results would become more interesting with: 1) a more local/regional analysis of the explanatory model and the importance of each of the parameters (in particular groundwater depth);

Answer: The model equation was substituted by a local one including the proportion of the local coefficients from the total variability of all the coefficients for each local GWR model. Local relative coefficients were considered as weighting factors instead of median values (please see revised Equation 6 in the manuscript). We also added a figure corresponding to the local variation of each coefficient in Figure 07 and commented the variations in the result section, in lines 424-443 and in the discussion section, lines 543-528 and in the conclusion section lines 658-661. The relative importance of each variable in the final model is now shown in Figure 08, representing the distribution of the local coefficient values in a box plot.

2) an assessment of the use of more/less/different parameters on the final model. It seems the soil type and to a lesser extent the aridity index are the dominating parameters in the regression model. How does a model based solely on these two parameters perform? And what about including a deeper (2nd-3rd layer) soil parameter to account for water holding and capillary rise capacity? Not much can be stated on the importance of soil type for the groundwater storage (as mentioned in line 495) if only the first soil layer is assessed.

Answer: We tested the effect of removing one of the variables on the model performance and found out that the model performance increased notably when soil types was removed (AIC divided by factor 2, Table 2). The removal of any other variable however, did not seem to impact the model performance as compared to the equation including all formerly selected variables. Therefore, we excluded soil types from the final GWR model and the rest of our analyses and multicriteria analysis. Data on deeper soil parameters was not available for the study area and therefore that information could not be included in the model.

3) scenario analysis: what happens if one or more of the parameters (such as climate or groundwater level) change? You do not have to develop climate scenarios, but an assessment of the impact of a relative change in aridity index or groundwater level on the resulting map would be of high added value.

Answer: The development of climate scenarios or the assessment of the impact of aridity change on GDV suitability was out of the scope of this manuscript, since the actual manuscript is quite long already. The full assessment of climate changes impacts and corresponding uncertainties will be the focus of our next publication. We calculated preliminary results of the relative change of AI and IOS4 expected for the near future (Table a and b below). Our ongoing calculations based on scenarios RCP 4.5 and 8.5 show that AI and IOS4 climate indexes are going to decrease in the studied region (-14 to -33% within 2099), drifting from a mostly dry sub-humid climate (0.5<AI<0.65) to a mostly semi-arid one (0.2<AI<0.5) by 2099 in scenario RCP8.5, and according to the classification of Middleton et al. (1992). Ios4 is also going to suffer a huge drop (-42 to -58% within 2099). Also, while most of the territory could be considered as non-Mediterranean based on the ombrothermic index (Ios4>2) during the historical period 1971-2000, it is becoming mostly Mediterranean by 2099 in scenario RCP8.5 and according to the classification of Rivas-Martínez et al. (2011). To include such preliminary results in the M&M, result and discussion regarding climate change impact would imply to considerably increase the manuscript, while providing an incomplete picture of the changes and associated uncertainties. We therefore chose not to include the suggested assessment of the impact of a relative change in aridity index or groundwater level on the resulting map in this manuscript. Nevertheless, we know discussed the relative importance of each predictor in our final map, which give an insight of how the groundwater dependent vegetation is expected to be affected according to the predicted increased aridity, lines 641-648, 658-662 and 668-673.

Table a. Mean relative changes expected for AI and IOS4 in the near future according to climate changes scenarios RCP 4.5 and 8.5, and respective standard deviations.

Changes were computed considering 30 yr means obtained from an ENSEMBLE of eleven EU-CORDEX climate models.

Table b. Evolution of percentiles 10 and 90 values of AI and IOS4 in Alentejo from the present to the near future according to scenarios RCP 4.5 and 8.5. ECAD are observed values for the reference period 1971-2000. Historical values for the reference period 1971-2000 as well as predicted values for the future were simulated by an Ensemble of EU-Cordex models.

Discussion Much of the discussion on the modelling approach is more of a summary of the manuscript, particularly lines 425-439. I miss the interpretation of the results obtained by regression modelling, and the this could further be enriched by the discussion of the added results as proposed above.

Answer: The discussion section has been considerably modified. The dominant impact of aridity on tree density and GDV suitability is now much more discussed, as well as the lower impact of groundwater depth. The relative weight of each predictor is also discussed and considered in the key limitations and conclusions sections. (see mostly lines 443-444, 537-542, 641-646 and 658-662).

PART 2: SUITABILITY MAPPING Suitability map building The authors decide to attribute the minimum score (in terms of suitability) to areas where groundwater depth is smaller than 1.5 m, considering that vegetation extracting water from shallow depths belongs to another type of GDV. This distinction between shallow and deep groundwater dependent vegetation, which I indeed think is useful (as most vegetation can use water in the first 1.5 m if present) needs to be briefly elaborated upon.

Answer: Providing a less probable score to host the GDV to the 0-1.5m GWDepth was made to exclude riparian vegetation and shrubby species which primarily use the water from streams and the superficial soil layer. An additional explanation and references were added to the manuscript in lines 302-307: "The depth class between 0 and 1.5m was based on the riparian vegetation in semi-arid Mediterranean areas which is mainly composed of shrub communities (Salinas et al., 2000) and present a mean rooting depths between 1 and 2m (Schenk and Jackson, 2002). The most common tree species rooting depth in riparian ecosystems is normally similar to the depth of fine sediment not reaching gravel substrates (Singer et al., 2012), but not reaching levels as deep as deep-rooted species.".

Line 284-286. I do not understand why shallow groundwater flow would be expected at steep slopes. Normally steeper slopes are found in mountainous areas, where groundwater levels are deep.

Answer: The reviewer is correct, and we appreciate for noticing the error. The sentence was corrected in lines 308-309 and the term water flow was substituted by runoff.

Results The main finding here is that "suitability to GDV in the Alentejo region was mainly driven by soil type". That is obvious, as the weight of this parameter is by far the largest in the suitability index (and given by two soil type variables)! The same holds for the observation "The aridity index also showed a strong influence on GDV's suitability", as the weight of the aridity index is highest following that of soil type. I would strongly suggest analysing alternative weights for each parameter (based for instance on the Delphi panel) and evaluating the corresponding sensitivity of the outcome, as well as the degree of success in the validation procedure.

Answer: Unfortunately, it has not been possible for us to perform this analysis within the time provided to review our manuscript. We hope that the discussion on the relative importance of each predictor in the model will be satisfactory enough for the reviewer, considering that every other request was fulfilled.

Line 395-396: "high aridity values restricted GDV's suitability in the south". Again, in my view it is exactly the opposite, as a high aridity is classified as class 3, i.e. of high suitability. In the south in fact aridity index is lowest, indicating the highest aridity and therefore higher suitability for GDV. I think I might have detected a mistake in the resulting map of Figure 7. Where aridity index (AI) values are low, corresponding suitability value is high (Figure B1b), which means that overall suitability should also increase in those areas (towards the southeast). In the map of Figure 7 the values actually decrease in that area, which is contrary to what would be expected and could result from a negative weight being assigned to this parameter (as it also has a negative coefficient in the regression equation). If this is the case, the presentation and interpretation of the results on suitability mapping needs to be redone.

Answer: After thoroughly verifying the model calculations (Eq. 6) and the weighting factors used for the final multicriteria analysis (Figure 09), we must agree with the reviewer that it was a mistake to apply a negative weighting to the Aridity Index layer. Indeed, where real values of AI were low (indicating a more arid area), our scoring was high in the multicriteria analysis. To directly apply a negative weight, we should have the real predictor values and the predictor scores co-varying (or growing) accordingly. We also verified that the same logic should be applied to the other quantitative variables Slope, Ios4 and GW Depth, since scoring and real values variation were opposite. However, in the revised manuscript we have adopted different scores for the Aridity Index (scores 1, 3, 2) which were not varying linearly, and it was no longer possible to apply a linear scoring. The same was applicable in the case of GWDepth, when we came to a dead end because scoring was not varying linearly according to class values (scores 1, 3, 1). As a solution we calculated the proportion of each local coefficients from the total variability of all the coefficients for each local GWR model (Eq. 6) as a local weighting factors reflecting the relative relevance of each predictor locally. This allowed us to apply scores not varying linearly and still interpreting the results easily. This way, the weighting factor obtained in from the proportions could be directly and correctly applied to the GW Depth and Aridity Index layers.

One example of this wrong interpretation is in lines 376-378, where the authors state that the positive impact of the rivers on the GDV suitability "is due to a higher water availability reflected by the values of omborthermic and aridity indexes. In my view it should be the contrary, i.e. due to a lower water availability, indicating a higher suit-

ability for GDV. Moreover, the positive impact is not visible in the map of Figure 7. And why is there a higher groundwater depth near the river? You would expect groundwater levels to be shallowest near the river. Another example of this is in discussion section, where the authors state that "The lower suitability to this vegetation in the eastern part of the studied area can be explained by less favorable climatic and geological conditions, resulting from the combination of a high aridity index and low water retention at deep soil layers". It is again the contrary, as the ariditiy index in this (south)eastern area is lower, indicating a higher suitability and therefore higher values on the map of Figure 7. Moreover, it is not clear why the "deep soil" layer is mentioned here now, if only the first soil layer has been analysed.

Answer: We appreciate the referee comment and agree with it. Indeed, groundwater levels are expected to be higher near the river, mainly in alluvial aquifers (associated with gentle slopes). However, the opposite also occurs in areas where the rivers are associated with hard rock aquifers (generally associated with steep slopes) and where the relation surface/groundwater is more heterogeneous. The slope predictor, also considered in the presented methodology, distinguishes these occurrences.

In Figure 7 please indicate how the values were calculated.

Answer: A thorough explanation was added in the methods section, in lines 327-333. The explanation in the methods section in the ms reads: "The final GIS multicriteria analysis was performed using the Spatial Analyst Tool by applying local model equations obtained for each of the 6242 coordinates of the Alentejo map (Eq.4), Suitability = Intercept + coef1 * [real value X1] + coef2 * [real value X2] + coef3 * [real value X3] + ..., (4) with brackets representing the reclassified GIS X layer corresponding to the scoring and coefpx indicating the relative proportion for the predictor x.". The final equation used for the calculation of the suitability map is presented in the results section, in lines 406-406, and is presented in the Equation 1 below. Suitability = Intercept + AI coefp * [reclassified AI value] + Ios4 coefp * [reclassified Ios4 value] + GWDepth coefp * [reclassified GWDepth value] + Dd coefp * [reclassified Dd value] + slope coefp

* [reclassified slope value], (1) If the authors decide to do the analysis per river basin, they should indicate the river basin boundaries in Figure 1.

Answer: As suggested by reviewer 1, we decided to use a 10 km grid mesh instead. The methodology was corrected in lines 203-204.

Line 382-383: "this high likelihood was hindered by the type of soil present in that área In terms of soil type in the Tagus basin". That is not true, as the suitability is mostly class 3 in the Tagus river basin.

Answer: The sentence was deleted according to the new results of the revised manuscript.

Line 416-419 belongs to the discussion section, not the results section.

Answer: The paragraph was deleted according to the new validation performed in the revised manuscript.

Technical corrections Overall the text is well written and structured, the main comments above concern the content of the manuscript. Line 47: decreased precipitation

Answer: Done in line 44.

Line 56: An integrated multidisciplinary methodology

Answer: Done in line 53.

Line 63: do not include

Answer: Done in line 60.

Line 167: listed in Table 1

Answer: Done in line 158.

Line 169: 2.3.1 Slope and soil characteristics

Answer: Done in line 161.

Line 205: division of the basin area by the total stream length

Answer: Done in line 204.

Line 244: was evaluated

Answer: Done in line 246.

Line 256: based on the selected variables

Answer: Done in line 259.

Line 277: score from 1 to 3

Answer: Done in line 296.

Line 367: In the GWR model

Answer: Done in line 421.

Line 380: with the exception of

Answer: Done in line 453-454.

Line 948: Table 2: Groundwater Depth

Answer: This table was eliminated from the revised manuscript. The variable Groundwater depth was, from now on, referenced as GWDepth.

Line 956: suitable areas for GDV

Answer: Done in line 1147, in Table 4.

Figure 1: add catchment limits

Answer: Done in the new version of fig01.

Figure 4: change soil colours, or combine

Answer: The map of soil type was removed form Figure 04.

Line 990: what kind of residuals?

Answer: This was clarified in line 1171.

Figure 7: consider changing the colour coding

Answer: A new suitability map was calculated, with new colors by classes, and was added as Figure09.

Figure B1: present the maps in the same order as in Figure 4.

Answer: Done in Figure B1.

Please also note the supplement to this comment:
https://www.hydrol-earth-syst-sci-discuss.net/hess-2018-208/hess-2018-208-AC2-supplement.pdf

―――――――――――――――――――

[Figure]

**Fig. 1.** Fig01

[Figure]

**Fig. 2.** Fig04

[Figure]

**Fig. 3.** Fig05

**Fig. 4.** Fig06

[Figure]

**Fig. 5.** Fig07

[Figure]

**Fig. 6.** Fig08

**GDV Suitability**

- Very Good (> 2.50)
- Good (2.140 to 2.499)
- Moderate (1.760 to 2.139)
- Poor (1.370 to 1.759)
- Very Poor (>1.369)

**Fig. 7.** Fig09

[Figure]

**Fig. 8.** Fig10

[Figure]

**Fig. 9.** FigA1

[Figure]

Fig. 10. FigA2

Classification Score

**Fig. 11.** FigB1

**Supplement:**

[revised manuscript text omitted]

The final map illustrating the suitability to GDV is shown in Fig. 09. The proportion of each suitability class was quite evenly distributed throughout the study area. The largest area (8 787km$^2$) presented a very poor suitability to GDV but corresponded only to approximately a quarter of the total study area (0.29%). This percentage was followed closely by the moderate suitability to GDV which occupied 0.26% (8000km$^2$). Overall, the two less suitable classes (very poor and poor) represented 0.47% of the study area, whilst the two best ones and the moderate class (very good, good and moderate) represented 0.53%. Consequently, most of the study area showed high to moderate suitability to GDV. The very good and good suitability classes corresponded to the most southern and eastern center area of the Alentejo region, mainly close to the coastal line, passing through the Sado Guadiana river basins. 
[revised manuscript text omitted]

Zomer et al. (2009) attempted to quantify the extent of agroforestry at the global level by performing a
geospatial analysis of remote sensing derived global datasets. They showed that the average tree cover
density within agricultural land can were closely linked to aridity with similar trends for different
geographical areas. Our results agree with these findings since the aridity and ombrothermic indexes were
the most important predictors of GDV density in the Alentejo region, according to our model outcomes.
This is in agreement with former studies linking tree cover/density of Mediterranean oak woodland to
climate drivers derived from precipitation (Gouveia and Freitas 2008, Joffre et al. 1999). Also, Waroux
and Lambin (2012) studied the degradation of the argania woodlands in semi-arid to arid Southwest
Morocco and found that a 44% decline of the forest density was mostly driven by the increasing aridity in
the region between 1970 and 2007. Similarly, many studies carried out on oak woodlands in Italy and
Spain identified drought as the main driving factor of tree die-back and as the main climate warning
threatening oak stands sustainability in the Mediterranean basin (Gentilesca et al. 2017). Tree mortality
linked to increasing drought stresses can also be associated to a geographical shift in vegetation
communities (Lloret et al., 2004). For example, xeric plant species Sahel have expanded in the north of
Sahel since the last half of the 20th century, toward areas of higher rainfall at an average rate of 500 to
600 m yr$^{-1}$ (Gonçalez P., 2001).

In environments with scarce water sources such as the Mediterranean basin, plants have developed
strategies to either avoid or escape drought stress (Chaves et al., 2003). The development of a dimorphic
root systems in woody species is an adaptation strategy to escape drougth (Dinis 2014, David et al.,
2013). When comparing different water limited ecosystems from a global dataset, Schenk and Jackson (2002) showed that rooting depth increased with aridity. Furthermore, a clear relationship between rooting depth and the water table depth was evidenced at global scale (Fan et al. 2017).

[revised manuscript text omitted]

that groundwater depth was only accounting for about 6% of the coefficient variation in the studied area,
against 89% of the variation represented by climate indexes AI and Ios4. Changes in climate conditions
only represents part of the water resources shortage issue in the future. Global-scale changes in human
populations and economic progresses also rules water demand and supply, especially in arid and semi-
arid regions (Vörösmarty et al., 2000). A decrease in useful water resources for human supply can induce
an even higher pressure on groundwater resources (Döll, 2009), aggravating the water table drawdown
caused by climate change (Ertürk et al., 2014). Therefore, additional updates of the model should include
human consumption of groundwater resources, identifying areas of higher population density or intensive
farming. Future model updates should also account for the interaction of deep rooting species with the
surrounding understory species. In particular, shrubs surviving the drought period, which can benefit from
the redistribution of groundwater by deep rooted species (Dawson, 1993; Zou et al., 2005).

**5 Conclusions**

Our results show a highly dominant contribution of water scarcity (Aridity and Ombrothermic indexes) on the density and suitability of deep-rooted groundwater dependent species. The contribution of groundwater depth was much lower than we initially expected, accounting only for 6% of the total coefficient variation. This might be underestimated however, 
[revised manuscript text omitted]

Drainage density; Thickness and Soil type refer to soil properties.

Figure B1 – Predictors maps after score classification. (a) – Aridity Index (AI); (b) – Ombrothermic Index of the
summer quarter and the immediately previous month (Ios4); (c) – Groundwater Depth (GWDepth); (d) – Drainage
density (Dd); (e) – Slope.

**Table 1: Environmental variables for the characterization of the suitability of GDV in the study area.**

| Variable code | Variable type | Source | Resolution and Spatial extent |
|---|---|---|---|
| Slope | Slope (%) | This work | 0.000256 degrees (25m) raster resolution |
| Soil type | Soil type in the first soil layer | SNIAmb (© Agência Portuguesa do Ambiente, I.P., 2017) | Converted from vectorial to 0.000256 degrees (25m) resolution raster |
| Thickness | Soil thickness (cm) | EPIC WebGIS Portugal (Barata et al., 2015) | Converted from vectorial to 0.000256 degrees (25m) resolution raster |
| GWDepth | Depth to groundwater (m) | This work | 0.000256 degrees (25m) raster resolution |
| Dd | Drainage Density | This work | 0.000256 degrees (25m) raster resolution |
| Spei_severe | Number of months with severe SPEI | This work | 0.000256 degrees (25m) raster resolution Time coverage 1950-2010 |
| SPEI_extreme | Number of months with extreme SPEI | This work | 0.000256 degrees (25m) raster resolution Time coverage 1950-2010 |
| AI | Aridity Index | This work | 0.000256 degrees (25m) raster resolution Time coverage 1950-2010 |
| Io | Annual Ombrothermic Index Annual average (January to December) | This work | 0.000256 degrees (25m) raster resolution Time coverage 1950-2010 |
| Ios1 | Ombrothermic Index of the hottest month of the summer quarter (J, J and A) | This work | 0.000256 degrees (25m) raster resolution Time coverage 1950-2010 |
| Ios3 | Ombrothermic Index of the summer quarter (J, J and A) | This work | 0.000256 degrees (25m) raster resolution Time coverage 1950-2010 |
| Ios4 | Ombrothermic Index of the summer quarter and the immediately previous month (M, J, J and A) | This work | 0.000256 degrees (25m) raster resolution Time coverage 1950-2010 |

**Table 2: Effect of variable removal in the performance of GWR model linking the Kernel density of *Quercus***
***suber*, *Quercus ilex* and *Pinus pinea* to predictors Aridity Index (AI); Ombrothermic Index of the summer**
**quarter and the immediately previous month (Ios4); Groundwater Depth (GWDepth); Drainage density (Dd);**
**Slope; and Soil type. The model with all predictors is highlighted in grey and the final model used in this study**
**is in bold.**

| Type | Model | Discarded predictor | AICc | Quasi-global R² |
|------|-------|---------------------|------|-----------------|
| GWR | Density~ios4 +ai + slope + Dd + GWDepth + soiltype | | 27389.74 | 0.926481 |
| GWR | Density~ios4 + slope + Dd + GWDepth + soiltype | Ai | 28695.14 | 0.9085754 |
| GWR | Density~ai + slope + Dd + GWDepth + soiltype | Ios4 | 28626.88 | 0.9095033 |
| GWR | Density~ios4 +ai + GWDepth + slope + soiltype | Dd | 27909.86 | 0.9184337 |
| GWR | Density~ios4 +ai + Dd + GWDepth + soiltype | Slope | 27429.55 | 0.924176 |
| GWR | Density~ios4 +ai + Dd + slope+ soiltype | GWDepth | 27742.67 | 0.9208344 |
| GWR | **Density~ios4 +ai + Dd + GWDepth + slope** | **Soiltype 3 levels** | **18050.76** | **0.9916192** |

**Table 3: Comparison of Adjusted R-squared and second-order Akaike Information Criterion (AICc) between**
**the simple linear regression and the GWR model.**

| Model | R-squared | AICc | p-value |
|-------|-----------|------|---------|
| OLS | 0.02 | 42720 | <0.001 |
| GWR | 0.99 * | 18851 | - |

*Quasi-global R$^2$

**Table 4: Classification scores for each predictor. A score of 3 was given to highly suitable areas and 1 to highly**
**less suitable areas for GDV.**

| Predictor | Class | Score |
|---|---|---|
| Slope | 0%-5% | 1 |
| | 5%-10% | 2 |
| | >10% | 3 |
| | | |
| Groundwater Depth | >15 m | 1 |
| | 1.5m-15m | 3 |
| | ≤1.5m | 1 |
| Aridity Index | 0.6-0.68 | 1 |
| | 0.68-0.75 | 2 |
| | ≥0.75 | 3 |
| Ios4 | <0.28 | 1 |
| | 0.28-0.64 | 2 |
| | ≥0.64 | 3 |
| Dd | ≤0.5 | 3 |
| | >0.5 | 1 |

[Figure]

**Figure 01: Study area. On the left the location of Alentejo in the Iberian Peninsula; on the right, the elevation characterization of the study area with the main river courses from Tagus, Sado and Guadiana basins. Names of the main rivers are indicated near to their location in the map.**

[Figure]

**Figure 02: Large well and piezometer data points used for groundwater depth calculation. Squares represent piezometers data points and triangle represent large well data points.**

[Figure]

**Figure 03: Map of Kernel Density weighted by cover percentage of** *Q. suber, Q. ilex* **and** *P. pinea***.**

[Figure]

**Figure 04: Map of environmental layers used in model fitting. (a) – Soil type; (b) – Slope; (c) – Groundwater**
**Depth (Depth); (d) – Ombrothermic Index of the summer quarter and the immediately previous month (Ios4);**
**(e) – Aridity Index (AI).**

[Figure]

**Figure 05: Spatial distribution of local $R^2$ from the fitting of the Geographically Weighted Regression.**

[Figure]

**Figure 06: Spatial distribution of model residuals from the fitting of the Geographically Weighted Regression**
**(a) and Simple Linear model(b).**

[Figure]

**Figure 07: Map of local model coefficients for each variable. (a) – Aridity Index; (b) - Ombrothermic Index of**
**the summer quarter and the immediately previous month (Ios4); (c) – Groundwater Depth (GWDepth); (d) –**
**Drainage density and (e) – Slope.**

[Figure]

**Figure 08 – Boxplot of GWR model coefficient values for each predictor. AI is Aridity Index; Ios4 is the**
**ombrothemic index of the hottest month of the summer quarter and the immediately previous month; GWDepth**
**is Groundwater Depth and Dd is drainage density.**

[Figure]

**Figure 09: Suitability map for Groundwater Dependent Vegetation.**

[Figure]

**Figure 10: Validation map corresponding to the NDWI anomaly considering the months of June, July and**
**August of the extremely dry year of 2005 in the Alentejo area. Brown colors (corresponding to more negative**
**values) indicate vegetation in water stress.**

**Appendix A**

**Table A1: Classification scores for the soil type predictor.**

| Predictor | Class | Score |
|---|---|---|
| Soil type | Eutric Cambisols; Dystric Regosol; Humic Cambisols; Haplic Luvisols; Gleyic Luvisols; Ferric Luvisols; Chromic Luvisols associated with Haplic Luvisols; Ortic Podzols | 3 |
| | Calcaric Cambisols; Dystric Regosol associated with Umbric Leptosols; Eutric Regosols; Vertic Luvisols; Eutric Planosols; Cambic Arenosols | 2 |
| | Chromic Cambisols; Eutric fluvisols; Chromic Luvisols; Gleyic Solonchak; Eutric Vertisols | 1 |

[Figure]

**Figure A1: Boxplot of the main predictors for the final Geographically Weighted Regression model fitting**
**(top) and the response variable (below), for the total data (left) and for the 5% subsample (right).**

[Figure]

**Figure A2: Correlation plot between all environmental variables expected to affect the presence of the Groundwater Dependent Vegetation. Ios1, Ios3, Ios4 are ombrothemic indices of, respectively, the hottest month of the summer quarter, the summer quarter and the summer quarter and the immediately previous month; Io is the annual ombrothermic index, Spei_extreme and Spei_severe are, respectively, the number of months with extreme and severe Standardized Precipitation Evapotranspiration Index; AI is Aridity Index; GWDepth is Groundwater Depth, ; Dd is the Drainage density; Thickness and Soiltype refer to soil properties.**

**Table A2: Correlations between predictor variables and principal component axis. The most important predictors for each axis (when squared correlation is above 0.3) are showed in bold. The cumulative proportion of variance explained by each principal component axis is shown at the bottom of the table**

| | PC1 | PC2 | PC3 | PC4 | PC5 | PC6 | PC7 | PC8 | PC9 | PC10 | PC11 | PC12 |
|---|---|---|---|---|---|---|---|---|---|---|---|---|
| **Slope** | <0.001 | **0.32** | 0.13 | 0.06 | 0.14 | 0.18 | 0.18 | <0.001 | 0.03 | 0.03 | <0.01 | <0.01 |
| **AI** | **0.94** | <0.001 | 0.01 | <0.01 | <0.001 | <0.01 | <0.001 | <0.001 | 0.22 | **0.33** | **0.40** | **0.68** |
| **Io** | **0.93** | <0.01 | 0.01 | <0.01 | <0.001 | <0.01 | <0.001 | <0.001 | 0.24 | **0.38** | 0.24 | **0.72** |
| **Ios1** | **0.89** | 0.02 | 0.04 | 0.01 | <0.001 | <0.001 | <0.001 | 0.02 | 0.03 | 0.14 | **0.82** | 0.10 |
| **Ios3** | 0.21 | 0.18 | **0.47** | <0.01 | <0.01 | <0.001 | <0.01 | 0.11 | **0.64** | **0.33** | <0.01 | <0.01 |
| **Ios4** | 0.15 | 0.19 | **0.53** | <0.001 | <0.001 | <0.01 | <0.001 | **0.33** | **0.53** | **0.33** | 0.05 | <0.01 |
| **Spei_severe** | **0.66** | 0.08 | 0.01 | <0.01 | <0.001 | -0.02 | <0.01 | **0.77** | 0.08 | **0.40** | 0.11 | 0.01 |
| **Spei_extreme** | **0.72** | 0.01 | 0.04 | 0.05 | <0.01 | <0.001 | <0.01 | **0.36** | **0.44** | 0.57 | 0.29 | 0.05 |
| **GWDepth** | 0.16 | 0.05 | 0.01 | **0.33** | 0.14 | 0.26 | 0.06 | 0.06 | 0.04 | 0.06 | 0.04 | 0.01 |
| **Dd** | <0.01 | 0.25 | 0.11 | 0.20 | 0.08 | **0.32** | <0.01 | 0.29 | 0.06 | 0.04 | <0.01 | <0.01 |
| **Soil type** | 0.02 | 0.19 | 0.03 | 0.22 | **0.46** | 0.05 | 0.02 | 0.06 | 0.03 | 0.05 | 0.03 | <0.01 |
| **Thickness** | 0.02 | **0.46** | 0.09 | 0.03 | 0.06 | 0.01 | **0.32** | 0.11 | 0.03 | 0.09 | 0.01 | <0.01 |
| **Cumulative proportion** | 0.39 | 0.54 | 0.66 | 0.74 | 0.81 | 0.88 | 0.93 | 0.96 | 0.98 | 0.99 | 0.99 | 1 |

**Appendix B**

[Figure]

**Figure B1 – Predictors maps after score reclassification. (a) – Aridity Index (AI); (b) – Ombrothermic Index of**
**the summer quarter and the immediately previous month (Ios4); (c) – Groundwater Depth (GWDepth); (d) –**
**Drainage density (Dd); (e) – Slope.**

---

## Author Response (AR1)

Instituto Dom Luiz, Faculty of Sciences

University of Lisbon

Campo Grande, 1749-016 Lisbon, Portugal

Tel: +351 927464067

E-mail: inesgmarques@fc.ul.pt

January 18th, 2019

Editorial Department of *Hydrology and Earth System Sciences*

Dear Dr. Miriam Coenders-Gerrits,

Please find enclosed the revised version of the manuscript (reference hess-2018-208) entitled "*Mapping the suitability of groundwater dependent vegetation in a semi-arid Mediterranean area*".

All the reviewer and editor suggestions were carefully considered and addressed accordingly. In the response to reviewers you will find the responses to all comments and all changes made, point by point, as suggested by the reviewers. In the present cover letter, you will find the response to the editor comments, as well as all changes made.

We are very thankful for all the comments, which allowed an improvement of the manuscript quality.

To facilitate the identification of changes along the manuscript, a version of the manuscript with tracked changes was uploaded in the journal platform.

We kindly ask the editor to add another institutional e-mail correspondence of the first author. This e-mail has already been added in the submitted manuscript with tracked changes and in the final manuscript.

All the information included in this manuscript is completely original and has been approved by all authors. The authors declare no conflict of interest. This manuscript has not been published previously or concurrently submitted for publication elsewhere.

Thank you for considering this revised manuscript for publication. Please do not hesitate to contact me if you require further details.

With our best regards, sincerely,

Inês Gomes Marques (on behalf of all authors)

**Editor Decision:** Reconsider after major revisions (further review by editor and referees) (07 Dec 2018) by Miriam Coenders-Gerrits

**Comments to the Author:**

Dear authors,
As can be seen by the comments of the 2 reviewers they are rather positive. In your reply you addressed correctly to the comments and proposed some major improvements. Nonetheless, I think the paper can be improved by making it less case specific. How general are the results for other (semi)-arid regions?

*Answer: Due to the similar climatic conditions and physiological responses of other semi-arid regions, the approach presented in this study can be applied as a first approximation to model the phreatophyte species. However, the model coefficients are highly and spatially variable and specific, which reinforces the need for model calibration on other regions, even though presenting the same climatic conditions. This was explained in lines 611-618, 657-660 of the discussion and 695-699 of the conclusion, and the possible applications, based on validations results, are presented in lines 706-713.*

The applied regression model is highly sensitive for the input as shown by your own correction to remove soil type from the analysis. Hence a proper sensitivity analysis plus a more elaborated discussion on the limitations of regression model would benefit the manuscript.

*Answer: As suggested by the editor, a sensitivity analysis was performed to the model outputs. The applied methodology was explained in the chapter 2.8, in lines 377-390. The respective results were presented in the chapter 3.6 (in lines 523-535).*

Furthermore, I had a minor comment on the use of symbols. I highly recommend to use single characters. So not Dd as in equation 1, but D_d (subscript). Otherwise Dd could be confused with D*d. Please check this throughout the entire manuscript. Related to this, it's also better to not use words in equations. So in the case of equation 4, please define symbols for density, depth, soil type, etc.

*Answer: All symbols were changes according to the editor suggestion. Drainage density symbol is now D, Aridity Index is $A_i$, Ombrothermic index is $O_4$, Groundwater depth is W and Slope is s. Equation 6 (lines 436-437) was changed accordingly, as well as all figures, tables and in the text.*

Dear Referee1,

Please find enclosed the revised version of the manuscript "Mapping the suitability of groundwater dependent vegetation in a semi-arid Mediterranean area".

All the suggestions were carefully considered and addressed accordingly. In the present letter, you will find the responses to all comments and all changes made, point by point. Particularly, we have clarified the comments on the methodology to calculate the map to water table and changes the validation method as suggested. As a result of the introduction of a new remote sensing for the suitability map validation we added the author Célia M. Gouveia to the authors list.

We are very thankful for all the comments, which allowed a very significant improvement of the manuscript quality.

To facilitate the identification of changes along the manuscript, a version of the manuscript with tracked changes was uploaded in the journal platform.

All the information included in this manuscript is completely original and has been approved by all authors. The authors declare no conflict of interest. This manuscript has not been published previously or concurrently submitted for publication elsewhere.

Thank you for considering this revised manuscript for publication. Please do not hesitate to contact us if you require further details.

Referee Comments 1

General Comments
Gomes Marques et al. present an analysis of the spatial distribution of groundwater dependent vegetation across the Iberian Peninsula. While the method used is perhaps not as novel as suggested in the text, the paper's main strength lies in the validation of the maps created against a fairly robust eternal dataset. The text is generally well written, although it is not as clear as it could be when discussing how the "model" was parameterized and validated. In general, the paper is a solid contribution to the literature on phreatophytes, but needs revision to enhance its clarity and address some lingering questions about the work.

Specific Comments
1. Throughout: What exactly is meant by a "suitability map"? Suitability for what? Or do you mean suitability of the terrain for hosting phreatophtyes? The concept is fine, but the word choice seems odd.

   *Answer: We appreciate the reviewer's comment on this matter. With a suitability map we aim to ascertain the suitability of the arboreous phreatophyte species to the climatic and local conditions. To clarify this matter, this information was provided in lines 122-123 of the introduction.*

2. Line 147 - 149: How heavily managed are these forestry systems? What species are harvested? And with what methods?

*Answer:* *In the Alentejo region, Cork oak, Holm oak and Stone pine represent 83% of the forest cover, covering about 36% of the geographical area. Cork oak covers 46% of the total forest area of the region, Holm oak 30%, and Stone pine only 7%, according to the last forest inventory. These species were already dominant species in the region and in Portugal two millennia after the beginning of holocene (Bugalho et al. 2009, Proença 2009). Since the 15th century, the agro-silvopastoral systems is largely dominant and steady in the province of Alentejo, on flat terrain. The system has a low tree density (40 to 80 trees/ha), trees being exploited for cork or seeds to feed cattle and the understory cleared of shrubs for pasture, crops (mainly wheat, barley and oats), or both. Tree density is determined by the need for space for pasture or cereal cultivation in the understory (Acacio & Holmgreen 2014) and by climate drivers, especially mean annual precipitation (Joffre 1999, Gouveia & Freitas 2008). Agro-silvopastoral systems are considered semi-natural ecosystems, which must be continually maintained through human management by thinning and understory use through grazing, ploughing and shrub clearing (Huntsinger and Bartolome 1992) to maintain a good productivity, biodiversity and ecosystem services. Cork oak trees are protected and cannot be harvested unless the tree has died, while holm oak trees are maintained with a low tree density (20 to 40 trees/ha) to guard against soil erosion and to provide shelter and shadow for cattle. Holm oaks are known to be more resilient to drought (David et al. 2007) and are mostly distributed in the most xeric area, on the oriental part of the Alentejo region. Some of this information has been added to the introduction section in lines 83-88 and in the discussion section in lines 547-550.*

**Bugalho M** *, Plieninger T, Aronson J , Ellatifi M, Gomes D Crespo 2009. Revista especializada. Cork oak woodlands on the edge. Ecology, adaptive management, and restoration, 1st edn. Society for Ecological Restoration International, Island Press, WashingtonChapter 3. Open Woodlands: A Diversity of Uses*
**Proença** *2009, Galicio-Portuguese oak forest of Quercus robur and Quercus pyrenaica: biodiversity patterns and forest response to fire. PhD Thesis,* *https://core.ac.uk/download/pdf/12421965.pdf*
**Acácio** *V. & Holmgreen M. 2009 Pathways for resilience in Mediterranean cork oak land use systems. Annals of Forest Science 71:5-13. DOI: 10.1007/s13595-012- 0197-0*
**Joffre** *R, Rambal S, Ratte PJ. 1999 The dehesa system of southern Spain and Portugal as a natural ecosystem mimic. Agrofor Syst 45:57–79*
**Gouveia** *A. & Freitas H., 2008 Intraspecific competition and water use efficiency in Quercus suber: evidence of an optimum tree density? Trees, 22 (2008), pp. 521-530*
**Huntsinger** *L, Bartolome JW. 1992 Ecological dynamics of Quercus dominated woodlands in California and southern Spain: a state transition model. Veg 99–100:299–305*
**David T.S.,** *Henriques M. O., Kurz-Besson C., Nunes J., Valente F., Vaz M., Pereira J. S., Siegwolf R., Chaves M.M., Gazarini L.C. and David J.S. Water use strategies in two co-occurring Mediterranean evergreen oaks: surviving the summer drought. Tree Physiology 27(6): 793-803* *https://doi.org/10.1093/treephys/27.6.793*

3. Line 170: Cite ASTER GDEM data in the manner requested on the NASA webpage (https://lpdaac.usgs.gov/citing_our_data).

*Answer: Done. The missing citation was added in the proper place, in line 171-174. The acknowledgement is already properly done in the acknowledgement section.*

4. Line 172: What is meant by superficial water use? Shallow groundwater? Surface water in streams? It's used several places but isn't well defined.

*Answer: We substituted the expression superficial water/groundwater by shallow soil water, which refers to the water between 0 and 1.5 m depth, in line 175. All water below 1.5 m depth was considered as groundwater. This was clarified in lines 306-308.*

5. Line 179: What were the three classes? How did soil parameters influence in classification?

*Answer: The three classes are presented in Table 4 and an additional explanation to clarify the conditions who led to each scoring was added to lines 182-186. After revision of the suitability model the predictor soil type was no longer included in model fitting, and therefore no further explanation on its effect on the suitability to GDV was added.*

6. Line 187-202, Figure 2: The location of piezometer data and well data are quite biased. What is this attributable to and how might it affect the results? It seems like the kriging in the south-central region could be quite problematic. Also, what is the distinction between a well and a piezometer here? This is also concerning because the most dense of the GDV species are roughly in this corridor as well.

*Answer: The region under study is an area with a very low population density, which reflected in the lack of points for piezometric level measurement, mainly in unconfined aquifers (~96% of the total area). Once the correlation between the piezometric level and the topography was successfully tested it was possible to estimate the piezometric level by kriging with external drift in areas where information was not enough. We added one reference to line 601 to support this methodology.*
*In the studied area, the presence of piezometers (exclusively dedicated structures for piezometric observations) is mostly associated with karst aquifers and areas with high abstraction volumes for public water supply. Oppositely, large wells are mainly devoted to private use and low volume abstractions. To complement the information given on the groundwater level estimation the following sentence was added to the ms, in lines 199-202: "In the studied area, piezometers are exclusively dedicated small diameter boreholes for piezometric observations, in areas with high abstraction volumes for public water supply. Large diameter wells in this region are usually low yielding and mainly devoted to private use and irrigation."*

7. Line 199 -202: I disagree with this method - the groundwater elevations should be determined by first determining the groundwater elevation at the piezometers and then interpolating that through kriging. This should introduce fewer errors and be more realistic.

*Answer: The relation between piezometric level and topography is quite high in most of the unconfined aquifers (Marcily, 1986). This relation allows to estimate the piezometry in areas with few piezometric points with enough confidence using external drift kriging. On the other hand, through this method the piezometric surface respects the orographic structures such as valleys, which is not the case with traditional interpolation methods.*

*"In the aquifer, the water flows toward the outlets, which are the low points in the topography (springs, streams in the surface flow network)." from Ghislain de Marsily, 1986. Quantitative Hydrogeology, Academic Press, Orlando. ISBN: 9780122089169, 9780080917634.*

8. Line 303 - 312: The rationale for this validation method is a bit shaky and could use more explanation. If the presence/absence of these trees is a good indicator, why is the rest of the analysis necessary? Is it more expansive? Precise? Also, how is this not a bit autoregressive, given that it sounds like kernal density was derived from the tree data. It starts to make more sense as the results are discussed, but it needs more clarity here. What about using a remote sensing method for validation instead (e.g., Munch et al. 2007, Barron et al. 2012, Gou et. al 2014)? How would that compare?

> *Answer: After consulting the authors of the EPIC suitability maps (Magalhães M. and Mesquita S.) we understood that the latter were indeed constructed based on the last forest inventories. Therefore, there was indeed autoregression in our validation (see Mesquita, S. and Capelo, J. (2016). Aptidão Bioclimática às Espécies Arbóreas. In Magalhães, M.R. coord): Ordem Ecológica e Desenvolvimento - o futuro do território português. Pp. 63-85. Centro de Estudos de Arquitectura Paisagista "Professor Caldeira Cabral". ISA Press. Lisboa. ISBN: 978-972-8669-64-5).*

> *We thus followed the reviewer suggestion to use remote sensing data to validate our GDV suitability map. We therefore compared our GDV suitability map with NDWI anomalies of June 2005 (extremely dry hydrological year in Portugal) with the median June NDWI of year 1999-2009 with a dataset shared by Gouveia et al. (2012). We chose the Normalized Difference Water Index (NDWI) for being more representative of water content in vegetation's leaves. This index is thus a proxy for vegetation stress, with low NDWI values representing less leave water content, corresponding to a higher drought stress. The NDWI map we present show in yellow and brown color the areas were the vegetation was more sensitive to the extreme drought of 2005. We obtain a very good agreement between maps that we commented in the results and discussion sections, in the section 3.5 – Map Validation, 634-639 and 641-644. The method and dataset used are described in the M&M section lines 349-374.*

9. Line 389-397: What soil types were the most likely to host phreatophtyes? What does "soil type 3" represent?

> *Answer: Soil type 3 represented soils with prevailing water storage at deeper soil depths, and therefore these soils were considered as more likely to host phreatophytes.*

10. Line 480-484: This paragraph seems to be saying that there must be some threshold by which no woody species can be supported, even if they are GDVs. These species get replaced by shortlived grasses and forbes, converting savanna to grassland. Is this correct? If so, this seems to contradict the next line about woody vegetation being replaced by shrublands. Wouldn't that presume shrubs are less susceptible to drought than trees? Please clarify.

> *Answer: The referenced paragraph was removed from the discussion, after the calculation of a new suitability map. Instead we discussed the strong relation between aridity and tree density and the degradation of ecosystems linked to increasing water scarcity, in lines 608-615.*

11. Line 495-511: This part of the discussion is problematic, because, as the authors note, the factor expected to be most key is poorly mapped. Regardless, they still say that soil type, as opposed to groundwater depth, is the most influential and claim that soil type defines the capacity for "groundwater storage". This appears to be overreach.

> *Answer: The most influential factors in the reviewed version of the manuscript are climate drivers. After removing soil types from the final GWR model, the contribution of*

*the W variable in the model improved (now corresponding to the 3rd most relevant variable in the model), but still remained far less relevant than climate drivers. The part of the discussion was slightly modified according to new results, lines 592-598.*

12. Figure 7: This figure needs more color variation. It is difficult to tell moderate, good, and very good apart.

*Answer: Done. The color scale was modified to improve readability in Figure 09.*

Technical Corrections
13. Line 88: Replace "genders" with genre

*Answer: Done in line 93.*

14. Line 102: Replace "5m" with "5 m". Noticed number/unit spacing issues in several other locations as well.

*Answer: Done in line 108. All other places where the same issue was found, were corrected.*

15. Lines 132 - 135: Replace "chapters" with "sections". But really, this whole paragraph isn't necessary, as the format doesn't deviate from standard expectations.

*Answer: Done. The paragraph was eliminated from the manuscript.*

16. Line 154: "Proxy for" not "proxy to".

*Answer: Done in line 154.*

17. Line 175: Is the copyright symbol here a typo?

*Answer: No, the copyright symbol is requested to reference the database.*

18. Line 201: Don't need to repeatedly cite Spatial Analyst and its version so frequently. Can this be converted to one mention at the beginning of the section?

*Answer: Done. We added the sentence "The software used in spatial analysis was ArcGIS® software version 10.4.1 by Esri and R program software version 3.4.2 (R Development Core Team, 2016)." to the ms, under the chapter 2.3 in lines 167-168. All mentions to R and ArcGIS software versions were removed from the text.*

19. Line 295: Put equation right after fist mention.

*Answer: We have added a general equation of the model (Eq. 4) in lines 337-338 and maintained the equation with the final predictors (Eq. 6) in lines 436-437 so that only in the results section we would present the final model equation used to calculate the suitability map.*

20. Line 306: Replace "to a" with "of a".

*Answer: This paragraph has been deleted from the ms, after the validation was performed with a different dataset.*

21. Line 434, Line 454: Delete first names of authors.

*Answer:* *The first reference has been removed from the paragraph. On lines 58, 303 and 563 however, the name "Condesso de Melo" was right, thus remained unchanged.*

22. Line 450: Pinpoint is one word.

   *Answer:* *Done in line 559.*

23. Lines 451-453: Awkward wording makes the sentence hard to parse.

   *Answer:* *The sentence was improved in lines 559-562.*

24. Line 466: Delete stray "s".

   *Answer:* *This paragraph was completely re-structured.*

Dear Referee2,

We are very grateful for your rigorous assessment and the valuable comments and suggestions you provided to improve our manuscript.

Please find enclosed the revised version of the manuscript "Mapping the suitability of groundwater dependent vegetation in a semi-arid Mediterranean area".

We believe that all your suggestions were carefully addressed. In the present letter, you will find our responses to each comments and changes made.

Particularly, we have corrected the methodology to calculate drainage density. We also clarified the error you detected regarding a mistake in the map of Figure 7 where for a low aridity index (AI) we predicted a high suitability value (Figure B1b), which you attributed to the negative sign assigned to AI used as weighting factor. We evaluated the impact of each predictor on the final model and discovered that soil type actually considerably worsened the performance of the GWR model. We therefore decided to remove it from the final model equation selected to build the suitability map.

We also attempted to provide a better evaluation of the importance of each predictor in the final model and improved the discussion section accordingly.

To facilitate the identification of changes along the manuscript, a version of the manuscript with tracked changes was uploaded in the journal platform. All the information included in this manuscript is completely original and has been approved by all authors. The authors declare no conflict of interest. This manuscript has not been published previously or concurrently submitted for publication elsewhere.

Also, we thank you for considering this revised manuscript for publication. Please do not hesitate to contact us for any further needed detail.

With our best regards, sincerely

Referee Comments 2

General comments

The current manuscript provides an interesting insight into the use of mapping and spatial regression to assess the occurrence of groundwater dependent vegetation (GDV). Such maps can subsequently be used to predict the effect of change in any of the explanatory variables, such as climate or groundwater depth, on the spatial distribution of GDV in an area. The paper is well written and structured, and is subdivided into two parts: the first on the building of regression models for predicting GDV occurrence based on actual data, and the second part where a parameter-based index is calculated to construct so-called "suitability maps" for GDV. While I find the first part strong and with high potential of publication by adding a scenario analysis, my main concern lies with the second part. In my opinion this part is less well developed and the interpretation of the results is largely straightforward, and to a certain degree incorrect. Interpretation is largely straightforward because of the large bias in weighting of the parameters, where the contribution of soil largely exceeds all other parameters, thus making it essentially a soil map. Additionally, interpretation is to a certain degree incorrect, due to an apparent mistake in the generation of the suitability map, where (inadvertently) a negative weighting was assigned to the aridity index, resulting in the inverse impact of this parameter on the soil map. I would therefore recommend focusing and further elaborating on the regression modelling, as I will specify in my detailed comments below.

Specific comments

Abstract

Line 13-19: The first part of the abstract is more of an introduction. I suggest starting with what was actually done (line 20). Moreover, groundwater depletion will not occur merely as a result of climate change. Finally, as groundwater level seems to have such a low impact in the regression model, the question rises to what extent groundwater depletion will play a role in the spatial distribution of GDV.

> *Answer: The first paragraph of the abstract was rearranged. The abstract was corrected according to new results.*

Line 48: When referring to climate change impact studies on recharge in Mediterranean areas add the paper of doi.org/10.1007/s10113-012-0377-3, where such a study is being reported.

> *Answer: Done. The suggested reference has been added in the text lines 49-50 and*
reference list lines 1083-1086.

PART 1: REGRESSION MODELLING

Parameter selection for the regression model

Soil type: the authors only use the first layer of the soil. To understand the importance of capillary rise feeding into the root zone, the texture of deeper soils also needs to be considered. The latter could further affect the role of groundwater depth in the model, as fine soils have a much higher capillary (and water-holding) capacity. In the model, soil type is subdivided into two sub-parameters (2 and 3, Equation 4 and Table 2), but this is not further explained. Evidently, this increases the weight of soil type in the regression model.

> *Answer: The classification of soils into 3 categories was explained in lines 182-186 of the M&M section and in Table A1 in appendix A. This predictor was removed from the model fitting after revision from the authors. It has not been possible to add the texture of deeper horizon into our study because such information was only available on inaccessible printed maps. Unfortunately, no such digital data were available when the manuscript was prepared or revised.*

Groundwater depth: please comment on the reliability of the results in the empty areas (areas without wells or piezometers). Can wells and piezometers be used together, in other words, are all wells installed in unconfined aquifers?

> *Answer: Please notice the previous answer to the reviewer 1.*
> *The region under study is an area with a very low population density, which reflected in the lack of points for piezometric level measurement, mainly in unconfined aquifers (~96% of the total area). Once the correlation between the piezometric level and the topography was successfully tested it was possible to estimate the piezometric level by kriging with external drift in areas where information was not enough. We added one reference to line 601 to support this methodology.*

*The estimation of the groundwater depth did not consider the simultaneous use of large wells and piezometers, with exception of the northwestern area, due to the lack of large wells.*

*In the studied area, the presence of piezometers (exclusively dedicated structures for piezometric observations) is mostly associated with karst aquifers and areas with high abstraction volumes for public water supply. Oppositely, large wells are mainly devoted to private use and irrigation.*

Drainage density: Drainage density was calculated for six river basins. That gives little variation across the area. Is it possible to map drainage density at a higher resolution, e.g. sub-basin scale, or a 10 km grid size? This would increase the importance of this parameter.

*Answer: Indeed, there was little spatial variation of the drainage density for the studied area, therefore, as suggested by the reviewer, we recalculated this variable considering a 10km resolution grid. The methodology concerning this calculation was corrected in the ms, in lines 212-213. Due to the creation of a new drainage density map, we performed a reassessment of the multicollinearity between variables and the selection of predictors (see section 2.4). This implied recalculating Pearson's coefficients and Principal Components Analysis (PCA), presented in table A2 and figure A2 both in appendix A. It also affected predictors and coefficients in the model linking GDV density to environmental predictors). By affecting model development, model performance (Tables 2 and 3), suitability and coefficient maps were also affected (Figures 7 to 9).*

Climate: The authors should provide a bit more explanation on the SPEI and particularly the ombrothermic index calculations. Please explain how/where the latter differs significantly from (and is thus not correlated to) the aridity index.

*Answer: Done. Clear explanations on SPEI calculations were already provided in lines 225-228. Since the SPEI predictor was excluded from modeling further explanation would unnecessarily extend the manuscript length. We however briefly altered paragraph lines 233-235, to better explain the discrepancies between SPEI and Ombrothermic indices, and to clarify the Ombrothermic indices calculation according to Table 1.*

Model development

It is not clear how the parameters were normalized before entering the regression model.

*Answer: The explanation of the normalization based on the z-score function was improved and changed to the M&M section, under the chapter 2.5 - Model development. Variables were standardized before entering the regression model through the calculation of a z-score. To clarify how the standardization was done the following sentence was added to lines 274-276 of the ms: "This allows to create standardized scores for each variable, by subtracting the mean of all data points from each individual data point, then divide those points by the standard deviation of all points, so that the mean of each z-predictor is zero and the deviation is 1.".*

How was the soil parameter transformed into a quantitative variable? If all parameters were classified/categorized (as is often done in e.g. factorial regression analysis), this can explain the low influence of the groundwater depth parameter, as there is very little variation (in large part of the area groundwater depth is between 1.5 and 15 m). In this case, I strongly suggest increasing the number of classes for groundwater depth.

*Answer: We greatly appreciate your comment. First, we would like to clarify that the main purpose of the model construction is to attribute coefficients of importance to each variable, so that these coefficients can be applied to classification scores given to each variable by expert judgement (table 4) and return a suitability map to groundwater dependent vegetation. This will allow the production of a suitability map where the coefficients of importance applied to each variable were calculated empirically. Therefore, the classification scores given to each variable were not applied in the model calculation, but rather after the local model coefficients were calculated (as a mean to construct the suitability map).*

*The soil parameter was used has a numeric categorical variable (with the values given initially from 1 to 3), through the use of the function as.factor() in R. The usage of this function will insure that the factor is seen as nominal and not as ordinal. Because the remaining variables showed continuous values, only the soil type variable was categorized, and the remaining variables used to run the model were continuous. The scoring applied is presented is Table A1and the explanation in lines 182-186 of the methods section.*

*The reviewer is correct about the groundwater depth variable and its very low variation above 15 m. As further explained below, it has not been possible to increase the number of classes for GWDepth, for the weighting factors to be correctly applied to the GWDepth layer in the multicriteria analysis. To overcome this situation the values of water depth above 15 m were replaced by a value above 15m (15.1 m), in order to emphasize the variation observed between 0 and 15 m depth, which matters the most to GDV. These values were only used for the model fitting. The species used as proxies for groundwater dependent vegetation are less probable to use water at depths lower than 15m, and so all the range of values above this threshold would be considered as inaccessible by those species.*

Please provide references showing that it is common practice to fit the model on a 5% random subsample. Also explain more lines 264-265.

*Answer: The sub-sampling size was mostly dictated by computing limitation in the sense that the random subsample size was decreased down to 5% until the GWR model could be fitted. The mean distance between neighbor points using 5% of the original dataset was about 6 km, with a maximum distance of 15km. Nevertheless, we could find a few studies using a 10% random sub-sample of the data corresponding to a 10km resolution grid to perform GWR modeling (Bertrand R., 2017), as well as linear regression (Bertrand et al. 2016). The authors were using such subsampling to restrain autocorrelation issues according to Kühn (2007). We modified our text to include those references as well as the benefit for autocorrelation issues in our study, lines 281-287 and in the reference list, lines 785-787 and 946-947*

*In addition, we calculated basic statistic indicators for the totality of the data and compared with the random subsample. Results are presented in line 284-286 of the ms and in Figure A1 in appendix A.*

*Bertrand R. Unequal contributions of species' persistence and migration on plant communities' response to climate warming throughout forests. bioRxiv, doi.org/10.1101/217497, 2017.*
*Bertrand R., Riofrío-Dillon G., Lenoir J., Drapier J., de Ruffray P., Gégout J.C., Loreau M.. Ecological constraints increase the climatic debt in forests. Nature Communications 7, 12643, doi: 10.1038/ncomms12643, 2016.*

*Kühn, I. Incorporating spatial autocorrelation may invert observed patterns. Divers. Distrib. 13, 66–69 (2007).*

Results

Overall section 3.2 on environmental conditions mainly consists of an explanation of each of the maps. To support the selection of the five parameters, the authors should provide all the results on correlation and PCA as supplementary material.

> ***Answer:*** *Done. PCA results have now been provided as supplementary material in Table A2 in appendix A and was modified according to new results, due to the construction of a new map of drainage density.*

The results of the model suggest, as stated by the authors, a low importance of the groundwater depth on explaining the spatial distribution of GDV (eq. 4). However, nothing is said on how this varies locally within in the area. Are there regions where the role of groundwater is larger? Can these regions be identified?

> ***Answer:*** *We have plotted the local coefficients of all predictors and present it in Figures 07 and 08. In addition, we added some paragraphs with an explanation of the spatial variation of each predictor in the results section, in lines 455-461 and in the discussion section in lines 578-583.*

Line 343-344: This requires quite a bit more explanation, but can be easier to follow once the calculation of Ios4 has been better explained in the methods section.

> ***Answer:*** *We altered lines 233-235 and 243-245 to better explain the discrepancies between SPEI and O and to clarify the O calculation according to Table 1.*

Line 362-364: Please elaborate on this outcome on the Moran index.

> ***Answer:*** *Bibliographic references for the Moran Index were added in line 446 and the respective references were added to the bibliography. In addition, we extended the results explanation on the Moran Index and the z-score, in lines 446-450.*

In eq. 4 the appearance of Soil type 2 and Soil type 3 is not explained.

> ***Answer:*** *After revising the methodology and predictors selection, the predictor soil type was no longer included in the model.*

The results would become more interesting with:
1) a more local/regional analysis of the explanatory model and the importance of each of the parameters (in particular groundwater depth);

> ***Answer:*** *The model equation was substituted by a local one including the proportion of the local coefficients from the total variability of all the coefficients for each local GWR model. Local relative coefficients were considered as weighting factors instead of median values (please see revised Equation 6 in the manuscript). We also added a figure corresponding to the local variation of each coefficient in Figure 07 and commented the variations in the result section, in lines 455-461 and in the discussion section, lines 578-583. The relative importance of each variable in the final model is now shown in Figure 08, representing the distribution of the local coefficient values in a box plot.*

2) an assessment of the use of more/less/different parameters on the final model. It seems the soil type and to a lesser extent the aridity index are the dominating parameters in the regression model. How does a model based solely on these two parameters perform? And what about including a deeper (2nd-3rd layer) soil parameter to account for water holding and capillary rise capacity? Not much can be stated on the importance of soil type for the groundwater storage (as mentioned in line 495) if only the first soil layer is assessed.

*Answer: We tested the effect of removing one of the variables on the model performance and found out that the model performance increased notably when soil types was removed (AIC divided by factor 2, Table 2). The removal of any other variable however, did not seem to impact the model performance as compared to the equation including all formerly selected variables. Therefore, we excluded soil types from the final GWR model and the rest of our analyses and multicriteria analysis. Data on deeper soil parameters was not available for the study area and therefore that information could not be included in the model.*

3) scenario analysis: what happens if one or more of the parameters (such as climate or groundwater level) change? You do not have to develop climate scenarios, but an assessment of the impact of a relative change in aridity index or groundwater level on the resulting map would be of high added value.

*Answer: The development of climate scenarios or the assessment of the impact of aridity change on GDV suitability was out of the scope of this manuscript, since the actual manuscript is quite long already. The full assessment of climate changes impacts and corresponding uncertainties will be the focus of our next publication.*

*We calculated preliminary results of the relative change of $A_i$ and $O_4$ expected for the near future (Table a and b below). Our ongoing calculations based on scenarios RCP 4.5 and 8.5 show that $A_i$ and $O_4$ climate indexes are going to decrease in the studied region (-14 to -33% within 2099), drifting from a mostly dry sub-humid climate ($0.5<A_i<0.65$) to a mostly semi-arid one ($0.2<A_i<0.5$) by 2099 in scenario RCP8.5, and according to the classification of Middleton et al. (1992). Ios4 is also going to suffer a huge drop (-42 to -58% within 2099). Also, while most of the territory could be considered as non-Mediterranean based on the ombrothermic index ($O_4>2$) during the historical period 1971-2000, it is becoming mostly Mediterranean by 2099 in scenario RCP8.5 and according to the classification of Rivas-Martínez et al. (2011).*

*To include such preliminary results in the M&M, result and discussion regarding climate change impact would imply to considerably increase the manuscript, while providing an incomplete picture of the changes and associated uncertainties. We therefore chose not to include the suggested assessment of the impact of a relative change in aridity index or groundwater level on the resulting map in this manuscript.*

*Nevertheless, we discussed the relative importance of each predictor in our final map, which give an insight of how the groundwater dependent- vegetation is expected to be affected according to the predicted increased aridity, lines 676-684 and 708-713.*

*Table a. Mean relative changes expected for AI and IOS4 in the near future according to climate changes scenarios RCP 4.5 and 8.5, and respective standard deviations. Changes were computed considering 30 yr means obtained from an ENSEMBLE of eleven EU-CORDEX climate models.*

| | Mean | Stdev |
|---|---|---|
| AI_rcp45_2011-2040 | -8.1 | 1.5 |
| AI_rcp45_2041-2070 | -16.0 | 1.5 |
| AI_rcp45_2071-2099 | -14.5 | 1.7 |
| AI_rcp85_2011-2040 | -12.4 | 1.8 |
| AI_rcp85_2041-2070 | -22.1 | 2.0 |
| AI_rcp85_2071-2099 | -33.0 | 2.1 |
| IOS4_rcp45_2011-2040 | -21.2 | 2.3 |
| IOS4_rcp45_2041-2070 | -37.8 | 1.3 |
| IOS4_rcp45_2071-2099 | -42.1 | 2.4 |
| IOS4_rcp85_2011-2040 | -29.7 | 2.4 |
| IOS4_rcp85_2041-2070 | -44.4 | 1.6 |
| IOS4_rcp85_2071-2099 | -57.8 | 1.3 |

*Table b.* *Evolution of percentiles 10 and 90 values of AI and IOS4 in Alentejo from the present to the near future according to scenarios RCP 4.5 and 8.5. ECAD are observed values for the reference period 1971-2000. Historical values for the reference period 1971-2000 as well as predicted values for the future were simulated by an Ensemble of EU-Cordex models.*

| | AI | | IOS4 | |
|---|---|---|---|---|
| | P10 | P90 | P10 | P90 |
| AI_ECAD_1971-2000 | 0.37 | 0.53 | 2.29 | 4.06 |
| AI_historical_1971-2000 | 0.37 | 0.66 | 2.46 | 4.10 |
| AI_rcp45_2011-2040 | 0.34 | 0.60 | 1.88 | 3.28 |
| AI_rcp45_2041-2070 | 0.31 | 0.55 | 1.53 | 2.58 |
| AI_rcp45_2071-2099 | 0.31 | 0.55 | 1.39 | 2.42 |
| AI_rcp85_2011-2040 | 0.32 | 0.58 | 1.66 | 2.97 |
| AI_rcp85_2041-2070 | 0.28 | 0.51 | 1.39 | 2.24 |
| AI_rcp85_2071-2099 | 0.25 | 0.44 | 1.05 | 1.70 |

Discussion
Much of the discussion on the modelling approach is more of a summary of the manuscript, particularly lines 425-439. I miss the interpretation of the results obtained by regression modelling, and the this could further be enriched by the discussion of the added results as proposed above.

> *Answer: The discussion section has been considerably modified. The dominant impact of aridity on tree density and GDV suitability is now much more discussed, as well as the lower impact of groundwater depth. The relative weight of each predictor is also discussed and considered in the key limitations and conclusions sections. (see mostly lines 586-591, 676-681 and 693-702).*

PART 2: SUITABILITY MAPPING
Suitability map building
The authors decide to attribute the minimum score (in terms of suitability) to areas where groundwater depth is smaller than 1.5 m, considering that vegetation extracting water from shallow depths belongs to another type of GDV. This distinction between shallow and deep groundwater dependent vegetation, which I indeed think is useful (as most vegetation can use water in the first 1.5 m if present) needs to be briefly elaborated upon.

> *Answer: Providing a less probable score to host the GDV to the 0-1.5m GWDepth was made to exclude riparian vegetation and shrubby species which primarily use the water from streams and the superficial soil layer. An additional explanation and references were added to the manuscript in lines 310-314: "The depth class between 0 and 1.5m was based on the riparian vegetation in semi-arid Mediterranean areas which is mainly composed of shrub communities (Salinas et al., 2000) and present a mean rooting depths between 1 and 2m (Schenk and Jackson, 2002). The most common tree species rooting depth in riparian ecosystems is normally similar to the depth of fine sediment not reaching gravel substrates (Singer et al., 2012), but not reaching levels as deep as deep-rooted species.".*

Line 284-286. I do not understand why shallow groundwater flow would be expected at steep slopes. Normally steeper slopes are found in mountainous areas, where groundwater levels are deep.

> *Answer: The reviewer is correct, and we appreciate for noticing the error. The sentence was corrected in lines 316-317 and the term water flow was substituted by runoff.*

Results
The main finding here is that "suitability to GDV in the Alentejo region was mainly driven by soil type". That is obvious, as the weight of this parameter is by far the largest in the suitability index (and given by two soil type variables)! The same holds for the observation "The aridity index also showed a strong influence on GDV's suitability", as the weight of the aridity index is highest following that of soil type. I would strongly suggest analysing alternative weights for each parameter (based for instance on the Delphi panel) and evaluating the corresponding sensitivity of the outcome, as well as the degree of success in the validation procedure.

> *Answer: Unfortunately, it has not been possible for us to perform this analysis within the time provided to review our manuscript. We hope that the discussion on the relative importance of each predictor in the model will be satisfactory enough for the reviewer, considering that every other request was fulfilled.*

Line 395-396: "high aridity values restricted GDV's suitability in the south". Again, in my view it is exactly the opposite, as a high aridity is classified as class 3, i.e. of high suitability. In the south in fact aridity index is lowest, indicating the highest aridity and therefore higher suitability for GDV.
I think I might have detected a mistake in the resulting map of Figure 7. Where aridity index (AI) values are low, corresponding suitability value is high (Figure B1b), which means that overall suitability should also increase in those areas (towards the southeast). In the map of Figure 7 the values actually decrease in that area, which is contrary to what would be expected and could result from a negative weight being assigned to this parameter (as it also has a negative coefficient in the regression equation). If this is the case, the presentation and interpretation of the results on suitability mapping needs to be redone.

> *Answer: After thoroughly verifying the model calculations (Eq. 6) and the weighting factors used for the final multicriteria analysis (Figure 09), we must agree with the reviewer that it was a mistake to apply a negative weighting to the Aridity Index layer. Indeed, where real values of $A_i$ were low (indicating a more arid area), our scoring was high in the multicriteria analysis. To directly apply a negative weight, we should have the real predictor values and the predictor scores co-varying (or growing) accordingly. We*

*also verified that the same logic should be applied to the other quantitative variables Slope, $O_4$ and Groundwater Depth, since scoring and real values variation were opposite.*

*However, in the revised manuscript we have adopted different scores for the Aridity Index (scores 1, 3, 2) which were not varying linearly, and it was no longer possible to apply a linear scoring. The same was applicable in the case of Groundwater Depth, when we came to a dead end because scoring was not varying linearly according to class values (scores 1, 3, 1). As a solution we calculated the proportion of each local coefficients from the total variability of all the coefficients for each local GWR model (Eq. 6) as a local weighting factors reflecting the relative relevance of each predictor locally. This allowed us to apply scores not varying linearly and still interpreting the results easily. This way, the weighting factor obtained in from the proportions could be directly and correctly applied to the Groundwater Depth and Aridity Index layers.*

One example of this wrong interpretation is in lines 376-378, where the authors state that the positive impact of the rivers on the GDV suitability "is due to a higher water availability reflected by the values of omborthermic and aridity indexes. In my view it should be the contrary, i.e. due to a lower water availability, indicating a higher suitability for GDV. Moreover, the positive impact is not visible in the map of Figure 7. And why is there a higher groundwater depth near the river? You would expect groundwater levels to be shallowest near the river.

Another example of this is in discussion section, where the authors state that "The lower suitability to this vegetation in the eastern part of the studied area can be explained by less favorable climatic and geological conditions, resulting from the combination of a high aridity index and low water retention at deep soil layers". It is again the contrary, as the ariditiy index in this (south)eastern area is lower, indicating a higher suitability and therefore higher values on the map of Figure 7. Moreover, it is not clear why the "deep soil" layer is mentioned here now, if only the first soil layer has been analysed.

> *Answer: We appreciate the referee comment and agree with it. Indeed, groundwater levels are expected to be higher near the river, mainly in alluvial aquifers (associated with gentle slopes). However, the opposite also occurs in areas where the rivers are associated with hard rock aquifers (generally associated with steep slopes) and where the relation surface/groundwater is more heterogeneous. The slope predictor, also considered in the presented methodology, distinguishes these occurrences.*

In Figure 7 please indicate how the values were calculated.

> *Answer: A thorough explanation was added in the methods section, in lines 335-341. The explanation in the methods section in the ms reads: "The final GIS multicriteria analysis was performed using the Spatial Analyst Tool by applying local model equations obtained for each of the 6214 coordinates of the Alentejo map (Eq.4),*

*Suitability = Intercept + $coef_1$ * [real value $X_1$] + $coef_2$ * [real value $X_2$] + $coef_3$ * [real value $X_3$] + ...,*

(4)

*with brackets representing the reclassified GIS X layer corresponding to the scoring and $coef_{px}$ indicating the relative proportion for the predictor x.".*

> *The final equation used for the calculation of the suitability map is presented in the results section, in lines 436-437, and is presented in the Equation 1 below.*

$S_{GDV} = Intercept + A_i\ coef_p * [reclassified\ A_i\ value] + O_4\ coef_p * [reclassified\ O_4\ value] + W\ coef_p *$

$[reclassified\ W\ value] + D\ coef_p * [reclassified\ D\ value] + s\ coef_p * [reclassified\ s\ value],$

(1)

If the authors decide to do the analysis per river basin, they should indicate the river basin boundaries in Figure 1.

*Answer: As suggested by reviewer 1, we decided to use a 10 km grid mesh instead. The methodology was corrected in lines 212-213.*

Line 382-383: "this high likelihood was hindered by the type of soil present in that área In terms of soil type in the Tagus basin". That is not true, as the suitability is mostly class 3 in the Tagus river basin.

*Answer: The sentence was deleted according to the new results of the revised manuscript.*

Line 416-419 belongs to the discussion section, not the results section.

*Answer: The paragraph was deleted according to the new validation performed in the revised manuscript.*

Technical corrections
Overall the text is well written and structured, the main comments above concern the content of the manuscript.
Line 47: decreased precipitation

*Answer: Done in line 48.*

Line 56: An integrated multidisciplinary methodology

*Answer: Done in lines 57-58.*

Line 63: do not include

*Answer: Done in line 64.*

Line 167: listed in Table 1

*Answer: Done in line 167.*

Line 169: 2.3.1 Slope and soil characteristics

*Answer: Done in line 170.*

Line 205: division of the basin area by the total stream length

*Answer: Done in line 213.*

Line 244: was evaluated

*Answer: Done in line 255.*

Line 256: based on the selected variables

*Answer: Done in line 268.*

Line 277: score from 1 to 3

*Answer: Done in line 304.*

Line 367: In the GWR model

*Answer: Done in line 450.*

Line 380: with the exception of

*Answer: Done in line 478.*

Line 948: Table 2: Groundwater Depth

*Answer: This table was eliminated from the revised manuscript. The variable Groundwater Depth was, from now on, referenced as W.*

Line 956: suitable areas for GDV

*Answer: Done in line 1190, in Table 4.*

Figure 1: add catchment limits

*Answer: Done in the new version of fig01.*

Figure 4: change soil colours, or combine

*Answer: The map of soil type was removed form Figure 04.*

Line 990: what kind of residuals?

*Answer: This was clarified in lines 1213-1214.*

Figure 7: consider changing the colour coding

*Answer: A new suitability map was calculated, with new colors by classes, and was added as Figure09.*

Figure B1: present the maps in the same order as in Figure 4.

*Answer: Done in Figure B1.*

[revised manuscript text omitted]

**Appendix B**

[Figure]

[Figure]

**Figure B1 – Predictors maps after classification. (a) – Aridity Index; (b) – Ombrothermic Index of the summer quarter and the immediately previous month; (c) – Groundwater Depth ; (d) – Drainage density (e) – Slope.**

---

## Author Response (AR2)

Instituto Dom Luiz, Faculty of Sciences
University of Lisbon
Campo Grande, 1749-016 Lisbon, Portugal
Tel: +351 927464067
E-mail: inesgmarques@fc.ul.pt

June 20ᵗʰ, 2019
Editorial Department of *Hydrology and Earth System Sciences*

Dear Dr. Miriam Coenders-Gerrits,

Please find enclosed the revised version of the manuscript (reference hess-2018-208) entitled *"Mapping the suitability of groundwater dependent vegetation in a semi-arid Mediterranean area"*.

We carefully considered and addressed each reviewer´s comment accordingly. In our joined letter you will find our answers and changes made, indicating the highlighted line numbers.

We are very thankful for giving us the opportunity to improve our manuscript to be accepted in your journal. To facilitate the identification of changes along the manuscript, a version of the manuscript with tracked changes was uploaded in the journal platform.

We once again declare that all the information included in this manuscript is completely original and has been approved by all authors. The authors declare no conflict of interest. This manuscript has not been published previously or concurrently submitted for publication elsewhere.

Thank you for considering this revised manuscript for publication. Please do not hesitate to contact me if you require further details.

With our best regards, sincerely,
Inês Gomes Marques (on behalf of all authors)

Dear Reviewer #2,

Please find enclosed the revised version of the manuscript "Mapping the suitability of groundwater dependent vegetation in a semi-arid Mediterranean area".
We are once again very grateful for your precious and pertinent revision of our manuscript.
All yours suggestions were carefully considered and addressed. In the present letter,
you will find our answers to your comments and changes made, with corresponding lines highlighted. To ease the revision, we highlighted line numbers in yellow in our answers.

We are very thankful for your detailed assessment, which allowed a very significant improvement of the overall manuscript. To facilitate the identification of changes along the manuscript, a version of the manuscript with tracked changes was uploaded in the journal platform.

All the information included in this manuscript has been approved by all authors. The authors declare no conflict of interest.

Thank you for considering this revised manuscript for publication.
Please do not hesitate to contact us if you require further details.

With the authors best regards.
* * *
**Report #1, Reviewer #2**
Suggestions for revision or reasons for rejection (will be published if the paper is accepted for final publication)

The manuscript has been quite significantly altered from its previous version, with many relevant and good aspects added to the model development, suitability map building, map evaluation and sensitivity assessment, in terms of methodology, results and discussion. It really has become an interesting paper to read, with very good use of references throughout. I maintain my opinion that the strongest part is the regression model, and that this model can be applied very well in the future for scenario analysis within the same study area. I continue to believe that the sustainability map is less interesting, because it basically uses the regression model that was already locally optimised using GWR to calculate a new map, but it is only applicable to the study area, precisely due to the local nature of the regression coefficients. Moreover, it does not fit the original vegetation (Kernel density) map as well as the authors claim based on their validation.

*Answer: It is not surprising that the final suitability map does not exactly fit the original Kernel density. Indeed, the proxy species (Quercus suber, Quercus Ilex and Pinus pinea) can perfectly grow under more mesic Mediterranean climate conditions (sub-humid), without relying as much on groundwater to survive as in more xeric areas (semi-arid) (Abad Vinas et al., 2016). Their presence/abundance is only an indication of a possible use of groundwater. This is also why we consider that our final map obtained after the multicriteria analysis provides a more reliable indication of the higher likelihood for groundwater use by facultative deep-rooted phreatophytes species in Alentejo. We also believe our final map also provides a better estimation of the relative contribution of groundwater used by plants to remain alive, than the information given by the model alone. A paragraph was added on* ==617-620== *to better explain the benefits of the final suitability map as compared to the model alone.*

Please allow me to start with these two main concerns I would like to see addressed in the discussion.

1. The calculation of R-squared of the GWR model provides very good results. Notwithstanding, the resulting local coefficients vary largely, from highly positive to highly negative. Moreover, the variations sometimes occur on very small distances. This means that the effect of Ai or groundwater depth on groundwater dependent vegetation can vary from highly positive to highly negative throughout the area and even within very short distances. This seems purely a statistical exercise with apparently little physical meaning and needs to be addressed in the discussion. What does it mean? How then is this method valid and applicable elsewhere?

*Answer: We agree on the fact that our modelling approach is stochastic and can be considered as a "statistical exercise". We also agree on your critic regarding the weak physical meaning of the model coefficients due to their high spatial variability. This is another argument for us not to use the model alone for prediction purposes. We are conscient that the method we developed is only locally optimized and thus difficult to apply in other regions, even under similar climate conditions, unless the methodology is fitted to local conditions/predictors. We modified the manuscript in the discussion (lines* ==630-632==*) and in the "key limitations" section (lines* ==742-746)== *to address those issues.*

2. Given the high weight of the aridity index (Ai) in the regression map, the groundwater dependent vegetation (GDV) suitability map now closely follows the Ai categorized map (Fig. B1a), as also mentioned by the authors. The good agreement observed by the authors between the suitability map and the groundwater depth map, in my view is a coincidence, as the groundwater depth map in fact follows the aridity index map. In other words, in the more humid areas the groundwater level seems shallower, and vice-versa. In addition, the GDV suitability map does not show a good correspondence with the GDV occurrence map (Fig. 1), unlike the previous suitability map that was produced in the first version of the manuscript. In the former map (version 1 of the manuscript) soil type was the most important parameter, but that parameter was now taken out. As a direct consequence, the highest GDV density in the central north now occurs in an area of very poor to poor mapped GDV suitability, whereas in the southeastern area the GDV density is very low in an area of very good suitability. I acknowledge that the reality is always more complex and that the authors already refer to this in their discussion, but please also address the issues I have mentioned.

*Answer:*

*As explained above, the proxy species (Quercus suber, Quercus Ilex and Pinus pinea) can perfectly grow under more mesic Mediterranean climate conditions (sub-humid), without relying as much on groundwater to survive as in more xeric areas (semi-arid). Their presence is only an indication of a possible use of groundwater. The study provided by Pinto et al. (2013) have shown that Cork oak can perfectly thrive were very shallow groundwater is available while suffering drought stress were groundwater source is lower (although using groundwater in both sites). We believe this satisfactory explains the discrepancies between the GDV density and suitability maps you question. We addressed the mismatches between maps in the result section, lines 563-565 and modified a paragraph in the discussion section, in lines 643-659.*

*Abad Viñas, R., Caudullo, G., Oliveira, S., de Rigo, D., 2016. Pinus pinea in Europe: distribution, habitat, usage and threats. In: San -Miguel-Ayanz, J., de Rigo, D., Caudullo, G., Houston Durrant, T., Mau ri, A. (Eds.), European Atlas of Forest Tree Species. Publ. Off. EU, Luxembourg, p. E01b4fc.*

*Pinto C., Nadezhdina N., David J. S., Kurz-Besson C., Caldeira M.C., Henriques M.O., Monteiro F., Pereira J.S., David T.S. Transpiration in Quercus suber trees under shallow water table conditions: the role of soil and groundwater. Hydrological processes, doi: 10.1002/hyp.10097, 2013.*

The fact that the suitability maps fits well with the NDWI map, could be a logical consequence of the fact that the latter represents moisture content in vegetation. Why would the highest stress be indicative for groundwater dependency? Wouldn't you expect stress to decrease if the trees have access to groundwater?

*Answer: Figure 10 does not present NDWI values, but anomalies considering the months of June, July and August of the extremely dry year of 2005, in reference to the median NDWI value of the same months over the period 1999-2009 (lines 544-545)). In June of the extreme dry year 2005, GDV vegetation experienced the highest moisture stress, as observed on Figure 10a by the negative NDWI anomaly values. GDV still contains moisture however, that changes/decreases with the onset of the summer period (aggravated by the dry winter-spring of 2005), thus reaching a point in August were the GDV has a very low water content, as expected in the end of the drought season (~null anomaly on Figure 10c). Oppositely, the vegetation over areas that do not manage to cope with summer drought (bare soil, grassland, shrubs...) uses to have the lowest moisture content since June until August with no change (null anomaly indicated in green that remains green from June to August on figure 10a-c). Therefore the GDV shows the highest absolute NDWI anomaly (highest leaf water loss), in spite of the use of groundwater to survive. Further former studies by co-authors of the present work have already shown that groundwater uptake by trees only take place in late June after the onset of the drought period (Kurz-Besson et al. 2006 & 2014, Otieno et al. 2006, David et al. 2013, Pinto et al. 2013). Those studies have also shown that trees grew new roots in deeper soil layers only after trees experienced drought stress. In extreme dry years, the piezometric drawdown is*

*expected to difficult GDV's physiological performances (Antunes et al. 2018). We are confident that those studies are in agreement with the NDWI anomaly validation maps provided. Nevertheless, we re-write the paragraph 3.5 for more clarity and added the references cited here above in the manuscript to support our arguments (Lines ==547-565==). We also modified changed figure 10 colours and caption in order to highlight the NDVI anomaly behavior aiming to avoid misleading issues (lines ==1241-1245==, ==1327-1331==).*

*Otieno, D.O., Kurz-Besson C., Liu J., Schmidt M.W.T., Lobo-do-Vale R., David T. S., Siegwolf R., Pereira J.S., Tenhunen J.D. (2006) Seasonal variations in soil and plant water status in a Quercus suber L. stand: roots as determinants of tree productivity and survival in the Mediterranean-type ecosystem. Plant and Soil 283: 119-13*

*Kurz-Besson C., Otieno D., Lobo-do-Vale R., Siegwolf R., Schmidt M.W.T., David T. S., Soares David J., Tenhunen J., Pereira J. S., Chaves M. (2006) Hydraulic lift in cork oak trees in a savannah-type Mediterranean ecosystem and its contribution to the local water balance. Plant and Soil 282: 361-378.*

*Pinto C., Nadezhdina N., David J. S., Kurz-Besson C., Caldeira M.C., Henriques M.O., Monteiro F., Pereira J.S., David T.S. (2013) Transpiration in Quercus suber trees under shallow water table conditions: the role of soil and groundwater. Hydrological processes.*

*David T.S. Pinto C.A. Nadezhdina N. Kurz-Besson C. Henriques M.O. Quilhó T. Cermak J. Chaves M.M. Pereira J.S., David J.S. (2013) Root functioning, tree water use and hydraulic redistribution in Quercus suber trees: A modeling approach based on root sap flow. Forest Ecology and Management 307, 136–146.*

*Kurz-Besson C., Lobo do Vale R., Rodrigues L., Almeida P., Herd A., Grant O.M., David T.S., Schmidt M., Otieno D., Keenan T., Gouveia C., Mériaux C., Chaves M.M., Pereira J.S. (2014). Cork oak physiological responses to manipulated water availability in a Mediterranean woodland. Journal of Agricultural and Forest Meteorology 184, 230-242.*

*Páscoa P. Gouveia C., Kurz-Besson C. Identificação de vegetação dependente de água subterrânea na península ibérica através de deteção remota. 10º Símposio de Meteorologia e Geofísica da APMG, Lisboa, Portugal. 2017,
https://drive.google.com/file/d/0B4ZF89Veh6ziZVVCbUxBZXh1MTA/view*

Some other comments are given below:

The abstract is well written.

The introduction provides a very good overview on the need of study, but could mention the other work/studies carried out so far in the field. That is currently limited to one sentence (ln 127-129), so that the paper does not show the added value of the implemented methodology as compared to existing studies, some of which are actually referred to later on in the manuscript (e.g. Barron et al., 2014; Condesso de Melo, 2015; Costa et al., 2008; Doody et al., 2017). Therefore, no new references are needed.
**Answer:**
*The introduction section was slightly restructured. We rearranged the short overview of the studies carried out in the field (now in lines ==54-78==), avoiding turning the introduction any longer. We also added a new reference based on field surveys and showing that Pinus pinea relies on groundwater to cope with summer droughts. We also indicated the added value of the*

*implemented methodology in lines ==132-139== and further improved the end of the introduction section in lines ==140-154==.*

In material and methods, section 2.3.1, attributing a low GDV suitability score to soils of high clay content can be debated. Soils of a finer texture will have large extinction depths due to an increased capacity of capillary rise. I would expect coarser soils to have vegetation of lower groundwater dependency. Please briefly elucidate on this aspect.

*Answer: We agree with your comment and the debate in the matter. Nonetheless, in this specific geographical region, deep rooting species reaching deep soil layers or groundwater are disfavored in waterlogged soils highly favored by clay content (Garcia et al. 2017; Ignacio Perez-Ramos & Marañón 2009; Dinis et al. 2014). We also believe that soils rich in clay will rather favor non-GDV species for providing more available water in shallow soil depths. This is not happening in sandy soils, therefore we gave a better score to those. We had already briefly justified this choice in our former version of the manuscript, (now in lines ==208-212==) and added a few more words to better justify our scoring choice.*

Garcia et al. 2017, https://ir.library.oregonstate.edu/downloads/wp988k05k;
Ignacio Perez-Ramos & Marañón 2009, https://www.researchgate.net/publication/222234643;
Dinis et al. 2014 https://core.ac.uk/download/pdf/62473102.pdf, (page 60)

In the model development (material and methods, section 2.5), how many data points are used (and what is the search radius) for the calculation of local model coefficients?

*Answer:*
*The number of points (6214) used to fit the model was already in the previous version, (now in lines ==311-312==). We corrected the sentence for the lector to understand that ultimately 6214 points were used to fit the model (line ==311==).*
*Before fitting the GWR model an Adaptive Kernel was applied to the data to find a search radius (as explained in lines ==323-324== of the manuscript) that would minimize the error of the localized regressions. The adapted search radius, given locally, was obtained through minimization of the CrossValidation score. We improved the methodological explanation in lines 294-295 of the manuscript.*

In section 2.7 of material and methods briefly explain for what purpose the NDWI anomaly map was calculated.
*Answer: We added a sentence to include this missing information in paragraph 2.7 on lines ==391-393==.*

Please explain why you select slope (s) rather than soil thickness (S), if the latter has a higher correlation with principle component axis 2 (PC2).

*Answer: As explained in lines ==318-321== of the manuscript, variables were selected under a sequential procedure. Both slope and thickness did not show correlation values higher than 0.4 and therefore where not discarded from the initial variables selection. If predictors showed correlations below 4, than the ones with the lower correlation values would be chosen. Thickness was removed from the final variable choices because it showed higher correlations with the remaining variables, as opposed to slope that showed lower correlations with the remaining variables.*

What happens to R-squared when reducing the set to four or even three variables? Given the large weight of Ai and O4 I would expect the impact to be small. Have you considered using a reduced set? This would largely facilitate the application of the method in other areas.

*Answer: The model performance assessed with global $R^2$ was little affected when only 2 or 3 predictors were used, remaining close to 0.99. Also, our modelling approach in this manuscript was only performed to provide weights for the GIS layers included in our final multicriteria analysis. On our last revision, we removed the soil type from the model equation because it drastically weakened its performance. The remaining predictors, however, did not affect the performance of the model as much, with $R^2$ remaining between the range of 0.98 to 0.99. We thus choose to keep the remaining predictors in the model (especially the groundwater depth) because of the objective of our study, in spite of their lower contribution to the model.*

Other minor comments and technical corrections:

Ln 17: delete the word "scenarios"
*Answer: Done, now line 38.*

Ln 19: delete the words "the density of"
*Answer: We improved the sentence, now line 19-20.*

Ln 25: "closely followed": this is not true. The other three parameters (groundwater depth, drainage density and slope) follow at a large distance, i.e. they are of much lower importance in the regression model.
*Answer: We corrected the sentence as "Climatic indices were the main drivers of GDV density, followed with a much lower influence by groundwater depth, drainage density and slope", now in line 25.*

Ln 28: "relative proportion". Please briefly clarify what it means. Is it the local coefficient divided by sum of local coefficients? When negative, do you use absolute values (which would make sense)? This needs to be explained in detail in section 2.6 (pg 11 ln 329-341).
*Answer: This as been clarified in lines 372-373, by adding "The relative proportion of the local coefficient x was calculated as the ratio between the modulus of the local coefficient x and the sum of the modulus of all local coefficients."*

Ln 60: include
*Answer: Corrected, Line 55*

Ln 61: "subsurface groundwater" seems a pleonasm, although I understand what you mean, when comparing it to "surface groundwater". Perhaps you could consider using the terms "emerging groundwater" vs. "resident groundwater".
*Answer: We totally agree with this suggestion. Therefore we modified the text accordingly along the manuscript (lines 56, 57, 74, 77, 120, 346).*

Ln 62: "a visible source"
*Answer: Done, line 58*

Ln 64-65: place the references after GDE
*Answer: Done, line 71*

Ln 74: "relying on", perhaps use "entirely relying on"
*Answer: Done, line 83*

Ln 76: "root system"
*Answer: Done, line 85*

Ln 115: "rising temperature"
*Answer: Done, line 125*

Ln 129-130: rephrase "coefficients proportions", e.g. to "coefficients as proportion of total sum of absolute coefficients".
*Answer: Done, line ==147-148==*

Ln 184: "low drainage capacity", "high clay fraction"
*Answer: Done, line ==208-209==*

Ln 325-328: lower drainage density leads to higher suitability, which is correct, but the explanation is incorrect, as the explanation in fact suggests the opposite, or so it seems.
*Answer: We improved the sentence, lines ==355-359==.*

Ln 342-343: I suggest using "representing" instead of the word "referred".
*Answer: Done, line ==375-376==.*

Ln 402: and in the south?
*Answer: Done in line ==439==.*

Ln 422: the maximum value on the map seems much higher than the value indicated in the text (0.714).
*Answer: We thank the reviewer for noticing the mistake. Indeed this was a typo, the true maximum value, excluding two outliers, is 1.166. This information was corrected in line ==459==.*

Ln 488: I suggest changing to: "poor suitability to GDV, corresponding to"
*Answer: Done, line ==525==.*

Ln 572: "did not only allow"
*Answer: Done, line ==625==.*

Figure 3: What are the units in this figure?
*Answer: We add a sentence to the figure legend as "The scale unit represent the number of occurrences per 10km search radius (~314 km2)" lines ==1224-1225==, ==1294-1295==. Note that ICNF forest inventory only provided information on the presence of each dominant and secondary species on 500m mesh points and their corresponding cover percentage. Therefore, on an area of 1 km2 the maximum occurrence possible is 4, thus on our map the maximum value is 4\*314=1256. We therefore also modify the M&M section on heatmap accordingly, lines ==176-177== .*

Figure 4: The reference to the different maps in the figure title is incorrect. Figure 4a is aridity index, not soil type, etc.
*Answer: We truly apologize for this mistake. This has been now corrected, lines ==1226-1227==, ==1298-1301==.*

Figure 10: I would not use green to indicate highest stress.
*Answer: Colors on Figure 10 have been modified, line ==1326==.*

Table 4: Values for slope and aridity index are incorrect in the table (the order of the scores 1-3 is inversed, as can be seen in the maps of Fig. B1, which are correct).
*Answer: Thank you for noticing these mistakes. This has been corrected in Table 4, line ==1281-1282==.*

Dear Reviewer #3,

Please find enclosed the revised version of the manuscript "Mapping the suitability of groundwater dependent vegetation in a semi-arid Mediterranean area".

We did our best to carefully address all your concerns. In the present letter, you will find our responses to each comment and changes made in the manuscript. To ease the revision, we highlighted line numbers in yellow in our answers.

We also attempted to provide a better evaluation of the importance of each predictor in the final model and improved the discussion section accordingly.

To facilitate the identification of changes along the manuscript, a version of the manuscript with tracked changes was uploaded in the journal platform.

All the information included in this manuscript is completely original and has been approved by all authors.

Also, we thank you for considering this revised manuscript for publication.

Please do not hesitate to contact us for any further needed detail.

With our best regards, sincerely
* * *
**Report #2, Reviewer #3**
Suggestions for revision or reasons for rejection (will be published if the paper is accepted for final publication)
The whole paper should be condensed and restructured. The relevance of the study should first be established in introduction by presenting the field of vegetation suitability mapping in ecology (with a better review of previous research), establish the niche by indicating the gap in the present body of literature, and finally present the aim and the approach of the study.
Next, the methodology should be clearly established, starting by the choice of the modeling method which appears to be a linear regression, improved in order to take into account spatial correlation of the explaining variables.

*Answer:*
**The Geographically weighted regression (GWR) extends the ordinary least squares (OLS) regression by considering spatial nonstationarity in variable relationships and allowing the use of spatially varying coefficients in linear models while minimizing spatial autocorrelation. We added a few words in lines** 307-309**.**
**The normal distribution for predictors is only recommended while using the linear OLS model when the model is used for statistical inference or to calculate confidence intervals . In our study, we only used the modeling approach to provide weighting factors for the GIS multicriteria analysis performed to obtain the final suitability map. We thus assumed that for such purpose predictors' normality was not necessary.**

The data should first be plotted to illustrate their departure from Normality. In second instance, the choice of transformation methods should be justified.
*Answer:*
**Although our dependent variable (Kernel density) did not rigorously match a normal distribution after root square transformation, the distribution shape was approximated to meet the linear model assumption (see Figure below). Also we relied on the article by Li et al. 2012, (https://iovs.arvojournals.org/article.aspx?articleid=2128171), which stipulates that when the dependent variable is not distributed normally, the linear regression remains a statistically sound technique in studies of large sample sizes (i.e., >3000), which can be used anyway, even if the normality assumption is violated.**
**The square-root transformation of the response variable was already indicated in lines** 297-299**.**

[Figure]

Regarding the criterium of groundwater availability in particular:

1)      Why should the soil type, aquifer permeability or aquifer transmissivity be relevant for the growth of groundwater dependent vegetation?
*Answer: We believe it would be relevant for the presence and permanence of more superficial groundwater accessible to roots.*

2)      If groundwater levels need to be used as suitability criteria for a type of vegetation, the fluctuation regime need to be established (for example mean levels, 5% low and high quantile determine over a given time period)
*Answer: We think the reviewer is correct, however groundwater depth data retrieved from large diameter wells (blue triangles in figure 02) had only one single measurement. These data points covered most of the study area, thus there was not enough data to establish a temporal fluctuation regime. This weakness was already discussed in lines 660-674.*

3)      The interpolation method needs to be better described. A suggestion is to follow the method used by Peterson and Barnett [2004]
*Answer: The method suggested by Peterson and Barnett [2004] (Kriging with External Drift) was also tried with the groundwater datasets used in this study. However, the resulting map of groundwater depth showed incoherent values, therefore we proceeded with the double approach: Ordinary Kriging for karts and porous aquifers and linear regression for undifferentiated geological type. We added a further explanation of the Ordinary Kriging method to lines 233-235 : "The ordinary kriging was calculated using a semivariogram in which the sill, range and nugget were optimized to create the best fit of the model to the data."*

4)      Why is the drainage density relevant in the method if the water table levels are known?
*Answer: Groundwater supply at deeper levels is important for groundwater dependent vegetation survival, since there is no other source of water during the dry season. However, when a large river system is present, water will be available closer to the surface. As written in lines 238-239, the drainage density is a measure of how well the water in the basin is drained by the stream channels, thus affecting infiltration process. Therefore, this predictor provides insights on well the superficial soil layers will be fed by stream water. On another hand, the vegetation dependent on groundwater studied in this manuscript can use water from the vadose zone at a rooting depth reaching up to 15m. The depth to groundwater (piezometer level) allowed the exclusion of GDV where groundwater was deeper than 15m.*

Finally, the argumentation needs to be considerably improved. For example expressions such as 'subsurface groundwater' should be avoided and expressions such as 'surface groundwater' (line 60) or 'subsurface groundwater dependent vegetation' are meaningless.

*Answer: As suggested by both reviewers, we renamed the term "subsurface groundwater" as "resident groundwater" being the groundwater beneath the soil surface, as opposed to "emerging groundwater" being the groundwater above the soil surface. We changed the text accordingly throughout the manuscript (lines ==56, 57, 74, 77, 120, 346==).*

[revised manuscript text omitted]